# BoundaryDPT: Pushing the Boundaries of Depth Pruning for Vision Transformers

## Abstract

While prior studies have successfully compressed vision Transformers (ViTs) through various pruning techniques, most have concentrated on width pruning to achieve significant reductions in model size. Depth pruning, which involves the removal of entire layers from a ViT, is notoriously difficult for accuracy recovery, although depth pruning usually leads to higher speedups of compressed ViTs. Consequently, existing joint approaches that incorporate both width and depth pruning have exhibited limited acceleration ratios due to the inefficiencies of previous depth pruning methods. To tackle the challenges in depth pruning, this work introduces BoundaryDPT, a novel depth pruning method by targeting redundancy of both attention layers and non-linearity within ViTs. To the best of our knowledge, we are the first to propose the pruning of activation function layers in ViTs. By reducing the redundancy of nonlinearity, instead of directly targeting linear layers in ViTs, the depths of ViTs are naturally reduced without incurring dimension mismatch. Moreover, we present a two-stage joint pruning method designed to address the heterogeneity of attention layers and activation function layers. Comprehensive experiments on ImageNet1k, CIFAR-100, and ADE20K have validated our methods. Firstly, BoundaryDPT achieves a 1.58× speedup for DeiT-B while maintaining accuracy, and a 1.39× speedup for DeiT-S with nearly lossless accuracy degradation. Furthermore, when combined with width pruning (referred to as BoundaryDPT+), our method sets a new state-of-the-art record in ViT pruning. For instance, BoundaryDPT+ enhances the acceleration ratio from 4.24× to 5.19× for the Isomorphic-Pruning-2.6G configuration while maintaining near-lossless accuracy, establishing new benchmarks in extreme ViT compression.

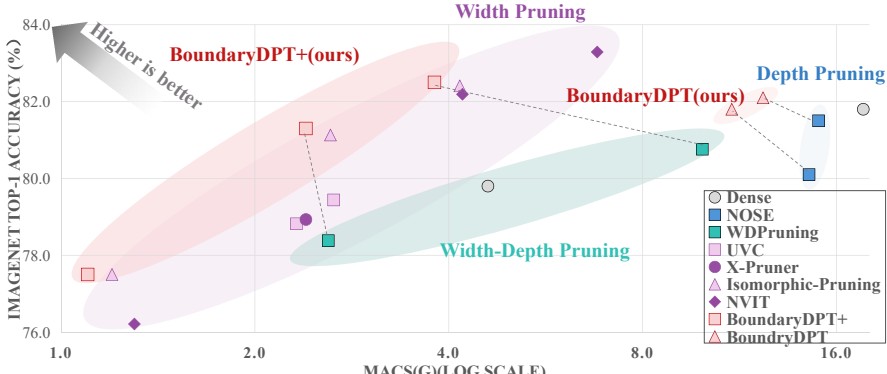

Figure 1: Compared with other work, our BoundaryDPT and BoundaryDPT+ offers an state-of-the-art accuracy-speedup Pareto frontier.

## 1 Introduction

Vision Transformers (ViTs) (Kolesnikov et al., 2021; Liu et al., 2021; Touvron et al., 2021; Han et al., 2021; Chen et al., 2021a; Li et al., 2022; Wang et al., 2022) have demonstrated remarkable

performance across various domains. However, their large parameter counts and high computational costs lead to extended inference latency. Structured pruning (He & Xiao, 2023) is effective for model compression while maintaining hardware compatibility.

**Depth pruning and its limitation.** As a kind of structured prunign, depth pruning denotes removing entire layers from ViTs. Compared to width pruning which only reduces channels or attention heads inside a layer, depth pruning delivers significantly higher speedups under equivalent sparsity budgets, as shown in Figure 2 (a). However, depth pruning typically incurs substantial accuracy degradation, especially with **aggressive layer removal**. Consequently, comprehensive methods that integrate both width and depth pruning are also limited by the depth pruning challenges, resulting in suboptimal performance.

**Joint depth pruning matters.** While prior research attributes the accuracy collapse in depth pruning to coarse granularity (He & Xiao, 2023; Mao et al., 2017), we challenge this perspective. Our analysis reveals that the true bottleneck lies in the neglect of joint pruning of different layers in a ViT by considering cross-layer heterogeneity. As shown in Figure 7, individually pruning attention layers or activation function layers leads to drastic accuracy drops at high pruning ratios. In contrast, joint pruning of these two types of layers with our method significantly enhances accuracy retention while maintaining efficiency.

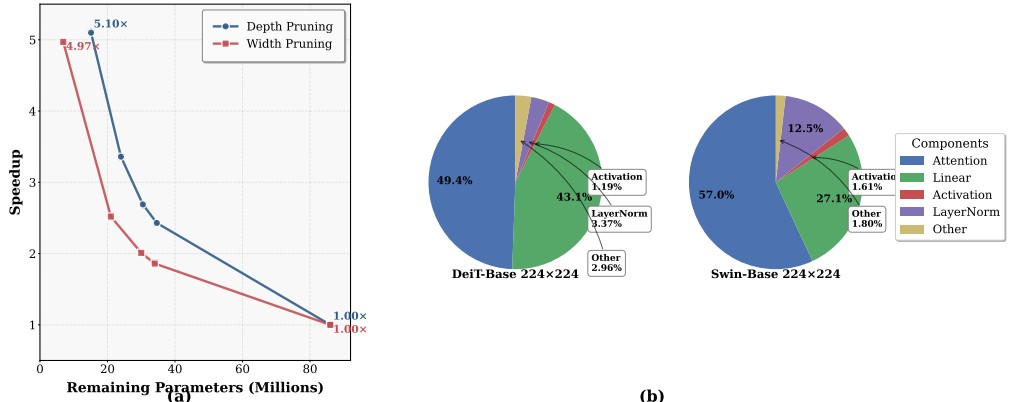

Figure 2: Practical inference speed analysis of ViTs. (a) Speedup comparsion: Depth pruning exhibits significantly higher speedup efficiency than width pruning. (b) Latency breakdown of ViTs.

**Dimension mismatch hinders joint depth pruning.** As shown in Figure 2 (b), the two most types of time-consuming layers in ViTs are attention layers and linear layers, which together account for over 50% of total inference time. While joint depth pruning of linear layers and attention layers is necessary, direct simultaneous removal may create *dimension mismatch*. As illustrated in **Figure 3**, if the first linear layer of a feedforward network (FFN) block in ViTs is removed, the output tensor from the previous attention layer cannot be passed through the second linear layer in the FFN. Similarly, pruned second linear layers prevent the output tensor from passing through the subsequent attention layers. In a word, dimension mismatch renders jointly depth-pruned ViTs unworkable..

**Our contribution.** To address the conundrums of accuracy recovery and dimension mismatch in depth pruning, our contributions are threefold:

- **Joint depth pruning of attention and activation function layers is proposed.** In particular, we tackle dimension mismatch by removing activation function layers situated between two linear layers, which allows for the natural merging of those linear layers to reduce model depth while aligning the dimensions of attention layers. Besides, to the best of our knowledge, we are the **first** to identify and mitigate the redundancy of the activation function layers during joint pruning in ViTs.

- **The heterogeneity in joint depth pruning is revealed and addressed.** We identify two unique phenomena related to the heterogeneity in joint depth pruning: gradient disparity and recovery asymmetry. Such heterogeneity has never been examined in the literature. In light of this, we introduce BoundaryDPT, a two-stage method featuring a model accuracy predictor to manage heterogeneity.

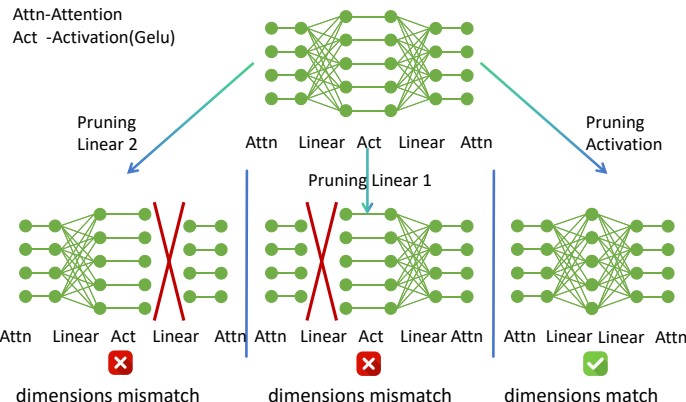

Figure 3: The visualization of dimensions mismatch.

- **Two key state-of-the-art records are established.** With BoundaryDPT, the depth-pruned DeiT-base achieves up to 1.6x speedup while maintaining lossless accuracy, which is the state-of-the-art among depth pruning works. More importantly, building on Boundary-DPT, we further present BoundaryDPT+, a depth-width pruning pipeline that **establishes a new state-of-the-art benchmark for extreme ViT compression**, as demonstrated in Figure 1. BoundaryDPT+ enhances the ViT inference speedup from 4.60x to 5.44x for the Isomorphic-Pruning-2.6G configuration while achieving near-lossless accuracy.

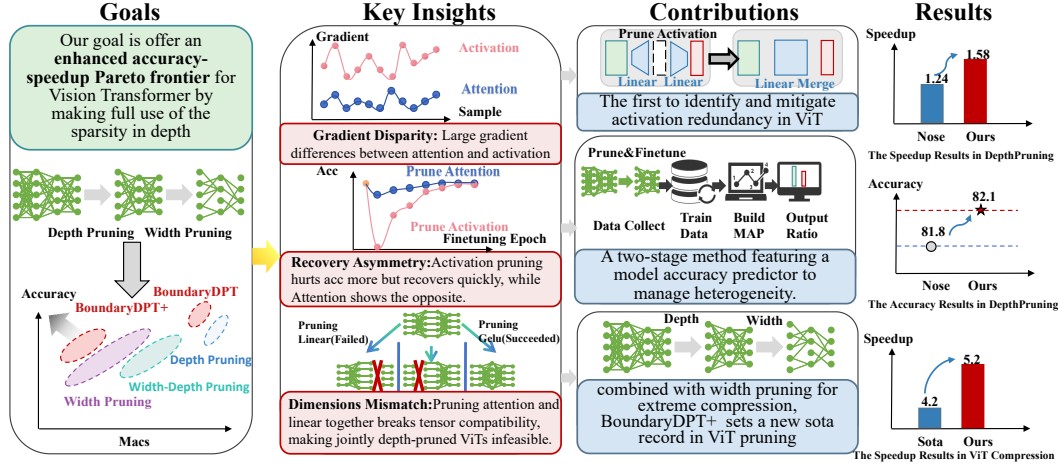

Figure 4: The complete methodological framework of BoundaryDPT and BoundaryDPT+, involving the research goals, key insights, contributions, and results.

## 2 RELATED WORK

**Structured Transformer pruning** Depending on the pruning targets, current ViT structure pruning methods can be divided into four categories: 1) width pruning, such as NViT (Yang et al., 2023), IsomorphicPruning (Fang et al., 2024) and some other work (Chavan et al., 2022; Zhu et al., 2021; Yu et al., 2022b; Zheng et al., 2022), which refers to typically reducing weight amounts or computations within a component by groups, such as channel pruning or attention head pruning. 2) depth pruning, such as NOSE (Lin et al., 2024), which involves the complete removal of entire components, such as pruning attention layers or linear layers. 3) width-depth joint pruning, such as WD-Pruning (Yu et al., 2022a); and 4) token pruning/merging, such as TOME (Bolya et al., 2023), GTP-ViT (Xu et al., 2024b),DynamicViT (Rao et al., 2021),Evo-ViT (Xu et al., 2022),EViT (Liang et al., 2022), TPS (Wei et al., 2023) and some other work (Chen et al., 2023a; Wu et al., 2024), which reduces

the number of input tokens before propagation between different Transformer blocks. However, the first three structural pruning approaches each have inherent limitations: width pruning achieves relatively low speedup ratios under equivalent pruning rates; depth pruning, while providing higher acceleration, often results in significant accuracy degradation that is difficult to recover; and width-depth joint pruning usually fails to fully leverage the advantages of either strategy due to the need to balance their respective deficiencies, thus preventing optimal ViT compression.

**Activation function layer pruning** . Activation function layers represent the non-linear capabilities of neural networks. Removing activation function layers situated between two linear layers allows for the natural merging of those layers, thereby reducing the model depth (Fu et al., 2022; Dror et al., 2021; Kim et al., 2023). However, **clipping activation function layers for joint pruning in CNNs intrinsically differs from that in ViTs**. In CNNs, the challenge arises from increased convolutional kernel size after merging (Kim et al., 2024). In contrast, the challenge in ViTs is the heterogeneity between attention layers and activation function layers, which is discussed in Section 3. The common activation function layer pruning and accuracy recovery methods used for CNNs are entirely ineffective for ViTs, as shown in Table 9.

**Other pruning techniques for Transformer** Besides structured pruning methods, other pruning techniques for Transformer compression include unstructured (Chen et al., 2021b; Frantar & Alistarh, 2023; Sun et al., 2024a) and semi-structured pruning (Zhao et al., 2024; Xu et al., 2024a; Lu et al., 2023; Liu et al., 2025). However, these methods require specialized hardware support—unstructured pruning produces sparse matrices needing dedicated acceleration, while semi-structured pruning like 2:4 sparsity is limited to specific GPU architecture, both lacking generalizability on standard hardware. Notably, there is another category of works called activation pruning (Chen et al., 2023b; Ganguli & Chong, 2024), which dynamically removes neurons or feature maps during inference, whereas our work statically removes activation function layers at the architectural level, **representing a fundamentally different paradigm from these dynamic approaches.**

## 3 KEY OBSERVATIONS ON HETEROGENEITY

Pruning redundant activation function layers alongside attention layers is non-trivial. Conventional importance metrics, whether gradient-based or handcrafted, are often insufficient in addressing the heterogeneity between attention layers and activation function layers. Specifically, We highlight two key observations embodying heterogeneity that simultaneously constitute critical challenges in designing an effective joint pruning method.

**Observation 1: gradient disparity. The attention layers and activation function layers exhibit significant differences in gradient scales during backpropagation, which can lead to biased importance estimations and suboptimal pruning decisions when employing training-based search strategies.** To be specific, we assign each layer in a ViT a learnable importance weight parameter and record the gradient magnitudes for each layer after one epoch of forward and backward propagation. As

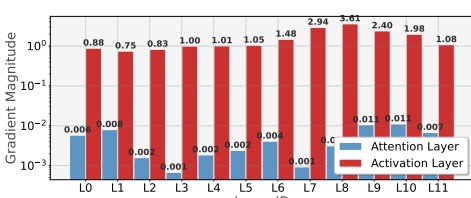

Figure 5: Backpropagation gradient magnitudes across attention and activation layers.

illustrated in Figure 5, the gradient magnitudes of activation function layers substantially exceed those of attention layers, a trend that persists across increased training epochs. Notably, we also establish a unified theoretical account of the two observations on heterogeneity, please see Appendix I for details.

**Challenge 1: failure of gradient-based pruning metrics.** Given the unique training dynamics in ViTs, using gradients to evaluate the importance of these two types of layers results in significantly biased outcomes, with all attention layers constantly outweighing linear layers.

**Observation 2: recovery asymmetry. Attention layer pruning causes moderate initial accuracy drops but requires extensive retraining for recovery, while activation function layer pruning results in severe initial degradation (often $\geq 90\%$) but enables rapid recovery during fine-tuning.** To exemplify this phenomenon, we individually prune each layer type in DeiT-S and record accuracy before and after 10-epoch fine-tuning. Figure 6(a) demonstrates that pruning activation

function layers leads to catastrophic accuracy declines, with most layers retaining only 0.1% accuracy. Conversely, pruning attention layers results in significantly milder degradation, with most layers experiencing less than 5% accuracy loss. However, as shown in Figure 6(b), after just 10 epochs of fine-tuning, the accuracy of both layer types converges to comparable levels, highlighting the rapid recovery capability of activation function layers following pruning.

**Challenge 2: failure of short-sighted pruning metrics.** The substantial asymmetry in accuracy dynamics between these two types of components in ViTs renders conventional handcrafted pruning metrics ineffective for joint pruning, as these metrics typically reflect only immediate post-pruning accuracy retention.

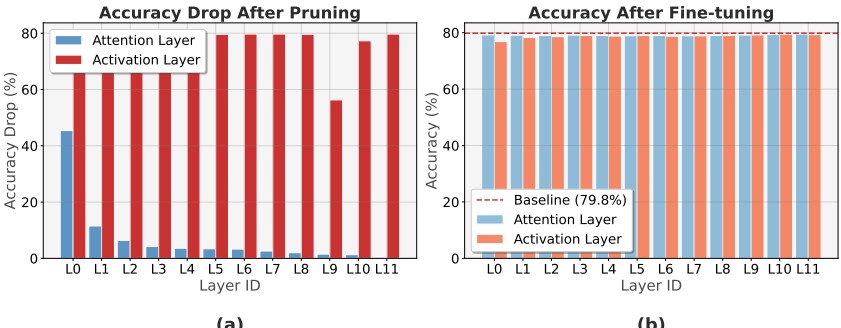

Figure 6: Recovery asymmetry phenomena in DeiT-Small pruning: (a) Accuracy drop after pruning attention layers and activation function at different positions. (b) Model accuracy after 10-epoch fine-tuning following pruning of attention layers and activation function layers at different positions.

## 4 METHODS

**Overview of BoundaryDPT.** Our method jointly prunes attention and activation function layers through a two-stage pipeline: (1) identification of redundant layers, and (2) optimization of the pruned model via fine-tuning and layer merging to accelerate inference. A detailed visualization of the BoundaryDPT pipeline is provided in Figure 8 in Appendix B.

**Key design principles from heterogeneity observations.** Our method is motivated by two principles: 1) Avoid direct cross-type layer importance comparison, especially when using gradient-based metrics. 2) Layer importance should be evaluated based on the final accuracy of fine-tuned pruned ViTs, rather than the immediate accuracy after pruning.

**Alignment with design principles.** The Stage 1 of our method is further divided into two steps: pruning budget allocation (Step 1) and specific redundant layer removal (Step 2). 1) Step 1 utilizes a model accuracy predictor, to help establish the optimal quantities of attention and activation function layers to prune based on the accuracy recovered after fine-tuning, directly implementing Principle 2. 2) Importantly, Step 1 focuses solely on determining the pruning quantities for each layer type, avoiding cross-type importance comparisons, thus adhering to Principle 1. 3) Step 2 employs gradient-based methods only within homogeneous layer groups, i.e., either attention or activation function layers, ensuring compliance with Principle 1 throughout the pruning process.

### 4.1 PROBLEM FORMULATION

Assume a ViT $\mathcal{F}$ has $L$ Transformer blocks. The $l$-th transformer block $f_l$ typically integrates: 1) two linear layers, i.e., $\mathcal{L}_{i,1}$ and $\mathcal{L}_{l,2}$, 2) an attention layers $\mathcal{A}_l$, 3) a GELU activation function layer $\mathcal{G}_l$. For simplicity, we omit auxiliary components such as residual connections and layer normalization. Thus,

$$f_l(\cdot) := \mathcal{L}_{l,2} \circ \mathcal{G}_l \circ \mathcal{L}_{l,1} \circ \mathcal{A}_l \circ f_{l-1}(\cdot), \tag{1}$$

where $\circ$ denotes the function composition operation. As evident from Figure 2(b), the primary computational overhead in Transformer blocks stems from the attention layers and linear layers. We aim to reduce computational overhead through joint pruning of attention layers and redundant

GELU activation function layers. That is,

$$
\begin{aligned}
f_l^{\hat{m}}(\cdot) &:= \mathcal{L}_{l,2} \circ h\left(\mathcal{G}_l\right) \circ \mathcal{L}_{l,1} \circ h\left(\mathcal{A}_l\right) \circ f_{l-1}^{\hat{m}}(\cdot), \\
h\left(\mathcal{G}_l\right) &= \hat{m}_l^g \circ \mathcal{G}_l + (1 - \hat{m}_l^g) \circ I, \\
h\left(\mathcal{A}_l\right) &= \hat{m}_l^a \circ \mathcal{A}_l + (1 - \hat{m}_l^a) \circ I, \\
\hat{m}_l^a, \hat{m}_l^g &\in \{0, 1\}. \\
\boldsymbol{M} &:= \left(\{\hat{m}_l^a\}_{l=1}^L, \{\hat{m}_l^g\}_{l=1}^L\right)
\end{aligned}
\tag{2}
$$

Notably, $\hat{m}_l^g$ and $\hat{m}_l^a$ are the binary masks. A mask's value of 1 preserves the corresponding layer, while 0 prunes it. $I$ denotes the identity mapping, i.e., a direct skip connection when the layer is pruned. $\boldsymbol{M}$ is the set of all the binary masks. A ViT with such masks is referred to as $\mathcal{F}^{\hat{m}}$. Thus, the joint pruning problem is formulated as:

$$
\begin{aligned}
&\underset{\boldsymbol{W}, \boldsymbol{M}}{\arg\min}\, \ell\left(\mathcal{F}^{\hat{m}}(\boldsymbol{X}, \boldsymbol{W}, \boldsymbol{M}), \boldsymbol{Y}\right), \\
&\text{s.t.} \quad \tilde{m}^a + \tilde{m}^g = k/L, \\
&\tilde{m}^a = \frac{1}{L}\sum_{l=1}^L \hat{m}_l^a, \tilde{m}^g = \frac{1}{L}\sum_{l=1}^L \hat{m}_l^g,
\end{aligned}
\tag{3}
$$

where $(\boldsymbol{X}, \boldsymbol{Y})$ are respectively samples and labels of fitting data, and $\boldsymbol{W}$ is the weights of $\mathcal{F}^{\hat{m}}$. $\ell(\cdot)$ refers to the loss function to measure data fitting quality, e.g., cross-entropy loss for classification tasks. $k$ is the sparsity constraint that determines the total number of layers, i.e., the sum of attention layers and activation function layers, to be retained after depth pruning. $\tilde{m}^a$ and $\tilde{m}^g$ represent the global retention ratios of attention layers and activation function layers, respectively.

## 4.2 STAGE 1: IDENTIFICATION OF REDUNDANT LAYERS

**Pruning budget allocation.** Given a sparsity constraint, the primary question is to determining how many attention layers versus activation function layers should be pruned. Naturally, the allocation scheme should maximize the accuracy of depth-pruned Transformers, i.e.,

$$
(\tilde{m}^a, \tilde{m}^g) = \underset{\tilde{m}^a, \tilde{m}^g, \Theta}{\arg\max}\, \mathcal{P}(\tilde{m}^a, \tilde{m}^g; \Theta),
\tag{4}
$$

where we call $\mathcal{P}(\tilde{m}^a, \tilde{m}^g; \Theta)$ the **Model Accuracy Predictor (MAP)**, which predicts the model's accuracy given $(\tilde{m}^a, \tilde{m}^g)$. $\Theta$ represents the parameters of the MAP itself. **The discreteness of $\tilde{m}^a$ and $\tilde{m}^g$ creates a very finite solution space. Once MAP is properly characterized, we can easily traverse all combinations of $(\tilde{m}^a, \tilde{m}^g)$ to obtain the optimal solution maximizing MAP values.**

*Polynomial approximation of MAP.* Generally, it is intractable to obtain an exact expression for the MAP, as determining the precise expression requires a substantial amount of experimental data, and the cost of each training session is high. Here, we approximate the MAP through polynomial fitting combined with rapid data collection. That is, the MAP can be expressed by the following polynomial:

$$
\mathcal{P}(\tilde{m}^a, \tilde{m}^g; \Theta) = \sum_{i,j=0}^{\kappa} \theta_{ij} (\tilde{m}^a)^i (\tilde{m}^g)^j,
\tag{5}
$$

where $\kappa$ denotes the degree of the polynomial. To avoid overfitting during subsequent function regression, a degree of 2 is typically sufficient. $\theta_{ij}$ represents the coefficients of the different terms. By approximating the expression of the MAP using a polynomial, the MAP becomes resolvable via regression (Tyagi et al., 2022). We provide a solid **theoretical grounding for such approximation**. For details, please refer to Appendix H.

*Light-weighted data collection for MAP regression.* Although MAP can be approximated using a polynomial, collecting regression data remains time-consuming due to high retraining costs for each pruned ViT configuration. To efficiently collect $((\tilde{m}^a, \tilde{m}^g), \mathcal{P})$ data, we design a lightweight data-collection procedure: 1) We employ a iterative prune → fast-finetune → evaluate cycle on a

representative subset of the training data. Crucially, each subsequent $(\tilde{m}^a, \tilde{m}^g)$ builds incrementally upon the previous one by pruning only a single **additional** layer, which enables direct weight inheritance from the previously fine-tuned model. Such continuity permits rapid accuracy recovery with minimal fine-tuning (typically 10 epochs), avoiding the computational burden of training each pruned ViT from scratch. The resulting dataset, though compact, still maintains high fidelity to the full accuracy landscape. Notably, Transfer entropy (TE) (Lin et al., 2024) is utilized as the metric to iteratively prune the model. 2) To ensure the collected data are representative, we design two sampling algorithms, including single-type progressive pruning and interleaved pruning. For algorithmic details, see Algorithm 1 and Algorithm 2) in Appendix. Besides, more details about data collection and MAP polynomial approximation,such as degree selection, runtime, are provided in the Appendix E.

**Specific redundant layer removal.** Having obtained pruning budget allocation scheme , we employ a learning-based mechanism to identify which specific layers should be pruned within each homogeneous layer group. A learnable importance parameter $\bar{m}$ is assigned to each candidate layer. In forward pass, the binary mask $\hat{m}$ in Eq. 12 controls either an activation function layer or an attention layer preservation. In backward pass, since $\hat{m}$ is not trainable, we propagate gradients of $\hat{m}$ to the learnable $\bar{m}$. That is:

$$\frac{\partial \ell}{\partial \bar{m}^a} \approx \frac{\partial \ell}{\partial \hat{m}^a}; \frac{\partial \ell}{\partial \bar{m}^g} \approx \frac{\partial \ell}{\partial \hat{m}^g}. \tag{6}$$

We then train the model, updating the importance parameters through backpropagation. Layers exhibiting the lowest importance scores in its layer type are progressively pruned until the cumulative pruned layer count matches the target pruning budget. By avoiding cross-type importance score comparisons, this approach **addresses the challenge from gradient disparity.**

### 4.3 STAGE 2: PRUNED MODEL OPTIMIZATION

**Fine-tuning.** After the multiple attention layers and activation function layers with the smallest $\hat{m}$ have been removed, fine-tuning is performed to restore the accuracy. In BoundaryDPT, we can optionally enable self-distillation during fine-tuning, which means conducting knowledge distillation (Hinton et al., 2015) under the guidance of the original ViT on to boost the pruned model's accuracy. Note that we only use the original unpruned ViT as the teacher model, without introducing any extra models. This setup aligns with the configuration adopted in related works that also employ single-model methodologies.

**Merging to speedup inference** After fine-tuning, the adjacent linear layers without an activation function layer in between are merged. The resulting Transformer block is thus formulated as:

$$
\begin{aligned}
f_l^{\hat{m}}(\cdot) &:= \mathcal{L}_{l,2} \circ I \circ \mathcal{L}_{l,1} \circ h\left(\mathcal{A}_l\right) \circ f_{l-1}^{\hat{m}}(\cdot) \\
&= \mathcal{L}_l \circ h\left(\mathcal{A}_{\theta_l}\right) \circ f_{l-1}^{\hat{m}}(\cdot),
\end{aligned} \tag{7}
$$

where $\mathcal{L}_l$ is the newly derived linear layer that has the same number of input channels as $\mathcal{L}_{l,1}$ and the same number of output channels as $\mathcal{L}_{l,2}$. In this manner, **the depth of a ViT is further reduced with rigorous dimension alignment among layers**. Besides, following such merging, the pruned ViT achieves significant inference speedups while its accuracy remains exactly the same as it was prior to merging.

### 4.4 BOUNDARYDPT+

Upon completion of the two-stage pipeline, we obtain a depth-pruned ViT that demonstrates enhanced computational efficiency while maintaining competitive accuracy. To further explore the potential of BoundaryDPT in extreme ViT compression scenarios, we integrate the depth-pruned model with established width pruning methodologies (Fang et al., 2024), thereby proposing BoundaryDPT+. BoundaryDPT+ represents a comprehensive pruning framework that addresses both depth and width dimensions of ViT architectures. The detailed performance is presented in Experiments.

## 5 EXPERIMENTS

We evaluate our method on three well-established benchmarks: (1) CIFAR-100 (Krizhevsky et al., 2009), containing 50,000 training and 10,000 test images across 100 categories; (2) ImageNet-

1k (Deng et al., 2009), a large-scale classification dataset with 1.28 million training and 50,000 validation samples spanning 1,000 classes; (3) ADE20K (Zhou et al., 2017), a semantic segmentation benchmark comprising 20,000 training and 2,000 validation images with 150 semantic categories. Inference profiling (MACs/throughput) is conducted on an NVIDIA H800 PCIE GPU. Additional experimental details are provided in the Appendix D.

## 5.1 MAIN RESULTS

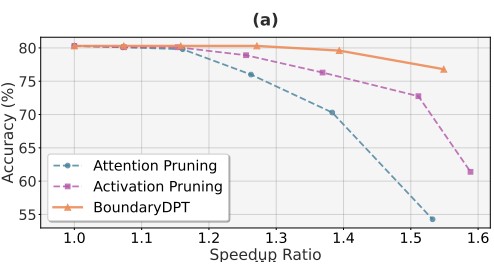 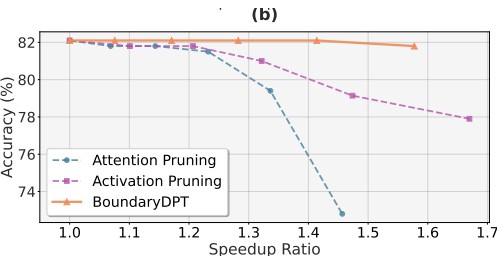

Figure 7: Comparison of BoundaryDPT and single layer type pruning on DeiT-S and DeiT-B. (a) Accuracy-speedup results on DeiT-S; (b) Accuracy-speedup results on DeiT-B.

**Accuracy-speedup tradeoff analysis of BoundaryDPT.** The isolated pruning of attention layers and activation function layers respectively using NOSE (Lin et al., 2024) is implemented for baseline comparisons. We specifically apply NOSE to activation functions because, to the best of our knowledge, no existing work has explicitly explored pruning strategies for the activation functions of ViTs. As shown in Figure 7, depth-pruned DeiT-S and DeiT-B with BoundaryDPT achieve significant higher accuracy at equivalent speedup ratios compared to isolated pruning counterparts. For DeiT-S, BoundaryDPT enables a $1.39\times$ speedup ratio with near-lossless accuracy retention (79.6% vs. baseline 79.9%), outperforming both attention-layer-only pruning (70.3%) and activation-function-only pruning (76.3%). For DeiT-B, BoundaryDPT achieves an enhanced $1.58\times$ acceleration while maintaining identical baseline accuracy (81.8% vs. baseline 81.8%), significantly outperforming the isolated pruning counterpart. These results validate our two-stage method. For an extended comparison against related methods, we provide detailed results in Table 6.

**Comparison of BoundaryDPT with depth pruning, depth-width pruning on ImageNet-1k.** As illustrated in Table 1, BoundaryDPT achieves **82.1%** Top-1 accuracy on DeiT-B with 11.8G MACs, surpassing DeiT-B's baseline accuracy (81.8%@17.6G) while reducing computation by 30.1%. This **negative efficiency-accuracy correlation** reveals our architectural pruning's ability to remove redundant computations without damaging critical information pathways. Compared to NOSE, the DeiT-B experiments demonstrate that our approach achieves lossless compression (81.8 vs. 81.8 baseline) while removing 10 layers, with superior performance in all metrics: parameter count (54.9M vs. 86.6M), MACs (10.5G vs. 17.6G), and throughput (1169.7 vs. 741.5 images/s). Notably, despite pruning substantially more layers than NOSE, BoundaryDPT still delivers stronger overall performance, underscoring the effectiveness of our method. For DeiT-S, our method maintains near-lossless accuracy when pruning 8 layers, while achieving better parameter efficiency (15.9M vs. 22.1M), compute efficiency (3.0G vs. 4.6G), and throughput (3275.0 vs. 2349.9 images/s) than competitors. These results confirm our joint pruning strategy for attention layers and activation function layers delivers superior depth pruning performance for ViTs. Our method also outperforms the baseline on Swin-Transformer while pruning Swin is more challenging since the elaborately designed hierarchical architecture has better parameter efficiency and less redundancy. A more comprehensive comparison with related methods is provided in Table 7.

**Comparison of BoundaryDPT+ with SOTA ViT compression methods on ImageNet-1k.** As illustrated in Table 2, we compare our BoundaryDPT+ against the state-of-the-art ViT extreme compression methods (predominantly width pruning approaches). Our BoundaryDPT+ achieves further compression breakthroughs when compared to the state-of-the-art Isomorphic Pruning. Specifically, compared to the Isomorphic Pruning-4.2G model, the BoundaryDPT+-3.7G model achieves higher accuracy (82.5% vs. 82.4%) while maintaining fewer parameters (20.0M vs. 20.7M), lower computational complexity (3.7G vs. 4.2G MACs), and higher throughput (2763.9 vs. 2516.0 images/s),

Table 1: Comparison with depth pruning, depth-width (DW) pruning on ImageNet-1k.

| Method | Top-1 (%) | MACs (G) | Throughput (images/s) | Params (M) | Speedup (×) |
|---|---|---|---|---|---|
| DeiT-B (Dense) | 81.8 | 17.6 | 741.5 | 86.6 | 1.00 |
| DW Pruning (0.2-12) | 81.9 | 13.6 | 907.6 | 68.2 | 1.22 |
| DW Pruning (0.2-10) | 80.9 | 10.8 | 1037.9 | 59.4 | 1.40 |
| NOSE (5 layers) | 81.8 | 14.6 | 894.7 | 75.5 | 1.21 |
| NOSE (6 layers) | 81.5 | 14.1 | 921.4 | 73.2 | 1.24 |
| **BoundaryDPT (8 layers)** | **82.1** | 11.8 | 1048.5 | 61.4 | 1.41 |
| **BoundaryDPT (10 layers)** | 81.8 | **10.5** | **1169.7** | **54.9** | **1.58** |
| DeiT-S (Dense) | 79.8 | 4.6 | 2349.9 | 22.1 | 1.00 |
| DW Pruning (0.3-12) | 78.6 | 3.1 | 2989.8 | **15.0** | 1.27 |
| NOSE (5 layers) | 79.6 | 3.7 | 2875.3 | 19.5 | 1.22 |
| **BoundaryDPT (6 layers)** | **80.1** | 3.3 | 2987.3 | 17.6 | 1.27 |
| **BoundaryDPT (8 layers)** | 79.7 | **3.0** | **3275.0** | 15.9 | **1.39** |
| SwinTransformer-S (Dense) | 83.2 | 8.7 | 1106.93 | 49.6 | 1.00 |
| DW Pruning (0.2-24) | 82.2 | 6.8 | 1140.14 | 39.2 | 1.10 |
| DW Pruning (0.2-22) | 81.8 | 6.3 | 1272.97 | 34.4 | 1.15 |
| **BoundaryDPT (10 layers)** | **82.4** | 7.3 | **1365.89** | 43.9 | **1.23** |
| SwinTransformer-T (Dense) | 81.2 | 4.5 | 1913.24 | 28.3 | 1.00 |
| DW Pruning (0.2-12) | 79.8 | 3.6 | 2093.08 | 22.7 | 1.09 |
| DW Pruning (0.2-10) | 79.3 | 3.3 | 2200.89 | 20.2 | 1.15 |
| **BoundaryDPT (5 layers)** | **80.2** | 3.7 | **2324.37** | 23.4 | **1.21** |

Table 2: Comparison with SOTA ViT compression methods on ImageNet-1k.

| Method | Top-1 (%) | MACs (G) | Throughput (images/s) | Params (M) | Speedup (×) |
|---|---|---|---|---|---|
| DeiT-B-KD | 83.3 | 17.7 | 741.5 | 87.3 | 1.00 |
| NViT-S | 82.2 | 3.9 | 2394.5 | 20.8 | 3.23 |
| Isomorphic-Pruning4.2G | 82.4 | 4.2 | 2516.0 | 20.7 | 3.39 |
| BoundaryDPT+-3.7G | 82.5 | 3.7 | 2763.9 | 20.0 | **3.73** |
| Isomorphic-Pruning2.6G | 81.1 | 2.6 | 3145.8 | 13.1 | 4.24 |
| BoundaryDPT+-2.4G | 81.4 | 2.4 | 3845.6 | 13.2 | **5.19** |
| NViT-T | 76.2 | 1.2 | 6013.7 | 6.7 | 8.11 |
| Isomorphic-Pruning1.2G | 77.5 | 1.2 | 6387.4 | 5.7 | 8.61 |
| BoundaryDPT+-1.1G | 77.4 | 1.1 | 6814.0 | 6.2 | **9.19** |

improving the speedup from 3.39× to 3.73×. Similarly, compared to the Isomorphic Pruning-2.6G model, our BoundaryDPT+-2.4G improves accuracy by 0.3% (81.4% vs. 81.1%) while significantly reducing computational complexity (2.4G vs. 2.6G MACs), with the speedup dramatically increasing from 4.24× to 5.19×, achieving nearly 1× improvement. For **ultra-extreme** compression scenarios, our method demonstrates equally impressive performance. Compared to DeiT-T and Isomorphic Pruning-1.2G, BoundaryDPT+-1.1G achieves 77.4% accuracy using only 1.1G MACs, with a remarkable speedup of 9.19× and an extraordinary throughput of 6814.0 images/s.

In short, by fully exploiting the potential of depth pruning, our method enables existing depth-width joint pruning techniques to advance further, thereby achieving new SOTA performance in ViT extreme compression.

**Transfer learning on CIFAR.** As illustrated in Tab. 3, BoundaryDPT exhibits transfer learning performance equivalent to or superior to related works. This indicates that the architecture-induced invariance learned by BoundaryDPT on ImageNet remains robust to domain shifts, and does not suffer from catastrophic forgetting during fine-tuning.

Table 3: Transfer learning results(CIFAR-100.)

| Method | Fine-tuning | Linear probing |
|---|---|---|
| DeiT-B(Dense) | 90.5 | 80.6 |
| Evo-ViT | 90.1 | 79.1 |
| EViT | 90.0 | 80.2 |
| TPS | 90.1 | 76.5 |
| NOSE(5 layers) | 90.3 | **81.3** |
| NOSE(6 layers) | 90.2 | 80.6 |
| Ours(8 layers) | **90.4** | 81.2 |
| Ours(10 layers) | 90.2 | 80.6 |

Table 4: Results on ADE20k.

| Method | mIoU (%) | mAcc (%) | aAcc (%) |
|---|---|---|---|
| Pre-trained on ImageNet-1k | | | |
| DeiT-B | 47.0 | 57.5 | 82.6 |
| EViT | 45.5 | 55.9 | 81.9 |
| TPS | 45.3 | 55.1 | 81.9 |
| NOSE(5 layers) | 46.2 | 56.5 | 82.2 |
| NOSE(6 layers) | 45.6 | 55.2 | 82.0 |
| Ours (8 layers) | **46.4** | 56.5 | **82.2** |
| Ours (10 layers) | 45.4 | 55.7 | 81.9 |

**Result on ADE20k.** We extend the proposed framework to the dense prediction task on ADE20k. As shown in Table4, when pre-trained on ImageNet-1k, our model with 8 layers removed exhibits a minimal gap of only approximately 0.6% compared to the baseline. Also, BoundaryDPT consistently outperforms other approaches.

**Excellent orthogonality to token pruning methods** BoundaryDPT exhibits remarkable compatibility with token pruning, as shown in Figure 9. When combined with GTP-ViT ($num_{prop} = 20$), BoundaryDPT further achieves a 1.24× speedup compared to BoundaryDPT alone, with only a 0.33% accuracy drop (81.47% vs. 81.8%). Meanwhile, we observe that under equivalent accuracy drops, the speedup compared to dense model remain consistently stable, maintaining between 1.5-1.6×, and are even slightly higher compared to methods without token pruning. This demonstrates that BoundaryDPT does not introduce the performance degradation typically associated with token pruning methods.

## 5.2 ABLATION STUDY

We evaluate the individual contributions of Stage 1's components on DeiT-S (8 layers pruned) and DeiT-B (10 layers pruned). For clarity, we designate pruning budget allocation as "Stage1-Step1", and specific redundant layer removal as "Stage1-Step2". More details are provided in the Appendix F.

**Ablation results.** As shown in Table 5, removing Stage1 Step2 causes significant performance drops: DeiT-B from 81.8% to 80.5% and DeiT-S from 79.7% to 79.4%. Using only TE (directly using TE metric for layer selection) leads to severe degradation: DeiT-B to 78.9%

Table 5: Ablation analysis of Stage 1 components.

| Models | TE | Stage1-Step1 | Stage1-Step2 | Acc |
|---|---|---|---|---|
| DeiT-S | ✔ | ✔ | ✔ | **79.7** |
| | ✔ | ✔ | | 79.4 |
| | | | ✔ | 78.5 |
| | ✔ | | | 78.5 |
| DeiT-B | ✔ | ✔ | ✔ | **81.8** |
| | ✔ | ✔ | | 80.5 |
| | | | ✔ | 78.1 |
| | ✔ | | | 78.9 |

and DeiT-S to 78.5%. Using only Stage1-Step 2 (directly using trainable importance scores for layer selection) performs even worse: DeiT-B to 78.1% and DeiT-S to 78.5%.

These results confirm that all components are essential: TE and Stage1-Step 1 tackle the challenge incurred by recovery asymmetry through separate layer-type statistics. Together, Stage1-Step1 and Stage1-Step2 mitigate the challenge incurred by gradient disparity via eliminating cross-type comparisons, and account for modeling layer interdependencies.

## 6 CONCLUSION

This work proposes BoundaryDPT, a two-stage method for depth pruning of ViTs. Importantly, BoundaryDPT addresses gradient disparity and accuracy recovery asymmetry via MAP-based pruning budget allocation, as well as separately quantifying the importance of attention layers and activation function layers. Extensive experiments demonstrate that BoundaryDPT achieves state-of-the-art performance in ViT depth pruning, and its extension BoundaryDPT+ can set a new state of the art in the extremely ViT compression scenarios.

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

## A    USE OF LLM

During the preparation of this paper, we employed a large language model (LLM) solely for text-related purposes, including linguistic polishing, grammar correction, and style refinement. The LLM was not involved in any aspects of the research process, such as problem formulation, algorithm design, experimental design, data analysis, or result interpretation. All scientific contributions, methodologies, and experimental findings in this paper are the sole work of the authors.

## B    THE DETAILED PIPELINE OF OUR METHOD

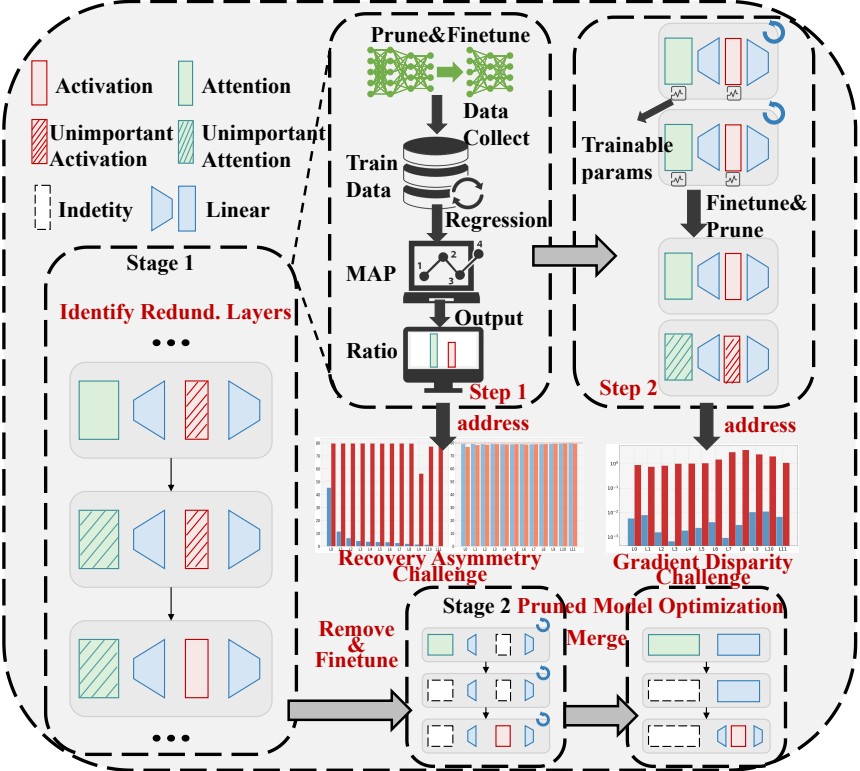

Figure 8: Overview of BoundaryDPT: a three-stage depth compression pipeline.

Our BoundaryDPT framework consists of a three-stage pipeline for efficient depth pruning of Vision Transformers. In Stage 1, the pruning process is divided into two steps: pruning ratio determination with a model accuracy predictor (MAP), and redundancy identification through training-based importance learning. These steps effectively resolve the challenges of recovery asymmetry and gradient disparity. In Stage 2, redundant attention and activation layers are removed, followed by fine-tuning, with optional self-distillation to further regain accuracy. Finally, Stage 3 merges adjacent linear layers to reduce model depth and accelerate inference while preserving accuracy. Building on this three-stage design, we further introduce BoundaryDPT+, which integrates depth pruning with established width pruning approaches for more comprehensive Transformer compression.

## C    EXPERIMENT SETUP

**Datasets and Benchmarks.** We evaluate our method on three established benchmarks: (1) CIFAR-100 (Krizhevsky et al., 2009), containing 50,000 training and 10,000 validation images across 100 categories; (2) ImageNet-1k (Deng et al., 2009), a large-scale classification dataset with 1.28 million training and 50,000 validation samples spanning 1,000 classes; (3) ADE20K (Zhou et al., 2017), a semantic segmentation benchmark comprising 20,000 training and 2,000 validation images with 150 semantic categories.

**Evaluation Protocol.** Our evaluation framework consists of three components: Primary validation on ImageNet-1k for classification accuracy, cross-task verification using ADE20K for dense prediction capability, and transfer learning analysis on CIFAR-100 to assess feature transferability. All experiments follow standardized evaluation metrics for their respective tasks.

**Training Configuration.** ImageNet-1k training employs 8 Ascend 910B2 NPUs with 128 samples per device, using AdamW optimizer ($\beta_1 = 0.9$, $\beta_2 = 0.999$) and a cosine-decayed learning rate from $5 \times 10^{-5}$ to $1 \times 10^{-5}$. CIFAR-100 experiments utilize SGD optimizer with fixed learning rate 0.1 and 384 batch size per GPU (224×224 resolution). ADE20K segmentation adopts polynomial learning rate decay (power=1.0) from initial 0.01 using SGD across 8 NPUs (2 images/device).

**Evaluation Details.** To ensure reproducibility and fair comparison, all baseline models are instantiated using the official implementation scripts and pre-trained weights provided by their respective authors. Computational complexity analysis, including FLOPs and parameter count estimation, is conducted using the standardized measurement APIs from the Torch Pruning library(Fang et al., 2024). Inference throughput evaluation follows a systematic protocol: each model undergoes 10 warm-up iterations to stabilize GPU memory allocation and kernel optimization, followed by 100 consecutive inference runs. Throughput is computed as the average inference time over the latter 100 iterations to mitigate the impact of initialization overhead and ensure statistical reliability. All experiments maintain uniform input specifications with RGB images of resolution 224×224 and a consistent batch size of 256 across all evaluated architectures.

**Hardware Platform and Training Time** All training phases are performed on Huawei Ascend 910B2 NPUs, while inference profiling (MACs/throughput) is conducted on an NVIDIA H800 Tensor Core GPU (PCIe) unless otherwise specified. Overall, the pipeline requires approximately 12 hours for pruning–ratio determination, 2 hours for layer identification on a single NPU, and 34 hours for final fine-tuning on 8 NPUs, amounting to roughly 48 hours of total computation.

# D   MORE EXPERIMENTS

## D.1   MAIN RESULTS

**Accuracy-Speedup Tradeoff Analysis.** We conducted a comprehensive evaluation of Boundary-DPT's joint pruning efficacy using DeiT-S and DeiT-B architectures. To establish baseline comparisons, we implemented isolated pruning of self-attention layers and activation layers respectively using the NOSE algorithm, followed by 400-epoch retraining for accuracy recovery (consistent with the NOSE setting). Subsequent application of BoundaryDPT's unified pruning framework under identical experimental conditions revealed statistically significant improvements: as shown in Figure 7, BoundaryDPT achieves higher model accuracy at equivalent speedup ratios compared to single-layer-type pruning baselines.

For DeiT-S, BoundaryDPT enables a 1.39× speedup ratio with near-lossless accuracy retention (79.6% vs. baseline 79.9%), outperforming both activation-layer-only pruning (76.3%) and self-attention-only pruning (70.3%). For DeiT-B, BoundaryDPT achieves an enhanced 1.58× acceleration while maintaining identical baseline accuracy (81.8% vs. baseline 81.8%), significantly surpassing activation-layer pruning (77.9%) and self-attention pruning (72.8%).

These empirical results substantiate key advantage of our joint optimization approach: through synergistic layer interaction, our method achieves superior speedup ratios compared to individual layer-type pruning strategies, demonstrating enhanced capability in eliminating architectural redundancies within Vision Transformers.

**Comparison BoundaryDPT with depth pruning, depth-width pruning, and token pruning methods on ImageNet-1k.** To demonstrate the effectiveness of our BoundaryDPT method, we conduct extensive comparisons with existing pruning strategies, including depth pruning, depth-width pruning, and token pruning methods. Since works focusing solely on depth pruning are limited, token pruning baselines are also incorporated for a more comprehensive evaluation. As shown in Table 6, BoundaryDPT consistently outperforms existing methods across multiple backbones.

For DeiT-B, BoundaryDPT achieves 82.1% Top-1 accuracy with 11.8G MACs, surpassing the dense baseline accuracy (81.8%@17.6G) while reducing computation by 30.1%. Notably, when pruning

Table 6: Comparison with depth pruning, depth-width pruning, and token pruning methods on ImageNet-1k.

| Method | Top-1 (%) | MACs (G) | Throughput (images/s) | Params (M) | Speedup (×) |
|---|---|---|---|---|---|
| DeiT-B (Dense) | 81.8 | 17.6 | 741.5 | 86.6 | 1.00 |
| DynamicViT | 81.3 | 11.2 | 1151.5 | 89.5 | 1.55 |
| Evo-ViT | 81.3 | 11.3 | 1147.0 | 87.3 | 1.55 |
| EViT | 81.3 | 11.2 | 1138.1 | 86.6 | 1.53 |
| TPS | 81.4 | 12.9 | 993.9 | 89.5 | 1.34 |
| ToMe | 80.6 | 13.2 | 949.2 | 86.6 | 1.28 |
| GTP-ViT | 81.5 | 13.2 | 955.3 | 86.6 | 1.29 |
| WD-Pruning (0.2-12) | 81.9 | 13.6 | 907.6 | 68.2 | 1.22 |
| WD-Pruning (0.2-10) | 80.9 | 10.8 | 1037.9 | 59.4 | 1.40 |
| NOSE (5 layers) | 81.8 | 14.6 | 894.7 | 75.5 | 1.21 |
| NOSE (6 layers) | 81.5 | 14.1 | 921.4 | 73.2 | 1.24 |
| **BoundaryDPT (8 layers)** | **82.1** | 11.8 | 1048.5 | 61.4 | 1.41 |
| **BoundaryDPT (10 layers)** | 81.8 | **10.5** | **1169.7** | **54.9** | **1.58** |
| DeiT-S (Dense) | 79.8 | 4.6 | 2349.9 | 22.1 | 1.00 |
| TPS | 79.7 | 3.3 | 3051.2 | 22.8 | 1.30 |
| ToMe | 79.5 | 3.3 | 2874.5 | 22.1 | 1.22 |
| GTP-ViT | 79.5 | 3.5 | 2793.4 | 22.1 | 1.19 |
| WD-Pruning (0.3-12) | 78.6 | 3.1 | 2989.8 | **15.0** | 1.27 |
| NOSE (4 layers) | 79.8 | 3.8 | 2774.8 | 20.1 | 1.18 |
| NOSE (5 layers) | 79.6 | 3.7 | 2875.3 | 19.5 | 1.22 |
| **BoundaryDPT (6 layers)** | **80.1** | 3.3 | 2987.3 | 17.6 | 1.27 |
| **BoundaryDPT (8 layers)** | 79.7 | **3.0** | **3275.0** | 15.9 | **1.39** |
| SwinTransformer-S (Dense) | 83.2 | 8.7 | 1106.93 | 49.6 | 1.00 |
| WD-Pruning (0.2-24) | 82.2 | 6.8 | 1140.14 | 39.2 | 1.10 |
| WD-Pruning (0.2-22) | 81.8 | 6.3 | 1272.97 | 34.4 | 1.15 |
| **BoundaryDPT (10 layers)** | **82.4** | 7.3 | **1365.89** | 43.9 | **1.23** |
| SwinTransformer-T (Dense) | 81.2 | 4.5 | 1913.24 | 28.3 | 1.00 |
| WD-Pruning (0.2-12) | 79.8 | 3.6 | 2093.08 | 22.7 | 1.09 |
| WD-Pruning (0.2-10) | 79.3 | 3.3 | 2200.89 | 20.2 | 1.15 |
| **BoundaryDPT (5 layers)** | **80.2** | 3.7 | **2324.37** | 23.4 | **1.21** |

10 layers, our method maintains baseline accuracy (81.8%) but dramatically improves efficiency, with only 54.9M parameters, 10.5G MACs, and 1169.7 images/s throughput, representing a 1.58× speedup. These results reveal that BoundaryDPT is able to remove redundant computations while preserving essential semantic representations, a clear indication of effective structural pruning.

For DeiT-S, BoundaryDPT achieves nearly lossless compression, reaching 80.1% accuracy with 6 layers or maintaining 79.7% accuracy with 8 layers pruned. In the latter case, it achieves superior parameter efficiency (15.9M vs. 22.1M), reduced computational complexity (3.0G vs. 4.6G), and higher throughput (3275.0 vs. 2349.9 images/s), substantially outperforming competitive token-pruning methods.

Moreover, the results generalize well to other backbones. For Swin-T and Swin-S, BoundaryDPT delivers consistent improvements over WD-Pruning baselines, achieving up to 1.23× speedup on Swin-S. These consistent gains across both plain and other ViTs confirm the versatility and robustness of our method.

**Comparison BoundaryDPT+ with SOTA ViT compression methods and other off-the-shelf ViT models on ImageNet-1k.** We further benchmark our enhanced BoundaryDPT+ against state-of-the-art extreme compression approaches, particularly width pruning methods such as Isomorphic Pruning and NViT, while also including results from efficient architecture designs like Swin, Effi-

Table 7: Comparison with SOTA ViT compression methods and other off-the-shelf ViT models on ImageNet-1k.

| Method | Top-1 (%) | MACs (G) | Throughput (images/s) | Params (M) | Speedup (×) |
|---|---|---|---|---|---|
| DeiT-B-KD | 83.3 | 17.7 | 741.5 | 87.3 | 1.00 |
| Cait-S24 | 83.5 | 8.6 | 800.8 | 46.9 | 1.08 |
| BoundaryDPT–KD | 83.3 | 10.5 | 1169.7 | 54.9 | 1.58 |
| DeiT-S-KD | 81.2 | 4.6 | 2370.6 | 22.1 | 3.20 |
| Swin-T | 81.2 | 4.5 | 1913.24 | 28.3 | 2.58 |
| NViT-S | 82.2 | 3.9 | 2394.5 | 20.8 | 3.23 |
| TNT-S | 81.5 | 4.8 | 1168.9 | 23.8 | 1.58 |
| CrossVit-S | 81.0 | 5.6 | 1958.4 | 26.9 | 2.64 |
| Efficientformer-L3 | 82.4 | 3.9 | 2482.5 | 31.4 | 3.35 |
| PVTv2-B2 | 82.0 | 3.9 | 1848.3 | 25.4 | 2.49 |
| Isomorphic-Pruning4.2G | 82.4 | 4.2 | 2516.0 | 20.7 | 3.39 |
| BoundaryDPT+-3.7G | 82.5 | 3.7 | 2763.9 | 20.0 | 3.73 |
| Isomorphic-Pruning2.6G | 81.1 | 2.6 | 3145.8 | 13.1 | 4.24 |
| BoundaryDPT+-2.4G | 81.4 | 2.4 | 3845.6 | 13.2 | 5.19 |
| DeiT-T-KD | 74.5 | 1.3 | 6760.5 | 5.9 | 9.12 |
| NViT-T | 76.2 | 1.2 | 6013.7 | 6.7 | 8.11 |
| Isomorphic-Pruning1.2G | 77.5 | 1.2 | 6387.4 | 5.7 | 8.61 |
| BoundaryDPT+-1.1G | 77.4 | 1.1 | 6814.0 | 6.2 | 9.19 |

cientFormer, and CrossViT for broader context. As summarized in Table 7, BoundaryDPT+ establishes new performance–efficiency trade-offs across different compression levels.

At moderate compression, BoundaryDPT+-3.7G surpasses Isomorphic Pruning-4.2G in both accuracy (82.5% vs. 82.4%) and efficiency, reducing parameters (20.00M vs. 20.7M), lowering MACs (3.7G vs. 4.2G), and improving throughput (2763.9 vs. 2516.0), thereby increasing speedup to 3.77×.

At stronger compression levels, BoundaryDPT+-2.4G achieves 81.4% accuracy compared to 81.1% with Isomorphic Pruning-2.6G, while delivering reduced compute cost and a faster 5.19× speedup, nearly 1× higher than the competitor.

Finally, in ultra-extreme settings, BoundaryDPT+-1.1G delivers 77.4% accuracy using only 1.1G MACs, outperforming both DeiT-T and Isomorphic Pruning-1.2G, and reaching an impressive 9.19× speedup with throughput exceeding 6800 images/s on H800 PCIe.

These comprehensive results demonstrate that BoundaryDPT+ pushes beyond the limitations of depth-width pruning, closing the gap with and even surpassing state-of-the-art width pruning methods, thus offering a new pathway towards extreme yet effective ViT compression.

**Excellent orthogonality to token pruning methods** We further investigate the orthogonality between BoundaryDPT's depth pruning and token pruning by applying GTP-ViT((Xu et al., 2024b)) to both Dense-Deit-B and BoundaryDPT architectures. As shown in Fig.9, BoundaryDPT exhibits remarkable compatibility with token pruning: when the $num-prop$ of GTP-ViT is set to 20, BoundaryDPT achieves a 1.24x speedup while experiencing only a 0.33% decrease in accuracy (81.47 vs. 81.8). Crucially, the compounded pruning effect (depth + tokens) demonstrates strict orthogonality, as BoundaryDPT's accuracy decay slope is shallower than that of Dense, indicating preserved parameter efficiency. This may be attributed to the fact that depth pruning concentrates the information in tokens, further enhancing the advantages of token pruning.

Meanwhile, we observe that under equivalent accuracy drops, the speedup ratios remain consistently stable, maintaining between 1.5-1.6×, and are even slightly higher compared to methods without token pruning. This demonstrates that our depth pruning approach, BoundaryDPT, does not introduce the performance degradation typically associated with token pruning methods.

# E   MORE DETAILS ABOUT POLYNOMIAL APPROXIMATION OF MAP

Roadmap. This appendix details: (i) the data collection procedures used to build the regression set for the MAP; (ii) the pre-transformation that maps discrete layer counts to normalized variables;

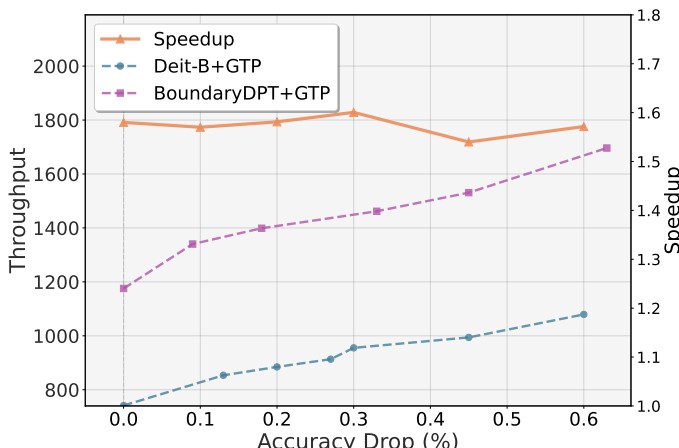

Figure 9: Orthogonality: BoundaryDPT boosts inference efficiency by combining token pruning (GTP-ViT).

(iii) the polynomial approximation of MAP with Leave-2-out cross-validation (L2OCV) for model selection; (iv) a concrete fitted result (degree, MAE/RMSE, and the polynomial); (v) the inverse transformation back to pruning ratios; (vi) runtime overhead; and (vii) a compact, colorized reference implementation.

### E.1  DATA COLLECTION

We employ two efficient data-collection procedures to obtain training samples for MAP regression: (1) single layer-type progressive pruning (Algorithm 1); and (2) interleaved pruning of attention and activation (Algorithm 2). Both procedures follow a prune–fine-tune–evaluate loop to produce representative samples at low cost.

---

**Algorithm 1:** Lightweight data collection for MAP regression ( single layer type ).

**Input:** pre-trained model $Y_0$, number of rounds $rds$, pruning budget $k$, total layers $L$

1  Initialize $train\_data = \{(\tilde{m}_0^a, \tilde{m}_0^g, a_0)\}$;
2  **Function** PruneAlong (*layer-type, $x_0$, $x_{\max}$*)**:**
3    **for** $n = 1$ **to** $rds$ **do**
        // Target pruning ratio along the specified layer type
4      $x_n = x_0 + n \cdot \frac{x_{\max} - x_0}{rds}$;
        // Prune the model
5      Prune $Y_{n-1}$ along *layer-type* to ratio $x_n$ to obtain $Y_n$;
        // Fine-tuning and evaluation
6      Fine-tune $Y_n$ and evaluate $\rightarrow (\tilde{m}_n^a, \tilde{m}_n^g, a_n)$;
7      Append $(\tilde{m}_n^a, \tilde{m}_n^g, a_n)$ to $train\_data$;
8    **end**
9  Set $\tilde{m}_{\max}^a = \frac{k}{L}$ and $\tilde{m}_{\max}^g = \frac{k}{L}$;
10  PruneAlong (*"attention"*, $\tilde{m}_0^a$, $\tilde{m}_{\max}^a$);
11  PruneAlong (*"activation"*, $\tilde{m}_0^g$, $\tilde{m}_{\max}^g$);
12  **return** $train\_data$

---

### E.2  PRE-TRANSFORMATION (FROM DISCRETE LAYER COUNTS TO NORMALIZED VARIABLES)

The original pruning configuration is specified by discrete layer counts. Let $n_a$ and $n_g$ denote the numbers of *retained* attention and activation layers, respectively, and let $L$ be the total number of

---

**Algorithm 2:** Lightweight data collection for MAP regression ( interleaved pruning ).

---

**Input:** pre-trained model $Y_0$, number of rounds $rds$, pruning budget $k$, total layers $L$

1 Initialize $train\_data = \{(\tilde{m}_0^a, \tilde{m}_0^g, a_0)\}$;

2 **Function** `PruneIterative`$((\tilde{m}_0^a, \tilde{m}_0^g), (\tilde{m}_{\max}^a, \tilde{m}_{\max}^g))$:

3    **for** $n = 1$ **to** $rds$ **do**

      // Step 1: prune attention

4       $\tilde{m}_n^a = \tilde{m}_0^a + n \cdot \frac{\tilde{m}_{\max}^a - \tilde{m}_0^g}{rds}$;

5       $\tilde{m}_{n-1}^g = \tilde{m}_0^g + (n-1) \cdot \frac{\tilde{m}_{\max}^g - \tilde{m}_0^g}{rds}$;

6       Prune $Y_{n-1}$ along *attention* to ratio $\tilde{m}_n^a$ to obtain $Y_n'$;

      // Fine-tuning and evaluation

7       Fine-tune $Y_n'$ and evaluate $\rightarrow (\tilde{m}_n^a, \tilde{m}_{n-1}^g, a_n)$;

8       Append $(\tilde{m}_n^a, \tilde{m}_{n-1}^g, a_n)$ to $train\_data$;

      // Step 2: prune activation

9       $\tilde{m}_n^g = \tilde{m}_0^g + n \cdot \frac{\tilde{m}_{\max}^g - \tilde{m}_0^g}{rds}$;

10      Prune $Y_n'$ along *activation* to ratio $\tilde{m}_n^g$ to obtain $Y_n$;

      // Fine-tuning and evaluation

11      Fine-tune $Y_n$ and evaluate $\rightarrow (\tilde{m}_n^a, \tilde{m}_n^g, a_n)$;

12      Append $(\tilde{m}_n^a, \tilde{m}_n^g, a_n)$ to $train\_data$;

13    **end**

14 Set $\tilde{m}_{a,\max} = \frac{k}{L}$ and $\tilde{m}_{g,\max} = \frac{k}{L}$;

15 `PruneIterative`$((\tilde{m}_0^a, \tilde{m}_0^g), (\tilde{m}_{\max}^a, \tilde{m}_{\max}^g))$;

16 **return** $train\_data$

---

layers. We convert these to *retention ratios*

$$a := \frac{n_a}{L}, \qquad t := \frac{n_g}{L}, \qquad a, t \in \left\{\frac{1}{L}, \frac{2}{L}, \ldots, 1\right\}.$$

If the original description uses "prune $x$ layers", then the retained layer count is $n = L - x$, hence the retained ratio is $a = (L - x)/L$ (and similarly for $t$). This transformation not only facilitates smooth performance prediction through low-degree polynomial regression in a continuous domain, but also reformulates the discrete pruning budget into an analytically tractable linear constraint ($a + t = 2 - k/L$) in the $(a, t)$ space, thereby simplifying theoretical analysis and enabling efficient optimization.

In the main text, pruning ratios are defined as $\tilde{m}^a = \sum_{l=1}^{L} \hat{m}_l^a / L$, and $\tilde{m}^g = \sum_{l=1}^{L} \hat{m}_l^g / L$, i.e., the pruned fractions. The mapping between the retained ratios $(a, t)$ and pruning ratios $(\tilde{m}_a, \tilde{m}_g)$ is

$$a = 1 - \tilde{m}^a, \qquad t = 1 - \tilde{m}^g.$$

Therefore, the budget constraint $\tilde{m}^a + \tilde{m}^g = \frac{k}{L}$ is equivalent, in the $(a, t)$ domain, to

$$a + t = 2 - \frac{k}{L}.$$

In the empirical tables (e.g., Table 8), each $(a, t)$ pair already denotes the retained ratios.

### E.3 POLYNOMIAL APPROXIMATION AND LEAVE-2-OUT CROSS-VALIDATION

We approximate MAP with a bivariate polynomial of total degree $\kappa$:

$$\mathcal{P}(a, t; \Theta) = \sum_{i+j \le \kappa} \theta_{ij} a^i t^j.$$

To reduce overfitting given limited samples, $\kappa = 2$ is typically sufficient in practice. We perform Leave-2-out cross-validation (L2OCV) over $\kappa \in \{1, 2, 3, 4\}$: for each $\kappa$, we repeatedly leave out two samples, fit least squares on the remaining data, predict the held-out samples, aggregate predictions

for all points, and compute MAE/RMSE. We then pick the $\kappa$ with the lowest validation RMSE, refit on the full dataset to obtain the final $\hat{\Theta}$, and search along the linear constraint $a + t = 2 - k/L$ over the discrete feasible set to find the recommended configuration.

With the dataset in Table 8 and the discrete constraint $a + t = 16/12$, L2OCV selects degree $\kappa = 2$ with

$$\text{MAE} = 0.4066, \quad \text{RMSE} = 0.4870.$$

The fitted polynomial MAP is

$$\hat{\mathcal{P}}(a,t) = 31.68 \;+\; 50.65\,a \;+\; 39.29\,t \;-\; 19.79\,a^2 \;-\; 8.34\,at \;-\; 11.70\,t^2. \tag{8}$$

For completeness, the unrounded coefficients (from the implementation) are: 31.684374, 50.653461, 39.298158, $-19.795489$, $-8.338992$, $-11.704586$, corresponding to the terms $1$, $a$, $t$, $a^2$, $at$, and $t^2$ respectively.

An example of DeiT-Base with $L = 12$ is shown in Table 8, where $a$ and $t$ are retained ratios $(n/L)$:

| $a$ | $t$ | Accuracy (%) | $a$ | $t$ | Accuracy (%) |
|---|---|---|---|---|---|
| 1.00 | 1.00 | 81.80 | 1.00 | 0.92 | 81.07 |
| 0.92 | 1.00 | 81.31 | 1.00 | 0.83 | 80.40 |
| 0.83 | 1.00 | 80.90 | 1.00 | 0.75 | 78.90 |
| 0.75 | 1.00 | 80.20 | 1.00 | 0.67 | 77.80 |
| 0.67 | 1.00 | 78.10 | 1.00 | 0.58 | 76.70 |
| 0.58 | 1.00 | 77.40 | 1.00 | 0.50 | 75.20 |
| 0.50 | 1.00 | 72.69 | 1.00 | 0.42 | 74.50 |
| 0.42 | 1.00 | 69.44 | 1.00 | 0.33 | 73.90 |
| 0.33 | 1.00 | 64.84 | 0.92 | 0.92 | 80.34 |
| 0.92 | 0.83 | 80.03 | 0.83 | 0.83 | 78.88 |
| 0.83 | 0.75 | 78.08 | 0.75 | 0.75 | 76.52 |

Table 8: Accuracy under different retained ratios $(a, t)$.

### E.4 INVERSE TRANSFORMATION AND REPORTING

Once the polynomial MAP is fitted as

$$\hat{\mathcal{P}}(a,t) = 31.68 \;+\; 50.65\,a \;+\; 39.29\,t \;-\; 19.79\,a^2 \;-\; 8.34\,at \;-\; 11.70\,t^2, \tag{9}$$

we can express it in terms of the pruning ratios by the inverse transformation

$$\tilde{m}^a = 1 - a, \qquad \tilde{m}^g = 1 - t.$$

This yields the equivalent polynomial

$$\mathcal{P}(\tilde{m}^a, \tilde{m}^g) = 81.79 - 2.73\,\tilde{m}^a - 7.55\,\tilde{m}^g - 19.79\,(\tilde{m}^a)^2 - 8.34\,\tilde{m}^a\tilde{m}^g - 11.70\,(\tilde{m}^g)^2. \tag{10}$$

The pruning budget requires

$$\tilde{m}^a + \tilde{m}^g = \tfrac{k}{L}.$$

Since both $\tilde{m}^a$ and $\tilde{m}^g$ take values from the finite discrete set $\{0, 1/L, \ldots, 1\}$ and must satisfy the budget constraint, the optimal configuration can be obtained simply by enumerating all feasible pairs $(\tilde{m}^a, \tilde{m}^g)$ and selecting the one with the highest value of $\mathcal{P}(\tilde{m}^a, \tilde{m}^g)$. This exhaustive search is computationally inexpensive given the small search space.

---

**Algorithm 3:** Exhaustive search for optimal pruning ratios.

**Input:** Pruning budget $k/L$, polynomial $\mathcal{P}(\tilde{m}^a, \tilde{m}^g)$
**Output:** Optimal configuration $(\tilde{m}^{a\star}, \tilde{m}^{g\star})$

1 best_score $\leftarrow -\infty$;
2 **for** $\tilde{m}^a \in \{0, 1/L, \ldots, k/L\}$ **do**
3    $\tilde{m}^g \leftarrow k/L - \tilde{m}^a$;
4    **if** $\tilde{m}^g \in \{0, 1/L, \ldots, 1\}$ **then**
5       score $\leftarrow \mathcal{P}(\tilde{m}^a, \tilde{m}^g)$;
6       **if** $score > best\_score$ **then**
7          best_score $\leftarrow$ score;
8          $(\tilde{m}^{a\star}, \tilde{m}^{g\star}) \leftarrow (\tilde{m}^a, \tilde{m}^g)$;
9       **end**
10    **end**
11 **end**
12 **return** $((\tilde{m}^a)^\star, (\tilde{m}^g)^\star)$;

---

Alternatively, one may directly optimize the polynomial in the retained-variable space $(a, t)$ along the line constraint $a + t = 2 - k/L$. The solution $(a^\star, t^\star)$ is then mapped back to pruning ratios via

$$(\tilde{m}^a)^\star = 1 - a^\star, \qquad (\tilde{m}^g)^\star = 1 - t^\star,$$

which automatically satisfies the budget. Both approaches are mathematically equivalent, as they differ only by a linear change of variables.

### E.5 RUNTIME

The MAP fitting is a small-scale least-squares regression on a handful of samples and low-degree polynomials. Its computational overhead is negligible compared to a single fine-tuning run. In practice, the MAP fitting adds virtually no runtime overhead to our pipeline.

### E.6 REFERENCE IMPLEMENTATION

We provide a compact, colorized reference implementation that performs L2OCV to select the polynomial degree, fits the final model, and searches the discrete feasible set. Inputs use retained ratios $(a, t)$; to report pruning ratios, apply $\tilde{m}^a = 1 - a$ and $\tilde{m}^g = 1 - t$.

```python
import numpy as np
from itertools import combinations

TOTAL_SUM_NUM = 16  # a + t = TOTAL_SUM_NUM / 12

data = np.array([
    (1.00, 1.00, 81.8), (0.92, 1.00, 81.31), (0.83, 1.00, 80.9),
    (0.75, 1.00, 80.2), (0.67, 1.00, 78.2), (0.58, 1.00, 77.4),
    (1.00, 0.92, 81.26), (1.00, 0.83, 80.4), (1.00, 0.75, 78.9),
    (1.00, 0.67, 77.8), (1.00, 0.58, 76.7),
    (0.92, 0.92, 80.34), (0.92, 0.83, 80.03),
    (0.83, 0.83, 78.88), (0.83, 0.75, 78.08),
    (0.75, 0.75, 76.52),
], dtype=float)

A, y = data[:, :2], data[:, 2]

def exps(k): return [(i, t-i) for t in range(k+1) for i in range(t+1)]
def Phi(A, k):
    E = exps(k)
```

```
21      return np.stack([np.prod(np.power(x, E), axis=1) for x in A], 0)
22  def fit(X, y): return np.linalg.lstsq(X, y, rcond=None)[0]
23  def pred(X, c): return X @ c
24  def l2ocv(A, y, k):
25      N = len(y); P = np.zeros(N); C = np.zeros(N, int)
26      for i, j in combinations(range(N), 2):
27          m = np.ones(N, bool); m[[i, j]] = False
28          c = fit(Phi(A[m], k), y[m])
29          for t in (i, j):
30              P[t] += pred(Phi(A[[t]], k), c)[0]; C[t] += 1
31      e = P/C - y
32      return np.mean(np.abs(e)), np.sqrt(np.mean(e**2))
33  def search_best(c, k, total=TOTAL_SUM_NUM):
34      kmin, kmax = max(1, total-11), min(11, total-1)
35      cand = []
36      for s in range(kmin, kmax+1):
37          a, t = s/12, (total-s)/12
38          acc = float(pred(Phi(np.array([[a, t]]), k), c)[0])
39          cand.append((s, a, t, acc))
40      return max(cand, key=lambda z: z[3]), cand
41
42  results = []
43  for deg in [1, 2, 3, 4]:
44      mae, rmse = l2ocv(A, y, deg)
45      C = fit(Phi(A, deg), y)
46      best, _ = search_best(C, deg)
47      results.append((deg, mae, rmse, C, best))
48  best_res = min(results, key=lambda r: r[2])
49  deg, mae, rmse, C, (s, a, t, acc) = best_res
50  print(f"Best degree: {deg}, MAE={mae:.4f}, RMSE={rmse:.4f}")
51  print(f"Recommended: a={a:.4f}, t={t:.4f}, pred acc={acc:.4f}")
52  print(f"Pruning ratios: tau_ma={1-a:.4f}, tau_mg={1-t:.4f}")
```

## F  MORE DETAILS ABOUT ABLATION STUDY

In the ablation study, we conduct experiments on the two steps of Stage 1 in BoundaryDPT. Additionally, we compare the effects of applying TE-Static and individual components for joint pruning. This section explicitly lists the pruning indices to better illustrate the challenges mentioned in our insights.

**Without TE-Static and Stage1-Step1** We initially evaluated a baseline pruning strategy that directly selects layers using trainable importance parameters without TE-Static and Stage1-Step1. This approach methodologically neglects the gradient disparity challenge between self-attention layers and activation layers, instead employing direct cross-type comparisons. As demonstrated in Tab.9, this method consistently induced systematic performance degradation across various architectures (DeiT-B and DeiT-S) and pruning intensities. Further analysis reveals its bias towards pruning more activation layers, likely due to inherent gradient magnitude disparities between layer types.

**Without Stage1-Step2 (Trainable importance scores)** We subsequently validated the critical role of Stage1-Step2 through a comparative approach that directly uses TE-Static for layer selection without trainable importance scores. This method fails to model layer interdependencies and eliminate cross-type gradient disparity challenges. As shown in Tab. 10, this method exhibits significantly degraded performance compared to BoundaryDPT, conclusively demonstrating the necessity of Stage1-Step2's trainable importance mechanism.

**TE-Static for joint pruning** Finally, we investigated a comparative method that directly employs TE-Static for joint pruning without any training components. This approach calculates the TE metric for both self-attention layers and activation layers, subsequently selecting layers with minimal TE scores as pruning candidates. As demonstrated in Table 11, the TE-Static joint pruning exhibits significant performance degradation. Notably, in contrast to the learning-based pruning observed in Table 9, this method disproportionately removes self-attention layers. This observation aligns

Table 9: Ablation on TE-Static and Stage1-Step1: BoundaryDPT (bold) vs. without TE-Static and Stage1-Step1 (non-bold)

| Models | Pruned | Self-attention Index | Activation Index | Acc |
|---|---|---|---|---|
| DeiT-S | 6 layers | [7,11] | [0,1,10,11] | 79.6 |
| | | **[1,10,11]** | **[8,10,11]** | **80.1** |
| | 8 layers | [7,8,11] | [0,1,9,10,11] | 78.5 |
| | | **[1,7,10,11]** | **[7,8,10,11]** | **79.7** |
| DeiT-B | 8 layers | [0,3] | [0,1,3,4,9,10] | 80.8 |
| | | **[0,3,7,11]** | **[2,7,8,11]** | **82.1** |
| | 10 layers | [0,1,2,5] | [0,1,2,3,9,10] | 78.1 |
| | | **[0,3,7,8,11]** | **[2,7,8,10,11]** | **81.8** |

Table 10: Ablation on Stage1-Step2: BoundaryDPT (bold) vs. without trainable importance scores (non-bold)

| Models | Pruned | Self-attention Index | Activation Index | Acc |
|---|---|---|---|---|
| DeiT-S | 6 layers | [1,9,11] | [9,10,11] | 79.9 |
| | | **[1,10,11]** | **[8,10,11]** | **80.1** |
| | 8 layers | [1,9,10,11] | [8,9,10,11] | 79.4 |
| | | **[1,7,10,11]** | **[7,8,10,11]** | **79.7** |
| DeiT-B | 8 layers | [0,3,9,11] | [7,8,9,10] | 81.1 |
| | | **[0,3,7,11]** | **[2,7,8,11]** | **82.1** |
| | 10 layers | [0,3,8,9,11] | [7,8,9,10,11] | 80.5 |
| | | **[0,3,7,8,11]** | **[2,7,8,10,11]** | **81.8** |

with the Recovery Asymmetry challenge discussed before, where self-attention layers demonstrate weaker accuracy recovery capacity during iterative pruning compared to activation layers.

# G    VISUALIZATION

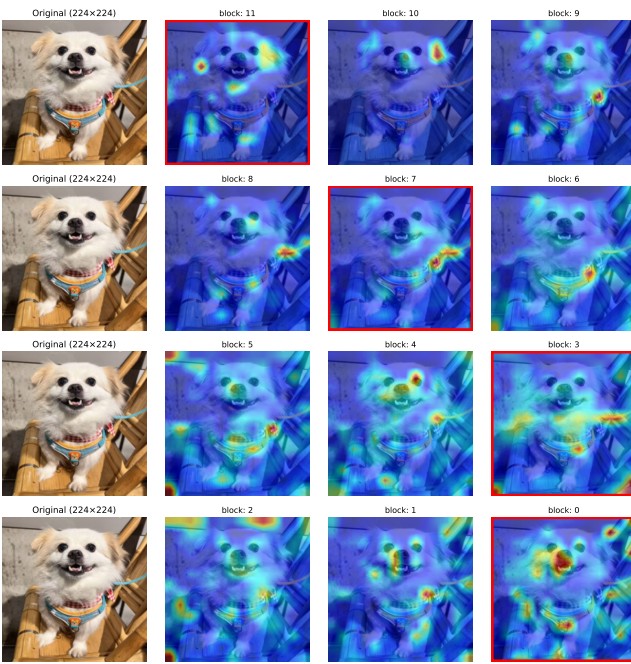

Figure 10: Visualization of cute dog: input image (left) and visualization of attention maps (right) of DeiT-Base. Attention maps with red bounding boxes are the attention layers to be removed.

We employ Attention Explanation (Chefer et al., 2021) to project the attention maps from the output token back onto the input image, thereby enabling the visualization and interpretation of different

Table 11: BoundaryDPT (bold) vs. TE-Static for joint pruning (non-bold)

| Models | Pruned | Self-attention Index | Activation Index | Acc |
|---|---|---|---|---|
| DeiT-S | 6 layers | [1,9,10,11] | [9,10] | 79.7 |
| | | **[1,10,11]** | **[8,10,11]** | **80.1** |
| | 8 layers | [1,8,9,10,11] | [9,10,11] | 78.5 |
| | | **[1,7,10,11]** | **[7,8,10,11]** | **79.7** |
| DeiT-B | 8 layers | [5,7,9,10,11] | [6,10,11] | 79.4 |
| | | **[0,3,7,11]** | **[2,7,8,11]** | **82.1** |
| | 10 layers | [5,7,9,10,11] | [5,6,9,10,11] | 78.9 |
| | | **[0,3,7,8,11]** | **[2,7,8,10,11]** | **81.8** |

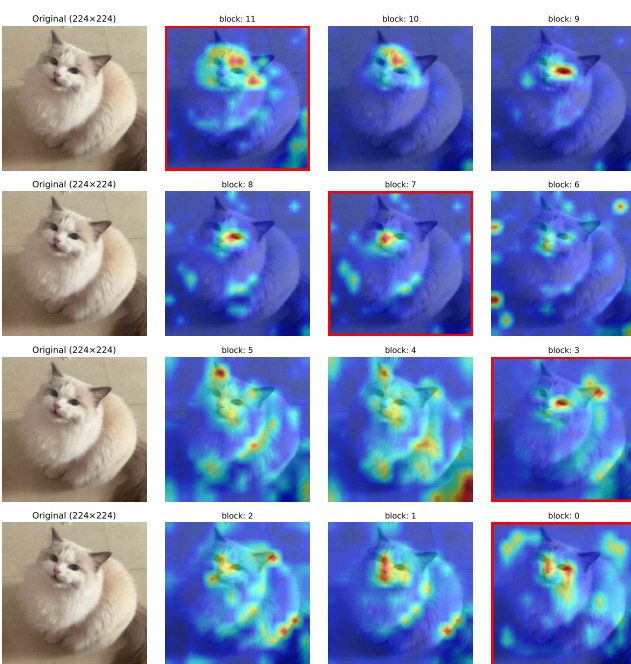

Figure 11: Visualization of cute cat: input image (left) and visualization of attention maps (right) of DeiT-Base. Attention maps with red bounding boxes are the attention layers to be removed.

blocks of multi-head self-attention within DeiT-Base. This approach provides a fine-grained perspective on how information is propagated and aggregated across layers, and allows us to explicitly observe which regions of the image are emphasized by different attention heads and blocks.

As illustrated in Figure 10 and Figure 11, our method identifies and automatically prunes attention layers that are redundant. Taking the results in Figure 10 as an example, the layers removed include Layer 0, Layer 3, Layer 7, and Layer 11. A closer examination shows that these pruned layers share highly similar attention distributions with their neighboring layers. Specifically, Layer 0 and Layer 1 both focus on the global context of the dog and its surrounding background; Layers 3, 4, and 5 collectively attend to the dog's body; Layers 6, 7, and 8 capture similar details of the clothing; and Layers 10 and 11 concentrate on the facial region. Such patterns suggest that the removed layers contribute little additional information beyond what is already captured by adjacent layers.

A similar trend can be observed in Figure 11 for the cat example, where the discarded layers also exhibit overlapping attention with those retained. This consistent redundancy across different samples underlines the robustness of our pruning strategy. Beyond its practical effect in reducing model complexity, this phenomenon also offers interpretability insights: it indicates that not all attention blocks within transformer architectures contribute equally to the final representation, and some primarily replicate context already modeled by their neighbors.

Collectively, these findings highlight that our approach not only suppresses unnecessary redundancy within the attention structure but also enhances the efficiency and interpretability of the model. By demonstrating that functionally overlapping layers can be safely pruned without sacrificing focus on

key semantic regions, our method indirectly validates the hypothesis that a more compact layer set can maintain — and potentially improve — the effectiveness of transformer-based vision models.

# H  THEORETICAL FOUNDATIONS OF MAP

This appendix provides a strengthened and fully rigorous theoretical foundation for **Model Accuracy Predictor (MAP)**. Relative to the main text, we supply additional lemmas on the discreteness, continuity, polynomial approximability, and consistency under noise. The results consolidate and mathematically justify Theorem 1–3.

Throughout, let $L$ denote the number of Transformer blocks.

## H.1  PRELIMINARIES

Assume a ViT $\mathcal{F}$ has $L$ Transformer blocks. The $l$-th transformer block $f_l$ typically integrates: 1) two linear layers, i.e., $\mathcal{L}_{i,1}$ and $\mathcal{L}_{l,2}$, 2) an attention layers $\mathcal{A}_l$, 3) a GELU activation function layer $\mathcal{G}_l$. For simplicity, we omit auxiliary components such as residual connections and layer normalization. Thus,

$$f_l(\cdot) := \mathcal{L}_{l,2} \circ \mathcal{G}_l \circ \mathcal{L}_{l,1} \circ \mathcal{A}_l \circ f_{l-1}(\cdot), \tag{11}$$

where $\circ$ denotes the function composition operation. We aim to reduce computational overhead through joint pruning of attention layers and redundant GELU activation function layers. That is,

$$
\begin{aligned}
f_l^{\hat{m}}(\cdot) &:= \mathcal{L}_{l,2} \circ h\left(\mathcal{G}_l\right) \circ \mathcal{L}_{l,1} \circ h\left(\mathcal{A}_l\right) \circ f_{l-1}^{\hat{m}}(\cdot), \\
h\left(\mathcal{G}_l\right) &= \hat{m}_l^g \circ \mathcal{G}_l + (1 - \hat{m}_l^g) \circ I, \\
h\left(\mathcal{A}_l\right) &= \hat{m}_l^a \circ \mathcal{A}_l + (1 - \hat{m}_l^a) \circ I, \\
\hat{m}_l^a, \hat{m}_l^g &\in \{0, 1\}. \\
\boldsymbol{M} &:= \left(\{\hat{m}_l^a\}_{l=1}^L, \{\hat{m}_l^g\}_{l=1}^L\right)
\end{aligned}
\tag{12}
$$

Notably, $\hat{m}_l^g$ and $\hat{m}_l^a$ are the binary masks. A mask's value of 1 preserves the corresponding layer, while 0 prunes it. $I$ denotes the identity mapping, i.e., a direct skip connection when the layer is pruned. $\boldsymbol{M}$ is the set of all the binary masks. A ViT with such masks is referred to as $\mathcal{F}^{\hat{m}}$. Thus, the joint pruning problem is formulated as:

$$
\begin{aligned}
&\underset{\boldsymbol{W}, \boldsymbol{M}}{\arg\min} \, \ell\left(\mathcal{F}^{\hat{m}}(\boldsymbol{X}, \boldsymbol{W}, \boldsymbol{M}), \boldsymbol{Y}\right), \\
&\text{s.t.} \quad \tilde{m}^a + \tilde{m}^g = k/L, \\
&\tilde{m}^a = \frac{1}{L}\sum_{l=1}^L \hat{m}_l^a, \tilde{m}^g = \frac{1}{L}\sum_{l=1}^L \hat{m}_l^g,
\end{aligned}
\tag{13}
$$

where $(\boldsymbol{X}, \boldsymbol{Y})$ are respectively samples and labels of fitting data, and $\boldsymbol{W}$ is the weights of $\mathcal{F}^{\hat{m}}$. $\ell(\cdot)$ refers to the loss function to measure data fitting quality, e.g., cross-entropy loss for classification tasks. Thus the admissible ratio set $\mathcal{D}$ is

$$\mathcal{D} = \{0, \tfrac{1}{L}, \dots, 1\}^2, \qquad |\mathcal{D}| = (L+1)^2.$$

For each $(\tilde{m}^a, \tilde{m}^g) \in \mathcal{D}$, the prune–finetune–evaluate pipeline yields a deterministic validation accuracy $\mathcal{P}$ :

$$\mathcal{P} : \mathcal{D} \to [0, 1].$$

## H.2 WELL-POSEDNESS OF PRUNING–RATIO OPTIMIZATION (THEOREM 1)

We collect and strengthen the results establishing that an optimal pruning configuration always exists under any pruning budget.

**Lemma H.1** (Finite representability). *$\mathcal{D}$ is a finite subset of $[0,1]^2$.*

*Proof.* Immediate from $\mathcal{D} = \{(i/L, j/L) : 0 \le i, j \le L\}$. $\square$

**Lemma H.2** (Functional well-definedness). *$\mathcal{P} : \mathcal{D} \to [0,1]$ is well-defined.*

*Proof.* Each point in $\mathcal{D}$ corresponds to binary masks $\hat{m}^a, \hat{m}^g$. The pipeline is deterministic and outputs a single accuracy value. $\square$

A pruning budget imposes

$$\tilde{m}^a + \tilde{m}^g = k/L,$$

yielding the constrained feasible set $\mathcal{D}_k$:

$$\mathcal{D}_k = \left\{ (\tilde{m}^a, \tilde{m}^g) \in \mathcal{D} : \tilde{m}^a + \tilde{m}^g = k/L \right\}.$$

**Lemma H.3** (Feasible set is finite). *$|\mathcal{D}_k| \le L + 1$.*

*Proof.* If $\tilde{m}^a = i/L$, then $\tilde{m}^g = (k-i)/L$, hence at most $L+1$ feasible pairs. $\square$

**Lemma H.4** (Attainment of maximum over a finite domain). *If set $S$ is finite and $f : S \to [0,1]$, then $\max_{x \in S} f(x)$ exists.*

*Proof.* A finite set has finitely many function values, whose maximum exists. $\square$

**Theorem H.5** (Existence of an optimal pruning configuration). *For any pruning budget $k$,*

$$((\tilde{m}^a)^\star, (\tilde{m}^g)^\star) \in \arg \max_{(\tilde{m}^a, \tilde{m}^g) \in \mathcal{D}_k} \mathcal{P}(\tilde{m}^a, \tilde{m}^g)$$

*exists.*

*Proof.* Combine Lemmas H.3 and H.4. $\square$

Thus pruning–ratio optimization is **well-posed**.

## H.3 POLYNOMIAL APPROXIMATION OF THE ACCURACY SURFACE (THEOREM 2)

## RELAXATION ASSUMPTION

**Assumption 1** (Continuous Relaxation of Pruning Ratios)**.** *Although structured pruning masks* $(\hat{m}^a, \hat{m}^g)$ *are discrete, their normalized pruning ratios*

$$(\tilde{m}^a, \tilde{m}^g) \in \mathcal{D} = \{0, \tfrac{1}{L}, \dots, 1\}^2 \subset [0, 1]^2$$

*admit a continuous relaxation in which* $(\tilde{m}^a, \tilde{m}^g)$ *is treated as a point in the full square* $[0, 1]^2$. *Furthermore, the accuracy functional* $\mathcal{P}$ *evaluated on* $\mathcal{D}$ *extends to a continuous map*

$$\tilde{\mathcal{P}} : [0, 1]^2 \to [0, 1], \qquad \tilde{\mathcal{P}}|_{\mathcal{D}} = \mathcal{P}.$$

**Justification.** This assumption is standard in pruning theory and inherits two empirical/structural facts:

1. *Sparsity relaxations.* Continuous formulations of pruning (e.g., magnitude relaxation, soft masks) induce validation accuracy that varies smoothly with pruning level.

2. *Observed one-dimensional smoothness.* Accuracy under $(\tilde{m}^a, 0)$ or $(0, \tilde{m}^g)$ varies smoothly with pruning ratio, suggesting that the bivariate surface is continuous.

Assumption 1 therefore formalizes a widely accepted—and empirically validated— interpretation of pruning ratios as living on a continuous compact domain.

## LEMMAS FOR STONE–WEIERSTRASS (LATTICE VERSION)

**Lemma H.6** (Domain compactness)**.** *The domain* $[0, 1]^2$ *is a compact Hausdorff space.*

*Proof.* Each interval $[0, 1]$ is compact Hausdorff; products preserve both properties. □

**Lemma H.7** (Polynomial algebra as separating lattice)**.** *Let* $A = \mathbb{R}[x, y]$ *denote the algebra of all real bivariate polynomials. Then:*

 1. *A contains all constant functions;*

 2. *A separates points of* $[0, 1]^2$;

 3. *The uniform closure* $\overline{A}$ *is closed under* $\max$ *and* $\min$, *i.e.,*

$$\max(f, g), \ \min(f, g) \in \overline{A}, \qquad \forall f, g \in \overline{A}.$$

*Hence A is a* separating lattice.

*Proof.* (1) Constant polynomials belong to $A$.

(2) If $(x_1, y_1) \neq (x_2, y_2)$, then either $x_1 \neq x_2$ or $y_1 \neq y_2$. Thus $p(x, y) = x$ or $p(x, y) = y$ separates the points.

(3) For any $f, g \in \overline{A}$,

$$\max(f, g) = \frac{f + g}{2} + \frac{|f - g|}{2}, \qquad \min(f, g) = \frac{f + g}{2} - \frac{|f - g|}{2}.$$

Since absolute value $| \cdot |$ is continuous and can be uniformly approximated on compact sets by polynomials, $|f - g| \in \overline{A}$, completing the proof. □

**Lemma H.8** (Stone–Weierstrass (Lattice Version))**.** *Let* $K$ *be a compact Hausdorff space and* $A \subset C(K)$ *a separating lattice containing constants. Then* $\overline{A} = C(K)$ *under the uniform norm.*

MAIN RESULT

**Theorem H.9** (Uniform Polynomial Approximation of the Accuracy Surface). *Under Assumption 1, for every $\varepsilon > 0$ there exists a polynomial $Q(x, y) \in \mathbb{R}[x, y]$ such that*

$$\sup_{(x,y) \in [0,1]^2} \left| \tilde{\mathcal{P}}(x, y) - Q(x, y) \right| < \varepsilon.$$

*Consequently,*

$$\max_{(\tilde{m}^a, \tilde{m}^g) \in \mathcal{D}} \left| \mathcal{P}(\tilde{m}^a, \tilde{m}^g) - Q(\tilde{m}^a, \tilde{m}^g) \right| < \varepsilon.$$

*Proof.* By Lemma H.6, $[0, 1]^2$ is compact Hausdorff. By Lemma H.7, the polynomial algebra $A$ is a separating lattice containing constants. Thus all hypotheses of the Stone–Weierstrass theorem (lattice version), Lemma H.8, are satisfied.

Therefore $\overline{A} = C([0, 1]^2)$, meaning that polynomials are uniformly dense in the space of continuous functions on the domain. Since $\tilde{\mathcal{P}} \in C([0, 1]^2)$ by Assumption 1, there exists a polynomial $Q$ such that

$$\sup_{(x,y) \in [0,1]^2} |\tilde{\mathcal{P}}(x, y) - Q(x, y)| < \varepsilon.$$

Because $\mathcal{D} \subset [0, 1]^2$, the same inequality holds when restricted to $\mathcal{D}$. $\qquad\square$

REMARK

The theorem ensures that the empirical pruning–accuracy landscape can be approximated arbitrarily well by finite-degree bivariate polynomials. This provides the theoretical foundation for MAP and other polynomial-based surrogate models used for pruning-ratio optimization.

## H.4 CONSISTENCY OF MAP UNDER NOISY FAST FINETUNING (THEOREM 3)

We now present a full statistical justification for the Meta Accuracy Predictor (MAP) when the observed accuracies are obtained via subset-based fast finetuning. Crucially, since MAP is approximated using the **least squares method** (regression), we show that the noise introduced by fast finetuning becomes an irreducible constant in the loss function and does not affect the optimization of the predictor.

OBSERVATION MODEL

For any pruning ratio pair $x = (\tilde{m}^a, \tilde{m}^g)$, let $\mathcal{P}(x)$ denote the ground-truth validation accuracy under full finetuning. Subset-based fast finetuning produces noisy observations $y$:

$$y = \mathcal{P}(x) + b + \varepsilon, \tag{14}$$

where:

- $b$ is a constant bias (systematic error) caused by the limited number of finetuning steps.
- $\varepsilon$ is a zero-mean random noise variable with finite variance $\text{Var}(\varepsilon) = \sigma^2$, assumed to be independent of $x$.

INVARIANCE OF MAXIMIZER UNDER CONSTANT BIAS

**Lemma H.10** (Bias preserves the maximizer). *For any objective function $\mathcal{P}(x)$ and a constant $b$, the set of maximizers remains unchanged:*

$$\arg\max_x \mathcal{P}(x) = \arg\max_x (\mathcal{P}(x) + b).$$

*Proof.* Adding a constant $b$ to the objective function shifts the value of the function uniformly for all $x$ but does not alter the relative ordering. Thus, if $x^*$ maximizes $\mathcal{P}$, it also maximizes $\mathcal{P} + b$. $\quad\square$

## LEAST SQUARES DECOMPOSITION

MAP is trained to approximate the observed accuracy $y$ by minimizing the **Least Squares (Mean Squared Error)** objective. Let $f(x; \theta)$ denote the MAP predictor parameterized by $\theta$. The population risk (loss function) is:

$$\mathcal{L}(\theta) = \mathbb{E}_{x,\varepsilon} \left[ (y - f(x; \theta))^2 \right].$$

Substituting the observation model $y = \mathcal{P}(x) + b + \varepsilon$:

$$\mathcal{L}(\theta) = \mathbb{E}_{x,\varepsilon} \left[ ((\mathcal{P}(x) + b - f(x; \theta)) + \varepsilon)^2 \right].$$

Expanding the square term $(A + \varepsilon)^2 = A^2 + 2A\varepsilon + \varepsilon^2$, where $A = \mathcal{P}(x) + b - f(x; \theta)$:

$$\mathcal{L}(\theta) = \underbrace{\mathbb{E}_x \left[ (\mathcal{P}(x) + b - f(x; \theta))^2 \right]}_{\mathcal{L}_{\text{clean}}(\theta)} + \underbrace{2\mathbb{E}_{x,\varepsilon} \left[ (\dots) \cdot \varepsilon \right]}_{\text{Cross-term}} + \underbrace{\mathbb{E}_\varepsilon [\varepsilon^2]}_{\text{Variance}}.$$

Since $\varepsilon$ is zero-mean ($\mathbb{E}[\varepsilon] = 0$) and independent of $x$, the cross-term vanishes. The term $\mathbb{E}[\varepsilon^2]$ is simply the noise variance $\sigma^2$. Thus, the loss decomposes into:

$$\mathcal{L}(\theta) = \mathbb{E}_x \left[ (\mathcal{P}(x) + b - f(x; \theta))^2 \right] + \sigma^2. \tag{15}$$

## MAIN THEOREM

**Theorem H.11** (MAP Consistency via Least Squares). *Let the observed accuracy be $y = \mathcal{P}(x) + b + \varepsilon$. If the MAP predictor $f(x; \theta)$ is trained by minimizing the least squares loss, then:*

1. *The optimization gradient is unaffected by the noise variance $\sigma^2$.*

2. *The predictor converges to the biased truth $\mathcal{P}(x) + b$.*

3. *The predictor recovers the optimal pruning ratios of the ground truth $\mathcal{P}(x)$.*

*Proof.* **1. Gradient Invariance:** From Eq. equation 15, the loss function is $\mathcal{L}(\theta) = \mathcal{L}_{\text{clean}}(\theta) + \sigma^2$. When computing the gradient with respect to parameters $\theta$:

$$\nabla_\theta \mathcal{L}(\theta) = \nabla_\theta \mathcal{L}_{\text{clean}}(\theta) + \nabla_\theta(\sigma^2) = \nabla_\theta \mathcal{L}_{\text{clean}}(\theta).$$

Since $\sigma^2$ is an irreducible constant, the gradient direction is identical to the gradient obtained if we were training on the noiseless (but biased) target $\mathcal{P}(x) + b$.

**2. Convergence:** Minimizing the least squares objective is equivalent to estimating the conditional expectation of the target. Assuming sufficient model capacity and training samples ($N \to \infty$):

$$f^*(x) = \mathbb{E}[y|x] = \mathbb{E}[\mathcal{P}(x) + b + \varepsilon|x] = \mathcal{P}(x) + b.$$

**3. Maximizer Recovery:** Since $f^*(x) \approx \mathcal{P}(x) + b$, by Lemma H.10, maximizing $f^*(x)$ is equivalent to maximizing $\mathcal{P}(x)$. Thus, MAP successfully identifies the optimal pruning configuration despite the noise and bias. $\square$

## REMARK

Although fast finetuning introduces a constant bias and random noise, our derivation shows that the **least squares objective** naturally handles these perturbations. The bias is rank-preserving, and the noise variance $\sigma^2$ becomes an additive constant in the loss function. Consequently, the noise does not alter the optimization landscape (gradients), ensuring that MAP remains a statistically consistent approximation for identifying the optimal pruning ratios.

## H.5 SUMMARY: THEORETICAL UNIFICATION OF MAP

Theorems 1–3 collectively construct a complete mathematical guarantee for the effectiveness of MAP. The logical closure is established as follows:

- **Existence of Solution (Theorem 1):** First, we establish that the pruning ratio optimization problem is well-posed. Since the domain $\mathcal{D}_k$ is finite and the accuracy metric is bounded, a global optimizer $((\tilde{m}^a)^\star, (\tilde{m}^g)^\star)$ is guaranteed to exist. This justifies the search for an optimal configuration.

- **Model Validity (Theorem 2):** Second, we justify the choice of the regressor. By proving that the accuracy surface $\mathcal{P}$ can be uniformly approximated by a polynomial $Q(x, y)$, we provide the theoretical license to use a polynomial-based predictor (MAP) instead of complex black-box models. This ensures that MAP has sufficient expressive power to capture the underlying landscape.

- **Robustness to Subset Data and Fast Finetuning (Theorem 3):** Finally, we validate the training strategy. Practical constraints force us to train MAP using noisy accuracy proxies obtained via fast finetuning and subset data. Theorem 3 proves that under the least squares objective, the noise variance is absorbed into the constant loss term and the systematic bias preserves the ranking. Consequently, MAP converges to the true maximizer of the ground-truth surface despite being trained on imperfect data.

**Conclusion:** MAP is theoretically sound because it searches for a guaranteed optimum (Thm 1) using a mathematically capable approximator (Thm 2) that remains consistent even when learned from noisy, low-cost signals (Thm 3).

## I  GRADIENT DISPARITY AND RECOVERY ASYMMETRY

This section establishes a unified theoretical account of two empirical phenomena observed in DeiT pruning: (i) the large gradient magnitude discrepancy between GELU paths and attention paths, and (ii) the resulting asymmetric pruning sensitivity and recovery behavior. Our analysis synthesizes the mechanisms formalized in Theorems 4 and 5 and strengthens them with additional lemmas.

### I.1  GRADIENT DISPARITY BETWEEN ACTIVATION AND ATTENTION PATHS (THEOREM 4)

We consider scalar interpolation parameters $\hat{m}^g$ and $\hat{m}^a$ applied respectively to the GELU activation output $\mathcal{G}_l \in \mathbb{R}^{N \times 4d}$ and the attention output $\mathcal{A}_l \in \mathbb{R}^{N \times d}$ of layer $l$:

$$h(\mathcal{G}_l) = \hat{m}^g \circ \mathcal{G}_l + (1 - \hat{m}^g) \circ I, \qquad h(\mathcal{A}_l) = \hat{m}^a \circ \mathcal{A}_l + (1 - \hat{m}^a) \circ I.$$

Let $\delta^g = \partial \mathcal{L}/\partial h(\mathcal{G}_l)$ and $\delta^a = \partial \mathcal{L}/\partial h(\mathcal{A}_l)$. The gradients of the loss $\mathcal{L}$ w.r.t. the interpolation scalars are

$$\nabla_{\hat{m}^g} = \sum_{i=1}^{N} \sum_{k=1}^{4d} \delta_{ik}^g (\mathcal{G}_{ik} - 1), \tag{16}$$

$$\nabla_{\hat{m}^a} = \sum_{i=1}^{N} \sum_{j=1}^{d} \delta_{ij}^a (\mathcal{A}_{ij} - 1). \tag{17}$$

#### I.1.1  ASSUMPTIONS

**Assumption 2** (Isotropic gradient distribution). *The entries of $\delta^g$ and $\delta^a$ are i.i.d. with zero mean and variance $\sigma_\delta^2$.*

**Assumption 3** (Activation–gradient independence). *Forward activations $(\mathcal{G}, \mathcal{A})$ are independent of the incoming gradient tensors $(\delta^g, \delta^a)$.*

#### I.1.2  SUPPORTING LEMMAS

**Lemma I.1** (Dimensionality amplification).

$$\dim(\mathcal{G}_l) = 4d, \qquad \dim(\mathcal{A}_l) = d.$$

*Thus, the GELU gradient aggregates four times more terms than the attention gradient.*

**Lemma I.2** (Activation variance disparity). *Define the second central moments*

$$M_{\mathcal{G}} = \mathbb{E}[(\mathcal{G} - 1)^2], \qquad M_{\mathcal{A}} = \mathbb{E}[(\mathcal{A} - 1)^2].$$

*In pre-trained DeiT models,*

$$M_{\mathcal{G}} \gg M_{\mathcal{A}}.$$

*Proof.* Attention outputs are regulated by LayerNorm and the Softmax operator, which constrain their values to a distribution close to $\mathcal{N}(0, 1)$; this behavior has been documented in prior empirical studies (Darcet et al., 2023). In contrast, GELU activations appear in an unnormalized FFN branch and exhibit heavy-tailed statistics with frequent large-magnitude outliers ($|x| \gg 1$), leading to significantly larger squared deviations. Such heavy-tailed activation patterns in FFNs have been consistently observed in recent analyses of Transformer models (Bondarenko et al., 2023; Sun et al., 2024b). $\square$

**Lemma I.3** (Expected gradient energy). *Under the assumptions,*

$$\mathbb{E}\|\nabla_{\hat{m}^g}\|^2 = 4Nd\sigma_\delta^2 M_{\mathcal{G}}, \qquad \mathbb{E}\|\nabla_{\hat{m}^a}\|^2 = Nd\sigma_\delta^2 M_{\mathcal{A}}.$$

*Proof.* Cross-terms vanish by zero-mean independence. The remaining terms accumulate linearly with tensor size. $\square$

### I.1.3 MAIN RESULT: GRADIENT DISPARITY

**Theorem I.4** (Gradient Disparity). *Let $\gamma = M_{\mathcal{G}}/M_{\mathcal{A}}$. Then*

$$\frac{\mathbb{E}\|\nabla_{\hat{m}^g}\|^2}{\mathbb{E}\|\nabla_{\hat{m}^a}\|^2} = 4\gamma, \qquad with \ \gamma \gg 1.$$

*Hence, GELU-path gradients dominate attention-path gradients by 1–2 orders of magnitude.*

*Proof.* Immediate from Lemma I.3 and $\gamma \gg 1$. $\square$

### I.2 PRUNING SENSITIVITY AND RECOVERY DYNAMICS (THEOREM 5)

We now examine the consequences of this gradient disparity for post-pruning behavior. Pruning is modeled as $\hat{m} = 0$ on a fraction $\rho$ of the channels.

Let $\Delta\mathcal{L}$ denote the immediate change in loss after pruning, before any fine-tuning steps.

### I.2.1 SENSITIVITY ANALYSIS (FORWARD DAMAGE)

**Lemma I.5** (Pruning-induced activation removal). *Let $\Delta y$ denote the output perturbation. Then under a first-order Taylor expansion,*

$$\mathbb{E}[\Delta\mathcal{L}] \propto \mathbb{E}\|\Delta y\|.$$

*Proof.* Taylor expansion of $\mathcal{L}(y + \Delta y)$ and independence of $\nabla_y \mathcal{L}$ from the pruning mask at initialization give $\Delta\mathcal{L} \approx \nabla_y \mathcal{L} \cdot \Delta y$ and $\mathbb{E}|\Delta\mathcal{L}| \propto \mathbb{E}\|\Delta y\|$. $\square$

**Lemma I.6** (Energy of removed GELU activation). *Pruning GELU removes a perturbation with energy*

$$\mathbb{E}\|\Delta y_g\|^2 \approx \rho \cdot 4d \cdot \mathbb{E}[\mathcal{G}^2],$$

*where $\mathbb{E}[\mathcal{G}^2]$ is large due to heavy-tailed outliers.*

**Lemma I.7** (Energy of removed attention activation). *Pruning attention removes a perturbation*

$$\mathbb{E}\|\Delta y_a\|^2 \approx \rho \cdot d \cdot \mathbb{E}[\mathcal{A}^2],$$

*with $\mathbb{E}[\mathcal{A}^2] = O(1)$ from LN stabilization.*

**Theorem I.8** (Pruning Sensitivity Disparity)**.**

$$\mathbb{E}[\Delta\mathcal{L}_g] \gg \mathbb{E}[\Delta\mathcal{L}_a].$$

*GELU pruning severely disrupts the residual stream, whereas attention pruning introduces only small perturbations.*

*Proof.* Immediate from $\mathbb{E}\|\Delta y_g\|^2 \gg \mathbb{E}\|\Delta y_a\|^2$. $\qquad\square$

### I.2.2 RECOVERY DYNAMICS (BACKWARD REPAIR)

Let $\theta$ be any parameter updated via gradient descent $\theta_{t+1} = \theta_t - \eta\nabla_\theta\mathcal{L}_t$.

**Lemma I.9** (One-step loss decrease)**.**

$$\mathcal{L}_{t+1} - \mathcal{L}_t \approx -\eta\,\|\nabla_\theta\mathcal{L}_t\|^2.$$

*Thus, the recovery rate is proportional to the gradient energy.*

**Lemma I.10** (Gradient scale gap after pruning)**.** *After pruning,*

$$\mathbb{E}\|\nabla_{\hat{m}^g}\|^2 \approx 4\gamma \cdot \mathbb{E}\|\nabla_{\hat{m}^a}\|^2, \qquad \gamma \gg 1.$$

**Theorem I.11** (Recovery Asymmetry)**.** *Let $R_g$ and $R_a$ denote the expected post-pruning recovery rates for GELU and attention, respectively. Then*

$$R_g \propto \mathbb{E}\|\nabla_{\hat{m}^g}\|^2 \gg R_a \propto \mathbb{E}\|\nabla_{\hat{m}^a}\|^2.$$

*Pruned GELU layers, despite suffering large initial damage (Theorem I.8), recover significantly faster than pruned attention layers due to much larger gradient signals.*

*Proof.* Combine the one-step loss decrease lemma with the gradient disparity Theorem I.4. $\qquad\square$

### I.3 UNIFIED INTERPRETATION

The two theorems jointly establish the following:

- **Forward damage is activation-dominated:** FFN–GELU channels contain high-energy outliers; removing them causes catastrophic accuracy drops.
- **Backward repair is gradient-dominated:** The same outliers magnify backpropagated gradients through the interpolation pathway, yielding orders-of-magnitude larger steps during fine-tuning.
- **Consequent asymmetry:** GELU pruning produces a "large-disruption, fast-recovery" regime, whereas attention pruning yields a "small-disruption, slow-recovery" regime.

This bidirectional amplification—large forward activations and large backward gradients—provides a rigorous theoretical explanation for the empirical pruning and recovery behavior in DeiT models.

## J LIMITATIONS

Although BoundaryDPT achieves strong empirical performance on vision tasks, our current investigation is primarily focused on Vision Transformers (ViTs). Given the architectural similarities between vision and language models, our method holds potential for broader applications. However, its effectiveness on Large Language Models (LLMs) has not yet been explored in this work. We leave the extension of BoundaryDPT to the language domain and other modalities as a subject for future research.

