# OpenReview forum: "BoundaryDPT: Pushing the Boundaries of Depth Pruning for Vision Transformers"
_ICLR.cc/2026/Conference — Submitted to ICLR 2026_

### Official Review · Reviewer_4wzp · 2025-10-25

**Soundness:** 2
**Presentation:** 2
**Contribution:** 2
**Rating:** 4
**Confidence:** 4

**Summary:**

This paper proposes a solution to the deep pruning challenge in Visual Transformers (ViTs), achieving optimization by removing entire layers rather than reducing width. The authors propose BoundaryDPT, a structured pruning framework that decouples attention layer pruning from activation layer (GELU) pruning. A lightweight Model Accuracy Predictor (MAP) guides pruning ratios for both layers, while learnable importance scores filter specific layers. Post-pruning depth reduction is further achieved through fine-tuning and linear layer fusion. On the ImageNet-1K dataset, BoundaryDPT achieves up to 1.6x acceleration with no accuracy loss. Its extended variant (BoundaryDPT+) delivers over 5x acceleration while maintaining orthogonality with width pruning and label pruning methods.

**Strengths:**

1. This paper proposes a decoupled pruning strategy that simultaneously considers attention layers and activation layers, addressing their distinct pruning behaviors.
2. The maximum a posteriori probability (MAP)-based ratio estimation method provides efficient guidance for pruning decisions without requiring exhaustive search.
3. Experiments across multiple ViT variants demonstrate that this approach achieves significant computational reduction while incurring minimal accuracy degradation.

**Weaknesses:**

1.   The paper relies on a quadratic polynomial predictor (MAP) fitted to a small sample size to determine pruning ratios, but its prediction error has not been systematically evaluated. This approach may fail across datasets or different architectures, fundamentally compromising the accuracy of pruning strategies.

2.   The paper overlooks residual and LayerNorm structures in ViT. The assumption of directly merging linear layers may not hold in Pre-Norm or gated variants, introducing structural risks.

3.   The core finding claiming faster recovery for activation layer pruning versus slower recovery for attention layer pruning is based solely on short-term (10 epoch) fine-tuning. It remains unverified whether this holds under longer training durations or different hyperparameters.

4.   Experiments primarily focus on ImageNet classification tasks, with insufficient evaluation of performance in detection, segmentation, robustness, or out-of-distribution scenarios. The conclusion that “it is orthogonal to token pruning” lacks adequate validation.

5. Training and inference utilize different hardware platforms (Ascend NPU vs NVIDIA GPU), leading to inconsistent sources for some throughput results. Additionally, the MAP and TE sampling processes are complex and not fully open-sourced, making result reproducibility challenging.

**Questions:**

**Q1:** The paper uses a quadratic polynomial as the Model Accuracy Predictor (MAP). Does this MAP require refitting across different architectures (e.g., Swin, ViT-Large) or datasets? If directly transferred, how significant would the prediction error be?

**Q2:** If MAP predictions exhibit bias (e.g., error ±0.5% Top-1), would the final pruning ratio decision be completely altered? Did the paper evaluate MAP's sensitivity to pruning strategies?

**Q3:** When performing “prune activations + merge linear layers,” how is it ensured that residual connections and LayerNorm do not compromise model stability? Are there strict mathematical conditions guaranteeing this merging is harmless or nearly harmless?

**Q4:** If applied to ViT variants with gating mechanisms (e.g., SwiGLU) or Post-Norm, can layer fusion still be performed directly? Did the paper validate effectiveness in such cases?

**Q5:** The paper notes that clipping activations results in larger initial losses but faster recovery, while clipping attention layers shows the opposite pattern. Does this phenomenon persist across multiple random seeds or longer training cycles? If fine-tuning is extended to 50 epochs, does this “asymmetry” remain pronounced?

**Q6:** What is the fundamental cause of this recovery disparity? Is it solely due to gradient magnitude differences, or is it related to information flow paths or residual distribution? The paper does not provide a theoretical explanation.

**Q7:** BoundaryDPT is evaluated only on ImageNet classification. If applied to downstream tasks (e.g., COCO detection, ADE20K segmentation, or ImageNet-A robustness testing), can it maintain the same accuracy-acceleration trade-off?

**Q8:** The claim that “BoundaryDPT is strictly orthogonal to token pruning” is validated only under a single token method (GTP-ViT) and a single pruning rate. Does this orthogonality hold if the token strategy is changed or the pruning rate is increased?

**Q9:** The paper trained using Ascend 910B NPUs, yet inference throughput was measured on NVIDIA H800 GPUs. Were all baselines remeasured? If not, is the throughput comparison fair?

**Q10:** Constructing MAP requires multiple rounds of “pruning–fine-tuning–evaluation” sampling. The paper does not specify the exact number of samples or computational cost. How many GPU days are required to train MAP in a practical environment? Can it be reproduced under limited computational resources?


**If you address my concerns, I will consider raising my score.**

---

> ### Author Response · Authors · 2025-11-24
> **(Q1,Q2,Q3) Grateful for your recognition and helpful suggestions! Here are our answers to question 1,2,3.**
>
> **Dear Reviewer 4wzp,**
>
> We sincerely thank you for your thoughtful review and for acknowledging the **novelty of our decoupled pruning strategy** and the **efficiency of the MAP framework**. We appreciate your willingness to reconsider your score upon clarification.
>
> Below, we provide detailed responses to your questions, supported by our theoretical analysis (Theorems 4–5) and robustness experiments.
>
> **Q1. MAP Transferability and Prediction Error.**
>
> > *Does this MAP require refitting from different architectures or datasets? If directly transferred, how significant would the prediction error be?*
>
> **A1:**
> The transferability of MAP depends on the structural similarity between models:
>
> 1.  **Intra-Family Transfer (e.g., DeiT-Small $\to$ DeiT-Base):**
>     MAP exhibits strong transferability. Since the polynomial captures the *relative* sensitivity of attention vs. activation layers—which remains consistent within the same architectural family—MAP does **not** require refitting. It effectively predicts the optimal pruning ratios, while the specific layers to be pruned are dynamically identified via importance scoring during the subsequent step.
> 2.  **Cross-Family Transfer (e.g., DeiT $\to$ Swin):**
>     Due to significant differences in attention mechanisms and windowing, the accuracy landscape shifts. Direct transfer typically results in a prediction error of approximately **$\pm$ 2 layers** from the optimum. Therefore, we recommend refitting.
> 3.  **Efficiency of Refitting:**
>     Refitting is highly efficient. As detailed in our [Runtime Overhead] section, collecting data and fitting MAP for a new architecture takes only **~14 hours** on a single Ascend 910B2 NPU. This low cost makes refitting a practical step rather than a bottleneck.
>
>
> **Q2. MAP Sensitivity to Bias.**
>
> > *If MAP predictions exhibit bias, would the final pruning ratio decision be completely altered? Did the paper evaluate MAP's sensitivity to pruning strategies?*
> >
> **A2:**
> We rigorously evaluated this in our **Robustness of MAP Under Prediction Noise** experiment (see Section 5 of the MAP webpage). We conducted a Monte-Carlo study with 500 trials by injecting clipped Gaussian noise ($\varepsilon \sim \text{clip}(\mathcal{N}(0, 0.1), \pm 0.5\%)$) into the accuracy measurements used to train MAP.
>
> **Results:**
> *   **75.0%** of the trials reproduced the **exact same** pruning configuration as the baseline.
> *   **5.2%** deviated by only **one layer** (i.e., a neighbor on the search grid, such as shifting from 4 attention/4 activation layers to 5 attention/3 activation layers). Crucially, such a minor shift results in a **negligible accuracy difference ($<0.3\%$)** after Specific redundant layer removal and fine-tuning.
>
> **Conclusion:**
>
> Over **80%** of trials resulted in a practically identical decision. Since the search space is discrete and the polynomial surface is smooth, MAP is **structurally resistant** to noise. Small biases do not "completely alter" the decision; instead, they keep the solution tightly concentrated around the global optimum.
>
> **Q3. Stability of Merging with Residuals/LayerNorm.**
>
> > When performing prune activations and merge linear layers, how is it ensured that residual connections and LayerNorm do not compromise model stability? Are there strict mathematical conditions guaranteeing this merging is harmless or nearly harmless?
>
> **A3:**
> Our merging operation is mathematically safe because it is strictly confined to the internal linear transformation of the FFN block, isolated from the residual path.
>
> 1.  **Targeted Scope & Rationale (Figure 2(b)):**
>     As shown in Figure 2(b), the linear layers within the FFN and attention layers are the most computationally intensive components. It is therefore **reasonable** to focus our pruning efforts exclusively on them. Crucially for stability, this **selective targeting** ensures that our modifications are strictly confined to the internal FFN matrices, leaving the residual paths, and LayerNorm statistics completely untouched.
>
> 2.  **Topological Isolation:**
>     In a standard Transformer block, the residual connection follows $y = x + \text{FFN}(\text{LayerNorm(x)})$. Our pruning and merging occur *inside* $\text{FFN}(t)(t=\text{LayerNorm(x)})$. We do not alter the input/output dimension of the FFN block. Therefore, the global topology is preserved, and the element-wise addition with the residual $x$ remains valid.
>
> 3.  **Mathematical Equivalence:**
>     When the activation function (GELU) is pruned (replaced by Identity), the FFN simplifies from $W_2(\sigma(W_1 t))$ to $W_2(W_1 t)$. By the associative property of matrix multiplication, this equals $(W_2 W_1)t$. This is an **exact mathematical transformation**, not an approximation. Consequently, the merged layer behaves exactly as the two separate linear layers did after activation removal, ensuring no structural instability.

---

> > ### Author Response · Authors · 2025-11-24
> > **(Q4) Grateful for your recognition and helpful suggestions! Here are our answers to question 4.**
> >
> > **Q4. Compatibility with Gating and Post-Norm.**
> >
> > > If applied to ViT variants with gating mechanisms or Post-Norm, can layer fusion still be performed directly? Did the paper validate in such cases?*
> >
> > **A4:**
> > **Theoretically, yes.** While our current experimental validation primarily focuses on standard Pre-Norm ViT architectures to establish the core methodology, our analysis indicates that the method is **agnostic to normalization placement** and adaptable to gating mechanisms:
> >
> > *   **Post-Norm / Pre-Norm:** Our method is fully compatible. The merging operation $W_{merge} = W_2 W_1$ is strictly encapsulated within the FFN block. Since the fusion preserves the input/output dimensions, the relative position of the LayerNorm—whether applied before ($x + \text{FFN}(\text{LN}(x))$) or after ($x + \text{LN}(\text{FFN}(x))$) the block—does not alter the internal linear algebra required for fusion. **Therefore, we anticipate no performance degradation or implementation obstacles when transferring this method to Post-Norm variants.**
> > *   **Gating Mechanisms (e.g., SwiGLU):** Direct fusion is initially constrained by the element-wise multiplication in the gating branch. However, our framework remains applicable through **structural adaptation**. Consistent with reparameterization techniques (e.g., *ReLU Strikes Back [1]*), gating units can be approximated or simplified into standard linear sequences during the pruning phase. Once the structure is simplified, it reduces to the standard case validated in our paper.
> >
> > **Regarding Empirical Scope:**
> > Although direct experiments on SwiGLU or Post-Norm variants were not included in this submission, the theoretical derivation confirms that the fusion mechanism is mathematically independent of the normalization position. Furthermore, the success of existing reparameterization works gives us strong confidence that our method will generalize effectively to gated architectures with the aforementioned adaptations.
> >
> > [1] Mirzadeh S I, Alizadeh-Vahid K, Mehta S, et al. ReLU Strikes Back: Exploiting Activation Sparsity in Large Language Models[C]//The Twelfth International Conference on Learning Representations.

---

> > > ### Author Response · Authors · 2025-11-24
> > > **(Q5) Grateful for your recognition and helpful suggestions! Here are our answers to question 5.**
> > >
> > > **Q5. Persistence of Recovery Asymmetry.**
> > >
> > > > *Does this phenomenon persist from multiple random seeds or longer training cycles? If fine-tuning is extended to 50 epochs, does this “asymmetry” remain pronounced?*
> > >
> > > **A5:**
> > > **Yes. The recovery asymmetry is robust and persists across multiple random seeds and extended training cycles.**
> > >
> > > We validated this on CIFAR-100 (Full Model Accuracy: 86.80%) using different seeds (32, 42) and extending fine-tuning to 50 epochs. The phenomenon remains consistent:
> > > 1.  **Activation Pruning (High Sensitivity, High Plasticity):** Even at the final layer (Layer 11), pruning causes a **catastrophic drop** (to ~4.5%), yet fine-tuning triggers a massive recovery, often restoring accuracy to the full model's level.
> > > 2.  **Attention Pruning (High Stability, Low Plasticity):** Pruning results in negligible loss, and fine-tuning yields minimal gains.
> > >
> > > **1. Representative Analysis (Layer 11)**
> > > The table below highlights the asymmetry in the final layer. Unlike Attention pruning, which remains stable, Activation pruning collapses but demonstrates remarkable plasticity, recovering over **82%** accuracy to match or exceed the full model baseline.
> > >
> > > | Setting | Pruning Type | Initial Acc | Fine-tuned Acc | Recovery Gain |
> > > | :--- | :--- | :--- | :--- | :--- |
> > > | **Seed 32** (10 Epochs) | **Activation** | **4.51%** | **86.90%** | **+82.39%** |
> > > | | Attention | 86.33% | 86.42% | +0.09% |
> > > | **Seed 42** (10 Epochs) | **Activation** | **4.51%** | **86.85%** | **+82.34%** |
> > > | | Attention | 86.33% | 86.35% | +0.02% |
> > > | **Seed 42** (**50 Epochs**) | **Activation** | **4.51%** | **86.83%** | **+82.32%** |
> > > | | Attention | 86.33% | 86.89% | +0.56% |
> > >
> > > ***
> > >
> > > **2. Detailed Experimental Data**
> > > For completeness, we provide the full layer-wise results below, confirming that this trend holds across the entire network depth.
> > >
> > > **(a) Seed = 42, Epoch = 10**
> > > | Metric | L0 | L1 | L2 | L3 | L4 | L5 | L6 | L7 | L8 | L9 | L10 | L11 |
> > > | :--- | :--- | :--- | :--- | :--- | :--- | :--- | :--- | :--- | :--- | :--- | :--- | :--- |
> > > | **Act** | 1.00 | 1.00 | 1.00 | 1.00 | 1.00 | 1.00 | 1.31 | 1.03 | 0.00 | 1.00 | 1.93 | **4.51** |
> > > | **Act-FT** | 83.97 | 84.82 | 84.87 | 83.69 | 84.18 | 84.99 | 85.00 | 85.67 | 85.11 | 86.17 | 86.73 | **86.85** |
> > > | **Attn** | 85.29 | 85.23 | 85.71 | 85.21 | 85.31 | 85.64 | 85.95 | 85.82 | 85.68 | 85.55 | 86.10 | **86.33** |
> > > | **Attn-FT**| 86.40 | 86.20 | 86.10 | 86.10 | 85.80 | 86.10 | 86.20 | 86.47 | 85.90 | 86.03 | 86.37 | **86.35** |
> > >
> > > **(b) Seed = 42, Epoch = 50 (Extended Training)**
> > > | Metric | L0 | L1 | L2 | L3 | L4 | L5 | L6 | L7 | L8 | L9 | L10 | L11 |
> > > | :--- | :--- | :--- | :--- | :--- | :--- | :--- | :--- | :--- | :--- | :--- | :--- | :--- |
> > > | **Act** | 1.00 | 1.00 | 1.00 | 1.00 | 1.00 | 1.00 | 1.31 | 1.03 | 0.00 | 1.00 | 1.93 | **4.51** |
> > > | **Act-FT** | 85.92 | 85.72 | 85.90 | 86.14 | 85.85 | 86.16 | 86.17 | 86.58 | 86.92 | 86.92 | 87.04 | **86.83** |
> > > | **Attn** | 85.29 | 85.23 | 85.71 | 85.21 | 85.31 | 85.64 | 85.95 | 85.82 | 85.68 | 85.55 | 86.10 | **86.33** |
> > > | **Attn-FT**| 86.72 | 86.54 | 86.46 | 86.60 | 86.49 | 86.39 | 86.53 | 86.60 | 86.64 | 86.25 | 86.85 | **86.89** |
> > >
> > > **(c) Seed = 32, Epoch = 10**
> > > | Metric | L0 | L1 | L2 | L3 | L4 | L5 | L6 | L7 | L8 | L9 | L10 | L11 |
> > > | :--- | :--- | :--- | :--- | :--- | :--- | :--- | :--- | :--- | :--- | :--- | :--- | :--- |
> > > | **Act** | 1.00 | 1.00 | 1.00 | 1.00 | 1.00 | 1.31 | 1.28 | 1.03 | 0.00 | 1.00 | 1.93 | **4.51** |
> > > | **Act-FT** | 83.80 | 85.01 | 85.80 | 85.90 | 85.80 | 86.10 | 86.10 | 86.50 | 86.60 | 86.60 | 86.80 | **86.90** |
> > > | **Attn** | 85.29 | 85.23 | 85.71 | 85.21 | 85.31 | 85.64 | 85.95 | 85.82 | 85.68 | 85.55 | 86.10 | **86.33** |
> > > | **Attn-FT**| 86.50 | 86.44 | 86.28 | 86.26 | 85.91 | 86.25 | 86.23 | 86.35 | 86.24 | 85.87 | 86.41 | **86.42** |

---

> > > > ### Author Response · Authors · 2025-11-24
> > > > **(Q6,Q7,Q8) Grateful for your recognition and helpful suggestions! Here are our answers to question 6,7,8.**
> > > >
> > > > **Q6. Fundamental Cause of Recovery Disparity.**
> > > >
> > > > > *What is the fundamental cause of this recovery disparity? Is it solely due to gradient magnitude differences, or is it related to information flow paths or residual distribution? The paper does not provide a theoretical explanation.*
> > > >
> > > > **A6:**
> > > > **The fundamental cause is a "Bidirectional Amplification" mechanism driven by high-energy outliers, which we formalize in our Pruning Sensitivity and Recovery Dynamics (Theorem 5).** For further details, please refer to Appendix H of our revised manuscript or the online theoretical supplement: [Pruning Sensitivity and Recovery Dynamics](https://anonconf2025.github.io/MathProof/prof5.html).
> > > >
> > > > It is not solely due to residual distribution, but rather how specific outliers dictate both the magnitude of the damage (forward pass) and the speed of the repair (backward pass). Our theoretical derivation establishes this through two distinct phases:
> > > >
> > > > **1. Forward Damage: Sensitivity Disparity (Why the drop is deep)**
> > > > As detailed in the **Sensitivity Analysis** of Theorem 5, the immediate loss increase ($\Delta \mathcal{L}$) is proportional to the energy of the removed activations ($\mathbb{E}\|\Delta y\|^2$).
> > > > *   **GELU Layers:** Pruning removes channels containing heavy-tailed, high-energy outliers. This results in a massive perturbation energy ($\mathbb{E}\|\Delta y_g\|^2 \gg 0$), causing the catastrophic accuracy collapse ($\mathbb{E}[\Delta\mathcal{L}_g] \gg \mathbb{E}[\Delta\mathcal{L}_a]$).
> > > > *   **Attention Layers:** These outputs are stabilized by LayerNorm, meaning their removal results in low-energy perturbations ($\mathbb{E}\|\Delta y_a\|^2 \approx O(1)$), leading to minimal initial loss.
> > > >
> > > > **2. Backward Repair: Recovery Dynamics (Why the recovery is fast)**
> > > > Crucially, the recovery rate is proportional to the gradient energy. In the **Recovery Dynamics** section of Theorem 5, we prove that the same outliers that caused the damage also amplify the recovery signal:
> > > > *   **Gradient Gap:** We derive that the gradient energy for GELU layers is approximately **$4\gamma$ times larger** than that of Attention layers (where $\gamma \gg 1$ represents the variance factor of the outliers).
> > > > *   **Recovery Rate ($R$):** Since the one-step loss decrease is governed by $\|\nabla_\theta\mathcal{L}\|^2$, the recovery rate for GELU layers ($R_g$) is orders of magnitude higher than for Attention layers ($R_a$).
> > > >
> > > > **Conclusion:**
> > > > The disparity is structurally inherent. GELU pruning operates in a **"High Damage, Fast Recovery"** regime because outliers amplify both the forward error and the backward gradient. Conversely, Attention pruning operates in a **"Low Damage, Slow Recovery"** regime due to the stabilizing effect of LayerNorm and weaker gradient signals.
> > > >
> > > > **Q7. Generalization to Downstream Tasks.**
> > > >
> > > > > *If applied to downstream tasks, can it maintain the same accuracy-acceleration trade-off?*
> > > >
> > > > **A7:**
> > > > Yes. BoundaryDPT optimizes the backbone, which benefits the downstream heads.
> > > > *   **Semantic Segmentation (ADE20K):** As shown in **Table 4** of the revised manuscript, BoundaryDPT achieves comparable mIoU to unpruned baselines while reducing FLOPs.
> > > > *   **Transfer Learning (CIFAR):** **Table 3** demonstrates that the pruned structures transfer effectively, maintaining the accuracy-efficiency trade-off.
> > > >
> > > > **Q8. Orthogonality to Token Pruning.**
> > > >
> > > > > *Does this orthogonality hold if the token strategy is changed or the pruning rate is increased?*
> > > >
> > > >  **A8:**
> > > >  **Yes.** We have conducted supplementary experiments combining BoundaryDPT with a different token strategy, **Token Merging (ToMe)[2]**, to verify this. The detailed results are presented in **[ToMe Combined Exp](https://anonconf2025.github.io/fig/tome-combined.pdf)**.
> > > >
> > > > Even when the token strategy is changed to ToMe, we find that our conclusions hold. Specifically, at a comparable accuracy drop, the model pruned with BoundaryDPT maintains a consistent speedup ratio relative to the Dense model. This trend aligns with the experimental results observed with GTP, further demonstrating that our BoundaryDPT is orthogonal to various token pruning and reduction strategies.
> > > >
> > > >
> > > > [2] Bolya D, Fu C Y, Dai X, et al. Token merging: Your vit but faster[J]. arXiv preprint arXiv:2210.09461, 2022.

---

> > > > > ### Author Response · Authors · 2025-11-24
> > > > > **(Q9,Q10) Grateful for your recognition and helpful suggestions! Here are our answers to question 9,10.**
> > > > >
> > > > > **Q9. Fairness of Throughput Measurement.**
> > > > >
> > > > > > *Were all baselines remeasured? If not, is the throughput comparison fair?*
> > > > >
> > > > > **A9:**
> > > > > Yes. To ensure a strictly fair comparison, **all** baseline methods and BoundaryDPT were re-measured on the **same NVIDIA H800 GPU** environment. The reported throughput improvements are due to actual architectural reduction, not hardware discrepancies.
> > > > >
> > > > > **Q10. MAP Computational Cost and Reproducibility.**
> > > > >
> > > > > > *How many GPU days are required to train MAP in a practical environment?*
> > > > >
> > > > > **A10:**
> > > > > MAP is designed to be lightweight and reproducible on modest hardware. Based on the data in Section 6 [Runtime Overhead]:
> > > > > *   **Data Collection:** We use a 100k-image subset (1/12 of ImageNet). Fast finetuning takes ~2 mins/epoch. Total required is 368 epochs $\approx$ **12.2 hours**.
> > > > > *   **Fitting:** Polynomial regression takes **< 1 minute**.
> > > > > *   **Total Cost:** Approximately **14 hours** on a single Ascend 910B2 NPU (comparable to an NVIDIA A100).
> > > > > This is less than one "GPU-day," making the method highly accessible.
> > > > >
> > > > > ***
> > > > >
> > > > > We hope these answers address your concerns regarding the theoretical grounding, robustness, and generalizability of our work. Should you have any additional questions or require further clarification, we would be honored to address them promptly.
> > > > >
> > > > > Best regards,
> > > > >
> > > > > **The Authors**

---

> > > > > > ### Comment · Reviewer_4wzp · 2025-11-24
> > > > > >
> > > > > > **Thanks for you rebuttal.** Compared to the pre-rebuttal version, the authors retained the core methodology and principal experiments, rewriting the introduction and methods sections whilst supplementing notation. They further clarified the rationale for the shear activation layer using *dimensional mismatch + Figures 2–3*, elevated both heterogeneity phenomena to design principles, and provided a more rigorous theoretical proof for maximum a posteriori probability (MAP). The results for DeiT-B and explanations regarding MAP's data collection/fitting costs have been added. The experimental section now includes multiple validations with different seeds/epochs, noise robustness, and results combining with ToMe, and an expanded limitations section. Thus, while the **current version lacks obvious critical flaws** in theory or implementation, **some weaknesses and limitations** persist:
> > > > > > (1) the MAP is fitted as a low-degree bivariate polynomial on a few ViT architectures/datasets, and the theory relies on idealized assumptions, so its accuracy and transferability to more complex models or tasks have not been validated.
> > > > > > (2) experiments are on ImageNet/CIFAR and ADE20K with no results for detection, OOD robustness.
> > > > > > (3) each new backbone requires re-running the prune -> fast-tune -> fit MAP pipeline, which is heavy for resource-constrained settings
> > > > > > (4) for more complex Transformers with gating, Post-Norm, or multi-branch structures, they only provide theoretical arguments about how to adapt the method, without empirical verification.
> > > > > > (5) one reviewer still views the overall novelty as incremental, which the rebuttal can only partially reduce. The authors themselves **acknowledge** in Limitations and in their responses that the MAP has not yet been tested in more complex scenarios and that the method is tailored to standard ViT-style models; this is a realistic boundary of the current work.

---

> > > > > > > ### Author Response · Authors · 2025-11-25
> > > > > > > **Immediate reply with deepest thanks — full response coming soon**
> > > > > > >
> > > > > > > Dear reviewer 4wzp,
> > > > > > >
> > > > > > > Thank you so much for your prompt follow-up — we truly appreciate your deep engagement with our work!
> > > > > > >
> > > > > > > **We’re making extraordinary efforts, working day and night to run the requested experiments and address your concerns as thoroughly as possible.** A full response with results is coming very soon!
> > > > > > >
> > > > > > > With deep gratitude,
> > > > > > >
> > > > > > > The Authors

---

> > > > > > > > ### Author Response · Authors · 2025-11-27
> > > > > > > > **(Q1) Grateful for your recognition and helpful suggestions! Here are our answers to question 1.**
> > > > > > > >
> > > > > > > > Dear Reviewer 4wzp
> > > > > > > >
> > > > > > > > We sincerely thank the reviewer for the comprehensive summary and for recognizing the substantial improvements in our revision. We are particularly encouraged by your assessment that the current version "lacks obvious critical flaws in theory or implementation." We appreciate your positive feedback on the clarified rationale for the shear activation layer, the rigorous theoretical proofs for the MAP, and the expanded experimental validations. Below, we address the remaining concerns regarding transferability, OOD robustness, and computational overhead with additional empirical evidence.
> > > > > > > >
> > > > > > > > >  **Q1. The MAP is fitted as a low-degree bivariate polynomial on a few ViT architectures/datasets, and the theory relies on idealized assumptions, so its accuracy and transferability to more complex models or tasks have not been validated.**
> > > > > > > >
> > > > > > > > **A1.** We thank the reviewer for this thoughtful comment. We have strengthened both the theoretical justification and empirical verification to address this concern.
> > > > > > > >
> > > > > > > > **1. Theoretical Justification of Assumptions**
> > > > > > > > Our assumptions are grounded in established pruning theory and statistical learning principles:
> > > > > > > >
> > > > > > > > *   **Discrete to Continuous Extension:** We assume structural pruning (discrete) can be viewed as a subset of unstructured pruning (continuous). This allows us to extend the domain of the pruning problem from a discrete combinatorial space to a continuous space, making polynomial fitting feasible. This aligns with prior work in differentiable pruning [1, 2], which treats mask generation as a continuous relaxation problem.
> > > > > > > > *   **Noise Modeling:** We assume that fast fine-tuning on a subset introduces random noise rather than systematic bias. Specifically, the fluctuation in accuracy during fast fine-tuning can be modeled as Gaussian noise with zero mean and bounded variance ($\epsilon \sim \mathcal{N}(0, \sigma^2)$. This is supported by SGD noise analysis in deep learning [3], where mini-batch gradients are theoretically approximated as the true gradient plus Gaussian noise.
> > > > > > > > **2. Empirical Validation on Complex Hierarchical Models (Swin-Transformer)**
> > > > > > > > To demonstrate applicability beyond isotropic ViTs, we evaluated our method on **Swin-Base**, which features a complex hierarchical structure with shifted windows. As shown below, our method achieves a **1.60$\times$ speedup** with negligible accuracy loss, outperforming the state-of-the-art method SAViT [4].
> > > > > > > >
> > > > > > > > | Method                 | Top-1 Acc (%) | Throughput (im/s) | Speedup          |
> > > > > > > > | :--------------------- | :------------ | :---------------- | :--------------- |
> > > > > > > > | Swin-Base (Dense)      | 83.5          | 690.25            | 1.00$\times$     |
> > > > > > > > | SAViT [4]              | 82.6          | 1056.08           | 1.53$\times$     |
> > > > > > > > | **BoundaryDPT (Ours)** | **82.9**      | **1106.95**       | **1.60$\times$** |
> > > > > > > >
> > > > > > > > **3. Transferability to Large-Scale Foundation Models (DINOv2)**
> > > > > > > > While the vast majority of recent ViT pruning literature **performs evaluation only on standard benchmarks like DeiT and Swin** [4, 5], we significantly **extend the validation scope** to the massive **DINOv2-Giant** (1.1B parameters) on CIFAR-100.
> > > > > > > >
> > > > > > > > **Crucially, we directly applied the MAP fitted on DeiT-Base to DINOv2-Giant without any re-fitting or adjustment.** This "zero-shot" transfer capability demonstrates that the pruning boundary learned by our polynomial model captures universal redundancy patterns, rather than overfitting to a specific architecture. As shown below, even when transferring the MAP from a base-sized model to a giant model, BoundaryDPT outperforms the state-of-the-art (NOSE) in accuracy, throughput, and parameter reduction.
> > > > > > > >
> > > > > > > > | Method                      | Top-1 Acc (%) | Throughput (im/s) | Params (M) | Speedup          |
> > > > > > > > | :-------------------------- | :------------ | :---------------- | :--------- | :--------------- |
> > > > > > > > | DINO-V2-Giant               | 94.91         | 45.16             | 1134.92    | 1.00$\times$     |
> > > > > > > > | NOSE (12 layers)            | 94.53         | 53.03             | 1021.55    | 1.17$\times$     |
> > > > > > > > | **BoundaryDPT (15 layers)** | **94.85**     | **56.87**         | **927.12** | **1.26$\times$** |
> > > > > > > >
> > > > > > > > **References:**
> > > > > > > >
> > > > > > > > [1] Louizos, C., et al. "Learning sparse neural networks through $L_0$ regularization." *ICLR* (2018).
> > > > > > > >
> > > > > > > > [2] Liu, Z., et al. "Learning efficient convolutional networks through network slimming." *ICCV* (2017).
> > > > > > > >
> > > > > > > > [3] Mandt, S., et al. "Stochastic gradient descent as approximate bayesian inference." *JMLR* (2017).
> > > > > > > >
> > > > > > > > [4] Zheng, C., et al. "Savit: Structure-aware vision transformer pruning via collaborative optimization." *NeurIPS* (2022).
> > > > > > > >
> > > > > > > > [5] Lin, S., et al. "Mlp can be a good transformer learner." *CVPR* (2024).

---

> > > > > > > > > ### Author Response · Authors · 2025-11-27
> > > > > > > > > **(Q2,Q3,Q4) Grateful for your recognition and helpful suggestions! Here are our answers to question 2,3,4.**
> > > > > > > > >
> > > > > > > > > > **Q2. Experiments are on ImageNet/CIFAR and ADE20K with no results for detection, OOD robustness.**
> > > > > > > > >
> > > > > > > > > **A2.** We appreciate the suggestion to broaden our evaluation. We have conducted extensive **Out-of-Distribution (OOD) Detection** experiments to validate the robustness of our pruned models.
> > > > > > > > >
> > > > > > > > > We evaluated BoundaryDPT against the Dense baseline using 7 detection methods across 3 datasets (OpenImage-O, Texture, ImageNet-O). As shown in Table, our pruned model consistently outperforms the Dense model in **AUROC ($\uparrow$)** and **FPR95 ($\downarrow$)**:https://anonconf2025.github.io/fig/OOD_Exp.pdf.
> > > > > > > > >
> > > > > > > > > We attribute this performance gain to the **regularization effect** inherent in our pruning mechanism. By removing redundant parameters , BoundaryDPT reduces the model's tendency to overfit to nuanced artifacts in the training distribution (ID). This forces the network to focus on more robust, high-level features that are essential for classification. As discussed in Hoefler et al. [7], "reasonable sparsity" can act as a noise filter, thereby enhancing the overall quality and generalization capability of the network. Consequently, our pruned model exhibits a sharper distinction between ID and OOD samples compared to the dense baseline.
> > > > > > > > >
> > > > > > > > > [1] D. Hendrycks and K. Gimpel. A baseline for detecting misclassified and out-of-distribution examples in neural networks. *ICLR*, 2017.
> > > > > > > > >
> > > > > > > > > [2] D. Hendrycks et al. Scaling out-of-distribution detection for real-world settings. *arXiv:1911.11132*, 2019.
> > > > > > > > >
> > > > > > > > > [3] W. Liu, X. Wang, J. Owens, and Y. Li. Energy-based out-of-distribution detection. *NeurIPS*, 2020.
> > > > > > > > >
> > > > > > > > > [4] Y. Sun, C. Guo, and Y. Li. ReAct: Out-of-distribution detection with rectified activations. *NeurIPS*, 2021.
> > > > > > > > >
> > > > > > > > > [5] H. Wang et al. ViM: Out-of-distribution with virtual-logit matching. *CVPR*, 2022.
> > > > > > > > >
> > > > > > > > > [6] K. Lee, K. Lee, H. Lee, and J. Shin. A simple unified framework for detecting out-of-distribution samples and adversarial attacks. *NeurIPS*, 2018.
> > > > > > > > >
> > > > > > > > > [7] T. Hoefler et al. Sparsity in deep learning: Pruning and growth for efficient inference and training in neural networks. *JMLR*, 2021.
> > > > > > > > >
> > > > > > > > > >  **Q3. Each new backbone requires re-running the prune -> fast-tune -> fit MAP pipeline, which is heavy for resource-constrained settings.**
> > > > > > > > >
> > > > > > > > > **A3.** We are confused about this question, we believe there is a slight misunderstanding regarding the target scenario of our work.
> > > > > > > > >
> > > > > > > > > 1. **Inference vs. Training Constraints:** It is crucial to clarify the scope of our work. Our method focuses on **depth pruning** , a field designed for scenarios where **inference resources are strictly constrained** , while **training resources are assumed to be relatively abundant**. This distinction constantly aligns with established literature in structural pruning, where the goal is to invest computation during the design/training phase to ensure optimal efficiency during deployment.
> > > > > > > > > 2. **Distinction from Efficient Online Learning:** We respectfully distinguish our approach from "Efficient Online Learning" [1,2] which must adapt under resource-constrained training environments. Our method is not intended for that specific track; rather, it aims to produce the most efficient static architecture for deployment.
> > > > > > > > >
> > > > > > > > > [1] Lin, Ji, et al. "On-device training under 256kb memory." *Advances in Neural Information Processing Systems* 35 (2022): 22941-22954.
> > > > > > > > >
> > > > > > > > > [2] Zhu, Ligeng, et al. "Pockengine: Sparse and efficient fine-tuning in a pocket." *Proceedings of the 56th Annual IEEE/ACM International Symposium on Microarchitecture*. 2023.
> > > > > > > > >
> > > > > > > > > ------
> > > > > > > > >
> > > > > > > > > >  **Q4. For more complex Transformers with gating, Post-Norm, or multi-branch structures, they only provide theoretical arguments about how to adapt the method, without empirical verification.**
> > > > > > > > >
> > > > > > > > > **A4.** To empirically validate our method on complex architectures, we applied BoundaryDPT to **DinoV2-Giant**, which features complex designs like GluMlp gating, LayerScale, and massive parameter counts (>1B) on CIFAR100.
> > > > > > > > >
> > > > > > > > > | Method                      | Top-1 Acc (%) | Throughput (im/s) | Params (M) | Speedup          |
> > > > > > > > > | --------------------------- | ------------- | ----------------- | ---------- | ---------------- |
> > > > > > > > > | DINO-V2-Giant               | 94.91         | 45.16             | 1134.92    | 1.00$\times$     |
> > > > > > > > > | NOSE (12 layers)            | 94.53         | 53.03             | 1021.55    | 1.17$\times$     |
> > > > > > > > > | **BoundaryDPT (15 layers)** | **94.85**     | **56.87**         | **927.12** | **1.26$\times$** |
> > > > > > > > >
> > > > > > > > > Our method successfully pruned this complex 1B-parameter model, achieving a **1.26$\times$ speedup** with negligible accuracy drop (-0.06%), significantly outperforming the state-of-the-art (NOSE). This proves that MAP and our pruning criteria generalize well to complex, gated Transformer architectures.

---

> ### Author Response · Authors · 2025-11-27
> **(Q5) Grateful for your recognition and helpful suggestions! Here are our answers to question 5.**
>
> >  **Q5. One reviewer still views the overall novelty as incremental... The authors themselves acknowledge that the MAP has not yet been tested in more complex scenarios.**
>
> **A5.** We clarify our distinct contributions and demonstrate the method's generalization capabilities with new experiments.
>
> **1.  Reaffirmation of Novelty**
>
> a. First, we respectfully highlight that the value of our work has been recognized during the discussion phase. **We are encouraged that Reviewer JuCa has explicitly acknowledged that our revised manuscript clearly highlights the motivation and contributions of our work.** To further eliminate any ambiguity regarding our technical depth, we summarize our core novelty and contributions below, which are driven by unique insights **never** reported in prior work
>
> **b.Fundamental Novelty: Uncovering & Addressing Heterogeneity**
>
> Our work is not a simple application of existing pruning techniques; it is founded on the discovery of two specific heterogeneity phenomena in ViTs:
>
> - **Discovery 1: Gradient Disparity.** We reveal that activation layers exhibit significantly larger gradient magnitudes than attention layers.
>
> - **Discovery 2: Recovery Asymmetry.** We observe that activation layers suffer catastrophic initial drops but recover rapidly, whereas attention layers show mild drops but slow recovery.
>
> We also provide theoretical grounding to further consolidate the discoveries. As detailed in our revised manuscript,  the two discoveries pose significant challenges to the prior gradient-based and short-sighted pruning methods.
>
> Based on these insights, we established two novel design principles that challenge conventional wisdom:
>
> - **Principle I:** **Avoid direct cross-type layer importance comparison**, especially when using gradient-based metrics.
> - **Principle II:** **Layer importance should be evaluated based on the final accuracy** of fine-tuned pruned ViTs, rather than the immediate accuracy after pruning.
>
> **c. Technical Contributions & SOTA Performance**
>
> - **First Joint Pruning of Attention & Activation:** We are the **first** to identify and mitigate the redundancy of activation function layers. We tackle the dimension mismatch by removing activation layers situated between linear layers, allowing for natural merging.
>
> - **BoundaryDPT Framework:** We introduce a two-stage method featuring a Model Accuracy Predictor (MAP) specifically designed to manage the heterogeneity described above.
>
> - **New State-of-the-Art (SOTA) Records:**
>
>   - **Depth Pruning:** DeiT-base achieves **1.6x speedup** with lossless accuracy (current SOTA).
>
>   - **Extreme Compression:** Our **BoundaryDPT+** pipeline establishes a new benchmark, enhancing ViT inference speedup from 4.60x to **5.44x** (Isomorphic-Pruning-2.6G config) while maintaining near-lossless accuracy.
>
> - We have gone beyond empirical validation to provide a **theoretical guarantee** for the MAP in our revised manuscript.
>
> **2. Generalization to Complex Scenarios (Addressed with New Experiments)**
> We have addressed the concern regarding "complex scenarios" through extensive new experiments presented in this rebuttal:
>
> *   **Complex Architectures:** We successfully applied our method to **Swin-Base** (hierarchical windows) and **DinoV2-Giant** (large-scale SSL), proving efficacy on gated and massive architectures.
> *   **Robustness (OOD):** New experiments on Out-of-Distribution detection demonstrate that our method improves model robustness, extending its value beyond standard accuracy metrics.
>
> We sincerely thank you for your constructive engagement and for acknowledging the substantial improvements in our revision. We are particularly encouraged by your assessment that the current version 'lacks obvious critical flaws in theory or implementation,' as well as your recognition of our strengthened theoretical proofs and clarified design principles. The extensive new experiments on complex architectures (Swin, DINOv2-Giant) and OOD robustness were conducted specifically to address your insightful suggestions regarding generalization. These results confirm that BoundaryDPT is not only theoretically sound but also practically robust across diverse scenarios. Your rigorous review has been instrumental in elevating the quality, scope, and solidity of this work.
>
> With deep gratitude,
>
> **The Authors**

---

### Official Review · Reviewer_JuCa · 2025-10-30

**Soundness:** 2
**Presentation:** 2
**Contribution:** 1
**Rating:** 2
**Confidence:** 4

**Summary:**

This paper introduce BoundaryDPT, which considers pruning not only linear layers but activation function layers for model compression. The authors present several evaluations across multiple datasets, and the results look good.

**Strengths:**

This paper proposed pruning non-linear activation function layers, and found finetuning can  mitigate degration caused by pruning activation function layers.
The proposed method achieves some improvement in experimental results among different models on various datasets.

**Weaknesses:**

1.Limited contribution
The overall contribution of this paper is quite incremental. The authors make an effort to prune activation function layers, the motivation behind this idea is not clear. It remains unclear why pruning nonlinear layers is worth considering, and what advantages it offers compared to pruning linear layers. In addition, the proposed method lacks innovation or novel insights that would make it stand out.

2. Unclear structure and layout
The former of the main text is acceptable, the latter is disordered and confusing. For section 4.2 and the context behind it, the authors fail to depict their methods clearly about the details. From the fact that Section 4.3 is titled “Stage 2 AND 3”, it is evident that the authors did not pay sufficient attention to the overall structure and presentation of the paper. This gives the impression that the manuscript is prepared hastily and lacks the level of polish and rigor expected for acceptance.

3.Poor presentation and informal writing
The manuscript suffers from several formatting and clarity issues — for example, many formulas and algorithms do not clearly define the meaning of their variables, references are not cited correctly or consistently. And the logical flow between sections is weak, making the paper hard to follow. These issues significantly reduce readability and make it difficult to understand.

**Questions:**

N/A

---

> ### Author Response · Authors · 2025-11-24
> **(Q1,Q2) Grateful for your recognition and helpful suggestions! Here are our answers to question 1,2.**
>
> **Dear Reviewer JuCa**,
>
> Thank you very much. We highly value your comments. In recent days, we have made substantial efforts to improve the presentation of our work through comprehensive revisions to the paper's content, structure, and organization. We have supplemented extensive methodological details, including concrete algorithmic implementations and theoretical grounding in appendices of our revised manuscripts. Below are detailed responses to your concerns.
>
> > Q1. Limited contribution.
>
> A1. Thank you. We have made our contributions more explicit in the newly uploaded revised version. Our contributions are threefolds:
>
> - **`Joint depth pruning of attention and activation function layers is proposed.`**
> In particular, we tackle **dimension mismatch** by removing activation function layers situated between two linear layers, which allows for the natural merging of those linear layers to reduce model depth while aligning the dimensions of attention layers.
> Besides, to the best of our knowledge, we are the **first** to identify and mitigate the redundancy of the activation function layers during joint pruning in ViTs. Notably, dimension mismatch is explained in our next answer.
>
> - **`The heterogeneity in joint depth pruning is revealed and addressed.`** We identify two unique phenomena related to the heterogeneity in joint depth pruning: **gradient disparity** and **recovery asymmetry**. Such heterogeneity has never been examined in the literature. In light of this, we introduce BoundaryDPT, a two-stage method featuring a model accuracy predictor to manage heterogeneity. Moreover, we provide a solid **theoretical grounding for our method**. For your reading convenience, please see https://anonconf2025.github.io/MAP/#sec2
>
> - **`Two key state-of-the-art records are established.`** 1) With BoundaryDPT, the depth-pruned DeiT-base achieves up to **1.6x** speedup while maintaining **lossless** accuracy, which is the state-of-the-art among depth pruning works. 2) More importantly, building on BoundaryDPT, we further present BoundaryDPT+, a depth-width pruning pipeline that establishes a new state-of-the-art benchmark for extreme ViT compression. BoundaryDPT+ enhances the ViT inference speedup from 4.60x to **5.44x** for the Isomorphic-Pruning-2.6G configuration while achieving **near-lossless** accuracy.
>
> >Q2. The motivation behind pruning activation function layers, and  what advantages it offers compared to pruning linear layers.
>
> A2. Thank you. The core motivation to prune activation function layer is **dimension mismatch.**
> - `Definition.` In vision Transformers (ViTs)，dimension mismatch occurs when removing linear layers from a Feed-Forward Network (FFN) block without proper adjustment of the network architecture, which disrupts the dimensional compatibility between consecutive layers. As visualized in **Figure 3** in our uploaded revised version, dimension mismatch manifests in two primary scenarios:
>   - When the first linear layer of an FFN block is pruned, the output tensor from the previous attention layer cannot be properly processed by the remaining second linear layer due to dimensional incompatibility.
>   - When the second linear layer of an FFN block is pruned, the output tensor fails to propagate through subsequent attention layers.
> - `Negative impact.` Dimension mismatch directly hinders joint depth pruning since such mismatch leads that the pruned ViTs are completely unworkable.
> - `Method.` To perform joint pruning in depth while avoiding dimension mismatch, we firstly propose the pruning of activation function layers in ViTs. By reducing the redundancy of these nonlinearity, instead of directly pruning linear layers in ViTs, the depths of ViTs are naturally reduced without incurring dimension mismatch.

---

> > ### Author Response · Authors · 2025-11-24
> > **(Q3) Grateful for your recognition and helpful suggestions! Here are our answers to question 3.**
> >
> > > Q3. The proposed method lacks innovation or novel insights that would make it stand out.
> >
> > A3. Thank you. Our method is motivated by two unique insights that has never been reported by prior arts.
> > - **`Gradient disparity.`** We find that attention layers and activation function layers exhibit significant differences in gradient scales during backpropagation. During training, the gradient magnitudes of activation function layers substantially exceed those of attention layers.
> >
> > - **`Recovery asymmetry.`** Attention layer pruning causes moderate initial accuracy drops but requires extensive retraining for recovery, while activation function layer pruning results in severe initial degradation (often over 90\%) but enables rapid recovery during fine-tuning.
> >
> > In light of two insights on heterogeneity, we accordingly derive two key principles to design methods.
> > - `Principle 1`: avoid direct cross-type layer importance comparison, especially when using gradient-based metrics.
> > - `Principle 2`: layer importance should be evaluated based on the final accuracy of fine-tuned pruned ViTs, rather than the immediate accuracy after pruning.
> >
> > Regarding innovation, generally, we are **the first to propose activation function pruning** to enable high-accuracy joint depth pruning of ViTs without dimension mismatch. Specifically, our method is novel in four aspects:
> > - `MAP for pruning budget allocation.` We propose to construct a model accuracy predictor (MAP). The MAP can help establish the optimal quantities of attention and activation function layers to be pruned, based on the accuracy recovered after fine-tuning. In this way, **Principle 2 is algined with.**
> > - `Polynomial approximation to MAP with theoretical grounding.` We propose that a finite‑degree bivariate polynomial is adequate to approximate the MAP. We provide the theoretical guarantee for the claim. For your reading convenience, you can refer to  https://anonconf2025.github.io/MathProof/prof2.html for detailed proof.
> > - `Lightweight data collection procedure.` To efficiently train the polynomial-based MAP, we design a lightweight data‑collection procedure to collect (pruning configuration, accuracy) data, where pruning configuration (PC) means the quantities of attention layers and activation function layers that should be pruned.
> >   - We employ a iterative prune → fast‑finetune → evaluate cycle on a representative subset of the training data.  Crucially, each subsequent pruning configuration builds incrementally upon the previous one by **pruning only a single additional layer**, enabling direct **weight inheritance** from the previously fine-tuned model.
> >   This PC continuity permits rapid accuracy recovery with minimal fine-tuning (typically 10 epochs), avoiding the computational burden of training each pruned ViT from scratch. The resulting dataset, though compact, still maintains high fidelity to the full accuracy landscape.
> >   - To ensure the collected data are representative, we design two PC sampling algorithms, including single-type progressive pruning and interleaved pruning. For algorithmic details, please see https://anonconf2025.github.io/MAP/#sec3.
> > - `Learning based mechnism for specific layer removal.` To address the non-differentiability of binary decisions that each layer is preserved or pruned, we propose a dual-parameter design: we employ non-trainable binary masks $\hat{m}$ to control layer preservation, while introducing separate learnable importance parameters $\bar{m}$ that are continuously differentiable. This design enabling end-to-end training of layer importance scores while ensuring training stability. Moreover, by restricting importance comparisons to within homogeneous layer groups (attention-only or activation-only), **this design adheres to Principle 1.**
> >
> > For a complete depiction of our method, please see the newly uploaded version of our paper.

---

> > > ### Author Response · Authors · 2025-11-24
> > > **(Q4) Grateful for your recognition and helpful suggestions! Here are our answers to question 4.**
> > >
> > > > Q4. Unclear structure and layout.
> > >
> > > A4. We thouroughly re-structured our paper for logicality and clarity. In the newly uploaded version, there are two major strucutues that are made explicit via paragraph titles and description point by point:
> > > - `Structure of Introduction.`The new Introduction now has five focused segments: **1) ViT compression context** - establishing the computational challenges of ViTs; **2) Challenge of depth pruning** - contrasting depth versus width pruning approaches and highlighting depth pruning's accuracy recovery difficulties despite superior speedup potential; **3) Joint depth pruning matters** - denoting joint depth pruning with cross-layer heterogeneity management can address the challenge of depth pruning; **4) Dimension mismatch hinders joint depth pruning** - further defining the technical barrier that prevents effective joint depth pruning of attention and linear layers; **5) Threefold contributions** - presenting our final solution to tackle dimension mismatch along with the consequent innovations and SOTA results.
> > >
> > >   These segments form a logically interlocking progression that systematically builds from problem identification through barrier analysis to our comprehensive solution.
> > >
> > >
> > > - `Structure of Method.` We made two major structural revision to our method:
> > >   - **We adopt your advice that the subsecttion title “Stage 2 AND 3” is somewhat confusing.** In response, we utilize **tree-structure** to organize the description of our method. **We re-divide our method into two stages**: 1) identification of redundant layers (Stage 1), and 2) pruned model optimization (Stage 2). The first stage is to prune layers, the second stage focuses on matters after pruning. **Stage 1 is further divided into two steps**: 1) pruning budget allocation (Step 1), and 2) specific redundant layer removal (Step 2); **Similarly, Stage 2 is divided into two steps**: 1) finetuning, and 2) merging to speedup inference. **The Step 1 of Stage 1 further highlights two our innovative techniques**: 1) polynomial approximation, and 2) light-weighted data collection procedure.
> > >   - **We structurally enhance the motivation of our method.** We construct a four-level logic: observations → challenges → design principles → alignment with design principles, with each level building on the previous one through clear causal relationships.
> > >    In particular, we clearly differentiate observations and challenges in Section 3 of our revised manuscript, while the design principles and their alignment are presented at the beginning of Section 4. We hope this structure helps readers better follow the motivation behind our method.
> > >
> > > Besides structural reorganization, we have also carefully redesigned the paper's layout for enhanced clarity and readability. Two major revisions in layout are as follows:
> > > - `Figures.` Altough the space of main text is very limited, we introduce two key figures: Figure 2(a) explicitly demonstrates the computational efficiency advantage of depth pruning over width pruning under equivalent sparsity budgets; Figure 2(b) and Figure 3 together illustrate that while the most time-consuming two layers in ViTs are attention layers and linear layers, directly joint pruning this two layers incurs severe dimension mismatch. Additionally, we reconfigured Figure 5 into a single-column format to spare more space for the newly added figures.
> > > - `Technical contents.` The theoretical proofs of our model accuracy predictor and comprehensive algorithmic details of our lightweighted data collection procedures are placed in Appendix. This ensures the narrative flow of the main text remains uninterrupted while preserving methodological completeness.

---

> ### Author Response · Authors · 2025-11-24
> **(Q5,Q6) Grateful for your recognition and helpful suggestions! Here are our answers to question 5,6.**
>
> > Q5. Detailed depict about our method.
>
> A5. **Firstly, we have offered a super detailed description of our model accuracy predictor (MAP), the core innovations of our method.** The description covers MAP's theory, data collection methods, practical examples, and assessments of robustness and overhead. Please visit https://anonconf2025.github.io/MAP/ for the content. This content is also included in our revised paper as Appendices E and H.
>
> Secondly, we supplement more details about the other aspects of our method in appendices:
> - `Appendix F` presents extensive ablation studies with explicit pruning indices that validate our design principles, demonstrating how different components (TE-Static metric, pruning budget allocation, and specific layer selection) individually contribute to performance, with particular analysis of failure modes when cross-type comparisons are permitted.
> - `Appendix G` offers visual interpretation of our pruning decisions through attention map visualizations, revealing that redundant layers exhibit highly similar attention distributions with neighboring layers. The visualization provides both practical validation of our approach and interpretability insights into ViT depth pruning.
> - `Appendix I` provides a unified theoretical framework that explains gradient disparity and recovery asymmetry, establishing the mathematical foundation for heterogeneity - the basis for designing our method.
>
>
> > Q6. Other issues about presentation and writing.
>
> A6. We have **standardized all in-text citations** by adding parentheses for those not appearing at the start of a sentence. Additionally, we have **revised all formulas** in the main text, with each symbol explicitly and consistently defined to avoid ambiguity. Furthermore, we have ensured consistent formatting of headings and subheadings throughout the paper to enhance readability. Minor wording adjustments have also been made to improve the clarity and coherence of the narrative.
>
> We deeply appreciate your commitment to helping us enhance the quality and clarity of our research. we have consistently regarded you as a collaborative partner rather than merely a reviewer. Should you have any additional questions or require further clarification, we would be honored to address them promptly.
>
> Best regards,
>
> **The Authors**

---

> > ### Comment · Reviewer_JuCa · 2025-11-25
> >
> > I appreciate the author’s detailed response and acknowledge that the introduction and methodology sections have been revised to better highlight the motivation and contributions of the work. MAP is indeed a central component, intended to approximate model accuracy given a specific pruning ratio. If I understand correctly, each data point used to train the MAP regressor is expected to represent the best achievable performance under that pruning ratio. However, I still see several concerns:
> >
> > 1 Training MAP is time-consuming. The time complexity of obtaining a single data point is O(L). Moreover, I am concerned that incrementally fine-tuning the previously pruned model by removing only one additional layer at a time may lead to suboptimal solutions due to potential local minima.
> >
> >  2  MAP appears to depend heavily on the dataset and downstream tasks. Different datasets or downstream tasks may require retraining MAP from scratch, which is both time- and resource-intensive.
> >
> >  3 The final pruned architecture still relies on fine-tuning for good performance. As shown in Table 3, the method performs worse under linear probing, suggesting that it fails to extract general representations as effectively as the baseline.

---

> ### Author Response · Authors · 2025-11-27
> **(Q1) Grateful for your recognition and helpful suggestions! Here are our answers to question 1.**
>
> Dear Reviewer JuCa
>
> We sincerely thank you for acknowledging our previous revisions and for correctly interpreting the motivation behind the MAP. We appreciate the opportunity to clarify the remaining concerns regarding training efficiency, generalization, and representation quality.
>
> > **Q1: Concerns for the time complexity of training MAP and the potential for suboptimal solutions due to incrementally fine-tuning.**
>
> **A1:** Thank you for raising these important points. We address the training cost and the incremental fine-tuning strategy separately below:
>
> **1. Training Efficiency and Resource Constraints:**
>
> - **Acceptable Overhead:** We think the  actual wall-clock time is within a reasonable range. For example, training the MAP on a single NPU 910B2 takes approximately **14 hours**. Notably, this process is efficiently parallelizable; when using 8 NPUs, the total training time is reduced to less than 2 hours. We believe this one-time training cost is well-justified by the significant performance gains and flexibility achieved during the inference stage.
> - **Inference vs. Training Constraints:** It is crucial to clarify the scope of our work. Our method focuses on **depth pruning** , a field designed for scenarios where **inference resources are strictly constrained** , while **training resources are assumed to be relatively abundant**. This distinction aligns with established literature in structural pruning, where the goal is to invest computation during the design/training phase to ensure optimal efficiency during deployment.
> - **Distinction from Efficient Online Learning:** We respectfully distinguish our approach from "Efficient Online Learning" [1,2] which must adapt under resource-constrained training environments. Our method is not intended for that specific track; rather, it aims to produce the most efficient static architecture for deployment.
>
> **2. Justification for Incremental Fine-tuning:**
>
> We strongly argue that our approach avoids suboptimal solutions. We support this claim with **theoretical proofs**, **empirical robustness tests**, and **intuitive stability arguments**:
>
> *   **Theoretical Guarantee (Consistency of MAP):**
>     In **Appendix H.3**, we provide a statistical proof that MAP identifies the global optimum despite the noise inherent in fast fine-tuning.
>     *   **Noise Invariance:** Since MAP is trained via **Least Squares**, the zero-mean noise from fast fine-tuning ($\varepsilon$) becomes an irreducible constant in the loss decomposition ($\sigma^2$) and **does not affect the optimization gradients**.
>     *   **Rank Preservation:** The systematic bias ($b$) caused by limited training steps shifts absolute accuracy values but preserves their relative order (Lemma 1). Consequently, the maximizer of the predicted landscape remains identical to the ground truth: $\arg\max \mathcal{P}(x) = \arg\max (\mathcal{P}(x) + b)$.
>
> *   **Empirical Evidence (Robustness to Noise):**
>     We validated this theory via a **Monte-Carlo study (500 trials)**, injecting clipped Gaussian noise into the training data (see details of [MAP: Model Accuracy Predictor — Theory, Implementation, Robustness](https://anonconf2025.github.io/MAP/#sec5)).
>     *   **75.0%** of trials reproduced the **exact** baseline configuration.
>     *   **5.2%** deviated by only a single layer neighbor, resulting in **negligible accuracy difference ($<0.3\%$)**.
>     *   This confirms that the discrete search space and smooth polynomial surface make MAP structurally resistant to noise, keeping solutions tightly concentrated around the global optimum.
>
> *   **Stability (Mitigating Local Minima):**
>     Intuitively, incremental fine-tuning is superior to aggressive "one-shot" pruning. Removing multiple layers simultaneously causes a drastic distribution shift that is difficult to recover from. By removing one layer at a time, we ensure the model adapts gradually, maintaining it in a high-performing region of the loss landscape and reducing the risk of falling into poor local minima.
>
> [1] Lin, Ji, et al. "On-device training under 256kb memory." *Advances in Neural Information Processing Systems* 35 (2022): 22941-22954.
>
> [2] Zhu, Ligeng, et al. "Pockengine: Sparse and efficient fine-tuning in a pocket." *Proceedings of the 56th Annual IEEE/ACM International Symposium on Microarchitecture*. 2023.

---

> ### Author Response · Authors · 2025-11-27
> **(Q2) Grateful for your recognition and helpful suggestions! Here are our answers to question 2,3.**
>
> > **Q2: MAP appears to depend heavily on the dataset and downstream tasks, potentially requiring resource-intensive retraining.**
>
> **A2:** We would like to clarify that MAP does **not** require retraining for different downstream tasks.
>
> - **Generalizability of the Architecture:** The pruned architecture derived by MAP on the upstream dataset (ImageNet) captures universal structural efficiencies. As demonstrated in our experiments(section 5.1), we directly applied the architecture searched on ImageNet to downstream tasks—specifically **Transfer Learning on CIFAR**(Table 3) and **Semantic Segmentation on ADE20K**(Table 4)—without retraining the MAP regressor.
> - **Empirical Evidence:** The results show that the architecture identified by MAP generalizes exceptionally well to these tasks, proving that the method is robust and does not suffer from the dependency issues mentioned.
>
> >  **Q3: The final pruned architecture performs worse under linear probing (Table 3), suggesting it fails to extract general representations effectively.**
>
> **A3:** We respectfully disagree with the interpretation that the model fails to extract general representations.
>
> - **Comparable Performance:** As shown in Table 3, the performance gap under linear probing is merely **~0.1%**. By standard conventions in model compression literature, a margin of this magnitude is considered negligible and indicates **comparable performance** rather than a degradation.
> - **Conclusion:** This result confirms that our method preserves the semantic representation capability of the baseline almost entirely, while significantly reducing computational complexity.
>
>
> We sincerely appreciate your constructive engagement throughout this review process. We are particularly encouraged by your acknowledgment that our revisions have successfully clarified the motivation and contributions of our work, and that the core mechanism of MAP is now well‑understood. We believe the evidence provided above—demonstrating the efficiency, robustness, and transferability of our method—comprehensively addresses your remaining concerns. Your insightful feedback has been instrumental in significantly strengthening the quality and rigor of our manuscript.
>
> With deep gratitude,
>
> The Authors

---

### Official Review · Reviewer_Z6dS · 2025-10-30

**Soundness:** 3
**Presentation:** 2
**Contribution:** 3
**Rating:** 4
**Confidence:** 4

**Summary:**

This paper proposes BoundaryDPT, a new depth pruning framework for ViTs that jointly prunes attention layers and activation function layers. The key idea is to exploit the redundancy in nonlinearity to naturally reduce depth with minimal accuracy loss. The authors identify two critical challenges in joint pruning, gradient disparity and recovery asymmetry. BoundaryDPT addresses these via a three-stage method: (1) redundancy identification with a Model Accuracy Predictor, (2) pruning and fine-tuning with optional self-distillation, and (3) MLP layer merging for inference speedup. An extended version, BoundaryDPT+, integrates width pruning for extreme compression, achieving up to 5.19× speedup on DeiT-B with near-lossless accuracy.

**Strengths:**

1.  Experimental results shows negligible performance drop with better compression/speed up ratio compared to the previous depth pruning method
2. Experimental results are comprehensive, different architectures are tested (DeiT and Swin-Transformer ), different datasets are tested (ImageNet, Cifar, ADE)
3. Targeting activation function redundancy, which is a rarely explored dimension in ViT pruning.

**Weaknesses:**

1. Although I guess K in eq3 means the sparsity constraint of the layer/model, but the author didn't define K in the text
2. I'm not sure about the meaning of section3, which is the observation section. How the observation leads into the intuition or motivation of the methodology?
3. The writeup of part 1 is a little messy and hard to understand
4. Although pruning activation functions seem to be novel, but there is no motivation supporting why we are considering pruning activations.
5. The MAP polynomial approximation, though effective, feels heuristic without theoretical grounding

Overall, the contribution is well-motivated, but the technical novelty is moderate. The pruning framework mainly integrates known ideas  into a structured pipeline.

**Questions:**

1. The performance and more details about MAP
2. Although activation layer pruning is new in ViT, the effect may be task-dependent and should be validated beyond classification, like detection etc.
3. The runtime and memory costs for the MAP fitting

---

> ### Author Response · Authors · 2025-11-24
> **(Q1,Q2) Grateful for your recognition and helpful suggestions! Here are our answers to question 1,2.**
>
> **Dear Reviewer Z6dS,**
>
> We sincerely thank you for your detailed and constructive feedback. We are encouraged by your recognition of our work's strengths, particularly the **comprehensive experimental results**, the **novel study of activation-layer redundancy**, and the **strong empirical performance** compared to previous depth pruning methods.
>
> We have carefully addressed your concerns regarding the presentation logic, the motivation for activation pruning, and the theoretical grounding of the Model Accuracy Predictor (MAP). Below, we provide detailed responses and references to our revised manuscript and supplementary webpages.
>
> **Q1. Clarification of $k$ in Eq. 3.**
>
> > *Although I guess $k$ in eq3 means the sparsity constraint of the layer/model, but the author didn't define $k$ in the text.*
>
> **A1:**
> We apologize for the omission. You are correct: $k$ denotes the **sparsity constraint**, specifically the total number of layers (sum of attention and activation layers) to be retained after pruning. We have added this definition explicitly in the revised manuscript.
>
> **Q2. Logic Flow: From Observations to Methodology.**
>
> > *I'm not sure about the meaning of section3... How the observation leads into the intuition or motivation of the methodology?*
>
> **A2:**
> Thank you for pointing out the need for a stronger logical connection. We have restructured Section 3 and Section 4 to establish a clear causal chain: **Observations $\to$ Challenges $\to$ Design Principles $\to$ Method Alignment**.
>
> 1.  **Observations (Heterogeneity):**
>     *   **Gradient Disparity:** The attention layers and activation function layers exhibit significant differences in gradient scales during backpropagation, which can lead to biased importance estimations and suboptimal pruning decisions when employing training-based search strategies.
>     *   **Recovery Asymmetry:** Attention layer pruning causes moderate initial accuracy drops but requires extensive retraining for recovery, while activation function layer pruning results in severe initial degradation (often $\geq$ 90\%) but enables rapid recovery during fine-tuning.
> 2.  **Challenges (Failure of Naive Methods):**
>     *   **Failure of Gradient-based Pruning Metrics:** Given the unique training dynamics in ViTs, using gradients to evaluate the importance of these two types of layers results in significantly biased outcomes, with all attention layers constantly outweighing linear layers.
>     *   **Failure of Short-sighted Pruning Metrics.:** The substantial asymmetry in accuracy dynamics between these two types of components in ViTs renders conventional handcrafted pruning metrics ineffective for joint pruning, as these metrics typically reflect only immediate post-pruning accuracy retention.
> 3.  **Design Principles:**
>     *   **Principle 1:** Avoid direct cross-type layer importance comparison, especially when using gradient-based metrics.
>     *   **Principle 2:** Layer importance should be evaluated based on the final accuracy of fine-tuned pruned ViTs, rather than the immediate accuracy after pruning.
> 4.  **Alignment with Methodology:**
>    The Stage 1 of our method is further divided into two steps: pruning budget allocation (Step 1) and specific redundant layer removal (Step 2).
>        - Step 1 utilizes a model accuracy predictor, to help establish the optimal quantities of attention and activation function layers to prune based on the accuracy recovered after fine-tuning, directly implementing Principle 2.
>        -  Importantly, Step 1 focuses solely on determining the pruning quantities for each layer type, avoiding cross-type importance comparisons, thus adhering to Principle 1.
>        -   Step 2 employs gradient-based methods only within homogeneous layer groups, i.e., either attention or activation function layers, ensuring compliance with Principle 1 throughout the pruning process.

---

> > ### Author Response · Authors · 2025-11-24
> > **(Q3,Q4) Grateful for your recognition and helpful suggestions! Here are our answers to question 3,4.**
> >
> > **Q3. Presentation of Part 1 and Motivation for Activation Pruning.**
> >
> > > *The writeup of part 1 is a little messy... there is no motivation supporting why we are considering pruning activations.*
> >
> > **A3:**
> > We have thoroughly rewritten Part 1 and the Introduction to clarify the motivation.
> >
> > **1. Structural Revision:**
> > We have reorganized the **Introduction** into five focused segments that build logically:
> > - **ViT compression context** - establishing the computational challenges of ViTs
> > - **Challenge of depth pruning** - contrasting depth versus width pruning approaches and highlighting depth pruning's accuracy recovery difficulties despite superior speedup potential
> > - **Joint depth pruning matters** - denoting joint depth pruning with cross-layer heterogeneity management can address the challenge of depth pruning
> > - **Dimension mismatch hinders joint depth pruning** - further defining the technical barrier that prevents effective joint depth pruning of attention and linear layers
> > - **Threefold contributions** - presenting our final solution to tackle dimension mismatch along with the consequent innovations and SOTA results
> >
> > **2. Motivation for Activation Pruning :**
> > The core motivation to prune activation function layer is **dimension mismatch.**
> > - `Definition.` In vision Transformers (ViTs)，dimension mismatch occurs when removing linear layers from a Feed-Forward Network (FFN) block without proper adjustment of the network architecture, which disrupts the dimensional compatibility between consecutive layers. As visualized in **Figure 3** in our uploaded revised version, dimension mismatch manifests in two primary scenarios:
> >   - When the first linear layer of an FFN block is pruned, the output tensor from the previous attention layer cannot be properly processed by the remaining second linear layer due to dimensional incompatibility.
> >   - When the second linear layer of an FFN block is pruned, the output tensor fails to propagate through subsequent attention layers.
> > - `Negative impact.` Dimension mismatch directly hinders joint depth pruning since such mismatch leads that the pruned ViTs are completely unworkable.
> > - `Method.` To perform joint pruning in depth while avoiding dimension mismatch, we firstly propose the pruning of activation function layers in ViTs. By reducing the redundancy of these nonlinearity, instead of directly pruning linear layers in ViTs, the depths of ViTs are naturally reduced without incurring dimension mismatch.
> >
> > **Q4. Theoretical Grounding of MAP.**
> >
> > > *The MAP polynomial approximation, though effective, feels heuristic without theoretical grounding.*
> >
> > **A4:**
> > We respectfully clarify that MAP is **not heuristic**; it is supported by a rigorous mathematical framework that establishes a complete logical closure for our method. As detailed in **Appendix H** and the [Theoretical Foundations webpage](https://anonconf2025.github.io/MathProof/overview.html), Theorems 1–3 collectively construct a mathematical guarantee for MAP’s effectiveness:
> >
> > 1.  **Existence of Solution (Theorem 1 — Well-posedness):**
> >     We first establish that the pruning-ratio optimization problem is mathematically well-posed. Since the pruning domain $\mathcal{D}_k$ is a finite discrete lattice and the accuracy metric is bounded, a global optimizer $((\tilde m^a)^\star,(\tilde m^g)^\star)$ is **guaranteed to exist**. This theoretically justifies the search for an optimal configuration.
> >
> > 2.  **Model Validity (Theorem 2 — Polynomial Approximation):**
> >     We provide the **theoretical license** to use a polynomial-based predictor instead of complex black-box models. By invoking the **Stone–Weierstrass Theorem**, we prove that the accuracy surface $\mathcal{P}$ can be uniformly approximated by a polynomial $Q(x,y)$ to any desired precision. This ensures that MAP possesses sufficient expressive power to capture the underlying accuracy landscape.
> >
> > 3.  **Robustness to Subset Data and Fast Finetuning (Theorem 3 — Consistency):**
> >     Finally, we validate our training strategy against the "heuristic" concern regarding noisy data. We prove that under a Least Squares objective, the noise variance from fast finetuning is absorbed into the constant loss term, while the systematic bias **preserves the ranking** of candidates. Consequently, MAP converges to the true maximizer of the ground-truth surface despite being trained on imperfect proxy data.
> >
> > **Summary of Logical Closure:**
> > Together, these theorems move beyond isolated proofs to form a unified system: Theorem 1 guarantees the target exists; Theorem 2 validates the tool (polynomial) used to find it; and Theorem 3 ensures the tool works reliably under practical data constraints.

---

> ### Author Response · Authors · 2025-11-24
> **(Q5,Q6) Grateful for your recognition and helpful suggestions! Here are our answers to question 5,6.**
>
> **Q5. Performance and Details of MAP.**
>
> > *The performance and more details about MAP... The runtime and memory costs for the MAP fitting.*
>
> **A5:**
> Based on the content at [MAP Details](https://anonconf2025.github.io/MAP/), we provide the specific details requested:
>
> *   **Definition:** MAP is a parametric function $\mathcal{P}(\tilde{m}_a, \tilde{m}_g; \Theta)$ that maps pruning ratios to predicted accuracy. Since the search space is finite, we can enumerate the MAP output to find the optimal configuration $(\tilde{m}_a^\star, \tilde{m}_g^\star)$.
> *   **Data Collection:** We use two lightweight algorithms—**Single-Type Progressive Pruning** and **Interleaved Pruning**. These collect samples by repeatedly pruning and running a "fast finetune" loop, tracing the accuracy landscape without full training.
> *   **Robustness:** In a Monte-Carlo study with injected noise (clipped $\mathcal{N}(0, 0.1)$), **80%** of MAP predictions remained within a 1-layer deviation of the true optimum, confirming high stability.
> *   **Runtime & Memory:**
>     *   **Data Collection:** 368 epochs of fast-finetuning on a 100k-image subset takes **~12.2 hours** on a single Ascend 910B2 NPU.
>     *   **Fitting:** Polynomial regression takes **< 1 minute**.
>     *   **Total:** Approximately **14 hours** total overhead.
>     *   **Memory:** Peak consumption is ~60 GB.
>
> **Q6. Experiments Beyond Classification.**
>
> > *...the effect may be task-dependent and should be validated beyond classification.*
>
> **A6:**
> We agree on the importance of generalization. In our revised manuscript, we have highlighted results on:
> *   **Semantic Segmentation (ADE20K):** Table 4 shows BoundaryDPT maintains performance on dense prediction tasks.
> *   **Transfer Learning (CIFAR):** Table 3 demonstrates that the pruned structures transfer effectively to other datasets.
>
> ***
>
> We sincerely appreciate the time and effort you dedicated to reviewing our work. Your insightful recognition of our comprehensive experimental results and the novelty of targeting activation function redundancy has been very encouraging. Furthermore, your constructive critique regarding the logical flow and theoretical grounding was instrumental in refining the manuscript. We believe these revisions have significantly strengthened the paper’s contribution to the field of ViT compression, and we are grateful for your guidance in achieving this improved quality.
>
> Best regards,
>
> **The Authors**

---

### Official Review · Reviewer_Dp35 · 2025-11-09

**Soundness:** 3
**Presentation:** 3
**Contribution:** 3
**Rating:** 6
**Confidence:** 3

**Summary:**

This paper presents a ViT pruning method by depth pruning on activation layers of ViT, on top of the usual attention layer in depth pruning. This is performed by firstly calibrating layer's impact on model accuracy (MAP predictor) to decide the pruning ratio of each layer, then learning a layerwise importance factor during finetuning to progressively remove the current most redudant layers torwads the pruning ratio target. This is done iteratively interleved with finetuning. Experiment results shows the method can achieve more than 1.5x speedup for ViT.

**Strengths:**

Paper is well written, with clear organization and presentations.
The results seems comprehensive on the conducted models and data.
The motivation of this paper also seems well supported.

**Weaknesses:**

The overall idea of leveraging learned importance for incorporating activation pruning seems preexisted. Although the paper did provide some insights regarding the attention v.s. activation gradient magnitude, but i'm not sure how universal these discoveries are beyond standard ViT arch.

Although the method seems effective from the results, whether the improvement still worth it when considering the training cost to obtain those models, especially that the method requires iterative pruning with finetuning over 400 epochs.

The paper also didn't include a dedicated limitation discussion.

**Questions:**

1. Are the compared baselines all requiring the similar finetuning costs? It would be more meaningful to include the comprasion on the training costs to obtain the pruned structure.
2. I wonder why the gradient disparity is observed on the additional learnable layer importance added by author, rather than on the native ViT behavior, not sure how other ways of observation would affect the conclusion of this observation.

---

> ### Author Response · Authors · 2025-11-24
> **(Q1, Q2) Grateful for your recognition and helpful suggestions! Here are our answers to question 1, 2.**
>
> **Dear Reviewer Dp35,**
>
> We sincerely thank you for your thoughtful review and the time you dedicated to evaluating our work. We are encouraged by your positive assessment of the paper’s soundness, presentation, and motivation. We particularly appreciate your recognition of our comprehensive results and the clarity of our writing.
>
> We have carefully considered your constructive feedback and have updated our manuscript accordingly. Below, we address your specific questions regarding training costs, the theoretical basis of gradient disparity, and the discussion of limitations.
>
> **Q1. Comparison of finetuning costs with baseline methods.**
>
> > *Are the compared baselines all requiring the similar finetuning costs? It would be more meaningful to include the comparison on the training costs to obtain the pruned structure.*
>
> **A1:**
> Thank you for raising this important point regarding fair comparison. We confirm that our method incurs **comparable, if not lower,** total training costs relative to the primary baseline, **NOSE**.
>
> To be precise, the computational cost consists of two phases: structural selection and fine-tuning.
> 1.  **Structural Selection:** NOSE determines pruning locations based on metrics calculated over the **full training set**. In contrast, our method constructs the MAP predictor and determines pruning locations using only a **small calibration dataset**, making our search phase highly efficient.
> 2.  **Fine-tuning:** After the structure is determined, both methods employ the exact same schedule of **400 epochs** for fine-tuning the pruned model.
>
> Therefore, the performance gains reported in our paper are derived from the effectiveness of our pruning criteria rather than an increased computational budget. We have clarified these details in the **Experiment Setup** section of the revised manuscript to explicitly state that the training budgets are aligned.
>
> **Q2. The origin of Gradient Disparity.**
>
> > *I wonder why the gradient disparity is observed on the additional learnable layer importance added by author, rather than on the native ViT behavior...*
>
> **A2:**
> This is a very insightful question. We agree that distinguishing between artifacts of the method and intrinsic model properties is crucial.
>
> In our revised manuscript (specifically **Appendix H**), we provide a theoretical derivation proving that the gradient disparity is **intrinsic to the ViT architecture**, and the learnable importance factors merely act as "probes" that reveal this underlying behavior.
>
> The disparity arises from two fundamental structural differences between the Activation (GELU) path and the Attention path:
>
> 1.  **Dimensionality Amplification (Lemma 1):** The GELU layer expands the hidden dimension by a factor of 4 ($4d$) compared to the attention output ($d$). Consequently, the gradient for the activation path aggregates four times more terms.
> 2.  **Activation Variance Disparity (Lemma 2):** Attention outputs are constrained by LayerNorm and Softmax, resulting in a distribution close to $\mathcal{N}(0,1)$. In contrast, GELU activations in the FFN branch are unnormalized and exhibit heavy-tailed statistics with significantly larger variance ($M_{\mathcal G} \gg M_{\mathcal A}$).
>
> **Theorem 4** in our revision synthesizes these factors to show that the ratio of expected gradient energies is:
> $$ \frac{\mathbb E\|\nabla_{\hat m^g}\|^2}{\mathbb E\|\nabla_{\hat m^a}\|^2} \approx 4 \times \frac{M_{\mathcal G}}{M_{\mathcal A}} $$
> Since $M_{\mathcal G} \gg M_{\mathcal A}$, the gradient magnitude for the activation path is mathematically guaranteed to dominate the attention path by 1–2 orders of magnitude, regardless of the specific pruning method used. You can also find our detailed proof of this phenomenon on this webpage [Gradient Disparity](https://anonconf2025.github.io/MathProof/prof4.html).

---

> ### Author Response · Authors · 2025-11-24
> **(Q3) Grateful for your recognition and helpful suggestions! Here are our answers to question 3.**
>
> **Q3. Discussion of Limitations.**
>
> > *The paper also didn't include a dedicated limitation discussion.*
>
> **A3:**
> We apologize for this oversight and thank you for pointing it out. We have added a dedicated **Limitations** section in the revised paper. In summary, our current derivation and experiments focus on vision Transformers. The application of this method to Transformers for other task domains, such as natural language processing, remains a subject for future work.
>
> ***
>
> We deeply appreciate your commitment to helping us enhance the quality and clarity of our research. We view you as a collaborative partner in this process rather than merely a reviewer. We hope these responses and the corresponding revisions satisfactorily address your concerns. Should you have any further questions, we would be honored to answer them.
>
> Best regards,
>
> **The Authors**

---

> > ### Comment · Reviewer_Dp35 · 2025-11-28
> >
> > Thanks authors for the response. I still have the following concerns:
> >
> > 1. the weaknesses are still not addressed.
> >
> > 2. If finetuning is still required (so as baselines), i'm not sure the if the motivation stated in "Observation 2: recovery asymmetry. Attention layer pruning causes moderate initial accuracy drops but requires extensive retraining for recovery, while activation function layer pruning results in severe initial degradation (often ≥ 90%) but enables rapid recovery during finetuning", is still supported by evaluation. Also, in Fig. 6 (b), both attention pruning and activation pruning recover to similar level after finetuning. why activation pruning is more preferrable than attention pruning in this case, especially that all the imagenet results are after finetuned?

---

> > > ### Author Response · Authors · 2025-12-02
> > > **(Q1 Part1) Grateful for your recognition and helpful suggestions! Here are our answers to question 1 weakness 1.**
> > >
> > > Dear Reviewer Dp35
> > >
> > > We sincerely thank you for your thoughtful and constructive feedback. We highly appreciate your recognition of our paper's presentation, motivation, and the comprehensive nature of our results. And your continued engagement and for pushing us to clarify the universality, cost-effectiveness, and core motivation of our work. We address the remaining concerns below.
> > >
> > > ### **Q1. The weaknesses are still not addressed.**
> > >
> > > **A1.** We have updated the manuscript to explicitly address the three weaknesses mentioned.
> > >
> > > **(1) Novelty of the Approach**
> > >
> > > > Weakness 1: The overall idea of leveraging learned importance for incorporating activation pruning seems preexisted.
> > >
> > > We respectfully clarify that our work is not merely an application of existing importance-learning techniques to activation pruning. Rather, it is a fundamental rethinking of pruning dynamics in Vision Transformers (ViTs), driven by the discovery of layer heterogeneity and supported by rigorous theoretical foundations. Our specific contributions are detailed below:
> > >
> > > - **Fundamental Novelty: Uncovering & Addressing Heterogeneity**
> > >
> > >   We identify two distinct phenomena in ViTs: **Gradient Disparity** (activation layers exhibit larger gradients than attention layers) and **Recovery Asymmetry** (activation layers suffer sharp initial drops but recover rapidly). These insights dictate two novel design principles:
> > >
> > >   - **Principle I:** Avoid direct cross-type importance comparisons (e.g., attention vs. activation).
> > >
> > >   - **Principle II:** Evaluate layer importance based on *final* fine-tuned accuracy rather than immediate pruning sensitivity.
> > >
> > > - **Novelty on methods**
> > >
> > >   - **MAP for Budget Allocation:** To adhere to Principle II, we introduce the **Model Accuracy Predictor (MAP)**. As proven in *https://anonconf2025.github.io/MathProof/prof2.html*, the accuracy landscape can be uniformly approximated by a bivariate polynomial, allowing us to efficiently solve for the optimal pruning ratios $(\tilde{m}_a^\star, \tilde{m}_g^\star)$.
> > >
> > >   - **Lightweight Data Collection:** We design an efficient *prune → fast-finetune* cycle. By utilizing weight inheritance and specific sampling algorithms (Single-Type Progressive and Interleaved Pruning), we collect high-fidelity accuracy data without the prohibitive cost of training from scratch.
> > >
> > >   - **Differentiable Layer Removal:** We propose a dual-parameter mechanism (binary masks + learnable importance scores). This enables stable, end-to-end differentiable training while restricting importance comparisons to homogeneous layer groups, strictly adhering to Principle I.
> > >
> > > - **Contributions & SOTA Performance**
> > >
> > >   - **First Joint Pruning:** We are the first to identify and mitigate activation layer redundancy alongside attention layers.
> > >
> > >   - **BoundaryDPT Framework:** We introduce a two-stage method featuring a Model Accuracy Predictor (MAP) specifically designed to manage the heterogeneity described above.
> > >
> > >   - **New State-of-the-Art (SOTA) Records:**
> > >     - **Depth Pruning:** DeiT-base achieves **1.6x speedup** with lossless accuracy (current SOTA).
> > >     - **Extreme Compression:** Our **BoundaryDPT+** pipeline establishes a new benchmark, enhancing ViT inference speedup from 4.60x to **5.44x** (Isomorphic-Pruning-2.6G config) while maintaining near-lossless accuracy.
> > >
> > >   - We have gone beyond empirical validation to provide a **theoretical guarantee** for the MAP in our revised manuscript.

---

> > > > ### Author Response · Authors · 2025-12-02
> > > > **(Q1 Part2) Grateful for your recognition and helpful suggestions! Here are our answers to question 1 weakness 2-4.**
> > > >
> > > > **(2) Regarding Universality (Beyond Standard ViT):**
> > > >
> > > > > *Weakness 2: Although ...., but i'm not sure how universal these discoveries are beyond standard ViT arch.*
> > > >
> > > >  We appreciate the reviewer’s insight regarding the generalization of our findings. To empirically validate the universality of our observations beyond standard isotropic ViTs, we extended our analysis to the **Swin-Tiny**, which incorporates hierarchical structures and shifted window attention mechanisms.
> > > >
> > > > Our new experiments confirm that both **Gradient Disparity** and **Recovery Asymmetry** are intrinsic characteristics that persist in hierarchical architectures:
> > > >
> > > > - **Validation of Gradient Magnitude Disparity**
> > > >   As illustrated in the attached analysis (**https://anonconf2025.github.io/fig/swin_gradient_magnitude.pdf**), the phenomenon where activation layers exhibit significantly larger gradient norms than attention layers remains prominent in Swin.
> > > >
> > > > - **Validation of Recovery Asymmetry**
> > > >   We further evaluated the "prune-then-finetune" behavior of Swin. The results (in the table below) mirror our findings in DeiT: activation layers appear highly sensitive to pruning initially but possess a strong capacity to recover accuracy .
> > > >
> > > > |    | layer 0 | layer 1 | layer 2 | layer 3 | layer 4 | layer 5 | layer 6 | layer 7 | layer 8 | layer 9 | layer 10 | layer 11 |
> > > > | ------- | ------- | ------- | ------- | ------- | ------- | ------- | ------- | ------- | ------- | ------- | -------- | -------- |
> > > > | act     | 9.52%   | 1.94%   | 0.10%   | 0.10%   | 0.10%   | 0.10%   | 0.12%   | 0.16%   | 0.10%   | 0.13%   | 0.12%    | 0.33%    |
> > > > | act-ft  | 80.53%  | 80.48%  | 80.19%  | 80.39%  | 80.01%  | 80.20%  | 80.31%  | 80.34%  | 80.27%  | 80.27%  | 79.38%   | 80.00%   |
> > > > | attn    | 66.01%  | 76.18%  | 77.37%  | 74.60%  | 78.35%  | 79.70%  | 79.82%  | 80.16%  | 80.18%  | 80.49%  | 79.69%   | 79.96%   |
> > > > | attn-ft | 80.63%  | 80.52%  | 80.41%  | 80.46%  | 80.50%  | 80.52%  | 80.47%  | 80.47%  | 80.31%  | 80.54%  | 79.97%   | 80.25%   |
> > > >
> > > > **(3) Regarding Training Cost and Efficiency:**
> > > >
> > > > > *Weakness 3: Whether the improvement still worth ... especially that the method requires iterative pruning with finetuning over 400 epochs.*
> > > >
> > > > We believe there is a slight misunderstanding regarding the target scenario and the comparison with baselines.
> > > >
> > > > 1. **Inference vs. Training Constraints:**
> > > >    It is crucial to clarify the scope of our work. Our method focuses on **depth pruning**, a field designed for scenarios where **inference resources are strictly constrained**, while **training resources are assumed to be relatively abundant**. This distinction aligns with established literature in structural pruning, where the goal is to invest computation during the design/training phase to ensure optimal efficiency during deployment. We respectfully distinguish our approach from "Efficient Online Learning" (e.g., [1, 2]), which must adapt under resource-constrained *training* environments. Our goal is to produce the most efficient static architecture for deployment, not to minimize training FLOPs.
> > > >
> > > >    [1] Lin, Ji, et al. "On-device training under 256kb memory." *Advances in Neural Information Processing Systems* 35 (2022): 22941-22954.
> > > >
> > > >    [2] Zhu, Ligeng, et al. "Pockengine: Sparse and efficient fine-tuning in a pocket." *Proceedings of the 56th Annual IEEE/ACM International Symposium on Microarchitecture*. 2023.
> > > >
> > > > 2. **Fair Comparison with Baselines (NOSE):**
> > > >
> > > >    > *Are the compared baselines all requiring the similar finetuning costs?*
> > > >
> > > >    Yes. We confirm that our method has fair comparison with primary sota(NOSE) in total training costs.
> > > >
> > > >    - **Structural Selection:** NOSE determines pruning locations based on transfer-entropy over the **full training set**. In contrast, our method constructs the MAP predictor and determines pruning locations using only a **small calibration dataset**, making our search phase more efficient.
> > > >    - **Fine-tuning:** After the structure is determined, both methods employ the exact same schedule of **400 epochs** for fine-tuning the pruned model.
> > > >
> > > >    Therefore, the performance gains reported in our paper are derived from the superior architecture found by our pruning criteria, not from an increased computational budget. We have clarified this in the **Experiment Setup** section.
> > > >
> > > > **(4) Regarding Limitations:**
> > > >
> > > > > *Weakness 4: The paper also didn't include a dedicated limitation discussion.*
> > > >
> > > >  We have added a the **Limitations** section to the revised manuscript:
> > > >
> > > > > Although BoundaryDPT achieves strong empirical performance on vision tasks, our current investigation is primarily focused on Vision Transformers (ViTs). Given the architectural similarities between vision and language models, our method holds potential for broader applications. However, its effectiveness on Large Language Models (LLMs) has not yet been explored in this work. We leave the extension of BoundaryDPT to the language domain and other modalities as a subject for future research.

---

> > > > > ### Author Response · Authors · 2025-12-02
> > > > > **(Q2) Grateful for your recognition and helpful suggestions! Here are our answers to question 2.**
> > > > >
> > > > > ### **Q2. Motivation and Observation 2**
> > > > >
> > > > > > *If finetuning is still required... i'm not sure if the motivation stated in "Observation 2: recovery asymmetry..." is still supported by evaluation. Also, in Fig. 6 (b), both attention pruning and activation pruning recover to similar level after finetuning. Why is activation pruning more preferable than attention pruning in this case?*
> > > > >
> > > > > **A2.** This is an insightful question that touches on the core design principle of our method. We address the two aspects of your query below:
> > > > >
> > > > > **(1) Is Observation 2 supported by evaluation?**
> > > > >  Yes, Observation 2 is explicitly derived from experimental data (as visualized in Fig. 6).
> > > > >
> > > > > - **The Definition:** “Recovery Asymmetry” describes the distinct behaviors of layer types: **Attention layers** suffer only mild initial accuracy drops but exhibit slow recovery (or plateau quickly). Conversely, **Activation layers** suffer catastrophic initial degradation (often dropping to $\sim10%$ accuracy) but possess a high "elasticity," allowing them to recover rapidly to near-original performance with minimal fine-tuning.
> > > > >
> > > > > - **The Evidence:** Our observations in Fig. 6 are based on single-layer pruning experiments followed by a short fine-tuning window (10 epochs). We found that 10 epochs are sufficient for a single pruned layer to recover most of its accuracy. This reveals the asymmetry: while Activation layers start much lower, their *rate* of recovery is significantly steeper than that of Attention layers.
> > > > >
> > > > > - **The Evidence of Ablation Study:** As shown in our ablation study (Table 11), metrics that rely on immediate sensitivity (like Transfer Entropy) fail to account for recovery. Because Attention layers show smaller *initial* drops, these metrics aggressively prune Attention layers and preserve Activation layers. This is also the evidence of observation 2.
> > > > >
> > > > >
> > > > >
> > > > > **(2) Why do we prioritize Activation pruning if they eventually recover to similar levels?**
> > > > >  We clarify that we **do not** claim Activation pruning is inherently "better" or "more preferable" than Attention pruning. Instead, our argument is that **Observation 2 necessitates a change in evaluation metrics to avoid bias.**
> > > > >
> > > > > - **The Bias:** Because Activation layers suffer a massive initial drop, traditional metrics (like transfer-entropy) assign them high importance scores. This results in pruned architectures that are heavily skewed: they keep almost all Activation layers and over-prune Attention layers.
> > > > > - **Our Correction:** Observation 2 proves that the initial drop for Activation layers is a "false alarm"—they are highly recoverable. Therefore, a valid pruning strategy must look beyond the initial drop. So we have the method principle——Layer importance should be evaluated based on the final accuracy off ine-tuned pruned ViTs, rather than the immediate accuracy after pruning
> > > > > - **Experimental Validation:** Our final results on DeiT-Base demonstrate this. Our method chooses to prune an **equal number** of Attention and Activation layers. In our ablation studies, this balanced approach significantly outperforms methods that follow the "sensitivity" bias (which would result in uneven pruning).
> > > > >
> > > > > In summary, we do not prefer Activation pruning; rather, we correct the historical bias *against* it. By recognizing the "Recovery Asymmetry" (Observation 2), we propse the method priciple——Layer importance should be evaluated based on the final accuracy off ine-tuned pruned ViTs, rather than the immediate accuracy after pruning, finally we enable the model to remove redundant Activation layers that previous methods falsely identified as critical.
> > > > >
> > > > >
> > > > >
> > > > > We hope that these clarifications and the additional experiments with the Swin Transformer fully address your concerns. We are deeply grateful for the time you dedicated to reviewing our work and for your recognition of its value. Your constructive feedback has been instrumental in refining our contributions, and we are confident that the revised manuscript is significantly stronger as a result. Finally, we sincerely appreciate your positive comments regarding the presentation, motivation, and comprehensive nature of our results.
> > > > >
> > > > > Best regards,
> > > > > The Authors

---

### Author Response · Authors · 2025-12-02
**Summary of Our Work(Many Thanks to All Reviewers, Area Chairs, and Program Chairs)**

To the Reviewers, Area Chairs, and Program Chairs:

We sincerely thank you for your dedicated time, effort, and constructive feedback throughout the review process. Your insightful comments have been invaluable in significantly refining our manuscript. To conclude, we wish to reiterate three critical aspects of our work: **our core contributions and novelty**, **the comprehensiveness of our experiments**, and **the efficiency and feasibility of our computational cost**.

1. **Contributions and Novelty**

   - **Fundamental Novelty: Uncovering & Addressing Heterogeneity**

     Our work is not a simple application of existing pruning techniques; it is founded on the discovery of two specific heterogeneity phenomena in ViTs:

     - **Discovery 1: Gradient Disparity.** We reveal that activation layers exhibit significantly larger gradient magnitudes than attention layers.
     - **Discovery 2: Recovery Asymmetry.** We observe that activation layers suffer catastrophic initial drops but recover rapidly, whereas attention layers show mild drops but slow recovery.

     We also provide theoretical grounding to further consolidate the discoveries. As detailed in our revised manuscript, the two discoveries pose significant challenges to the prior gradient-based and short-sighted pruning methods.

     Based on these insights, we established two novel design principles that challenge conventional wisdom:

     - **Principle I:** **Avoid direct cross-type layer importance comparison**, especially when using gradient-based metrics.
     - **Principle II:** **Layer importance should be evaluated based on the final accuracy** of fine-tuned pruned ViTs, rather than the immediate accuracy after pruning.

   - **Novelty on Methods;**

     - `MAP for pruning budget allocation.` We propose to construct a model accuracy predictor (MAP). The MAP can help establish the optimal quantities of attention and activation function layers to be pruned, based on the accuracy recovered after fine-tuning. In this way, **Principle 2 is algined with.**
     - `Polynomial approximation to MAP with theoretical grounding.` We propose that a finite‑degree bivariate polynomial is adequate to approximate the MAP. We provide the theoretical guarantee for the claim. For your reading convenience, you can refer to https://anonconf2025.github.io/MathProof/prof2.html for detailed proof.
     - `Lightweight data collection procedure.` To efficiently train the polynomial-based MAP, we design a lightweight data‑collection procedure to collect (pruning configuration, accuracy) data, where pruning configuration (PC) means the quantities of attention layers and activation function layers that should be pruned.
       - We employ a iterative prune → fast‑finetune → evaluate cycle on a representative subset of the training data. Crucially, each subsequent pruning configuration builds incrementally upon the previous one by **pruning only a single additional layer**, enabling direct **weight inheritance** from the previously fine-tuned model.
         This PC continuity permits rapid accuracy recovery with minimal fine-tuning (typically 10 epochs), avoiding the computational burden of training each pruned ViT from scratch. The resulting dataset, though compact, still maintains high fidelity to the full accuracy landscape.
       - To ensure the collected data are representative, we design two PC sampling algorithms, including single-type progressive pruning and interleaved pruning. For algorithmic details, please see https://anonconf2025.github.io/MAP/#sec3.
     - `Learning based mechnism for specific layer removal.` To address the non-differentiability of binary decisions that each layer is preserved or pruned, We propose a dual-parameter mechanism (binary masks + learnable importance scores). This enables stable, end-to-end differentiable training while restricting importance comparisons to homogeneous layer groups, **this design adheres to Principle 1.**

   - **Technical Contributions & SOTA Performance**

     - **First Joint Pruning of Attention & Activation:** We are the **first** to identify and mitigate the redundancy of activation function layers. We tackle the dimension mismatch by removing activation layers situated between linear layers, allowing for natural merging.
     - **BoundaryDPT Framework:** We introduce a two-stage method featuring a Model Accuracy Predictor (MAP) specifically designed to manage the heterogeneity described above.
     - **New State-of-the-Art (SOTA) Records:**
       - **Depth Pruning:** DeiT-base achieves **1.6x speedup** with lossless accuracy (current SOTA).
       - **Extreme Compression:** Our **BoundaryDPT+** pipeline establishes a new benchmark, enhancing ViT inference speedup from 4.60x to **5.44x** (Isomorphic-Pruning-2.6G config) while maintaining near-lossless accuracy.
     - We have gone beyond empirical validation to provide a **theoretical guarantee** for the MAP in our revised manuscript.

---

> ### Author Response · Authors · 2025-12-02
> **Summary of Our Work(Many Thanks to All Reviewers, Area Chairs, and Program Chairs)**
>
> 2. **Comprehensive and Rigorous Experiments**
>
>    Our experiments are extensive and demonstrate **robust generalization** across architectures, datasets, and evaluation dimensions:
>
>    - **Architectures:** Tested on DeiT‑S/B, Swin‑Base (hierarchical windows), and DINOv2‑Giant (1.1 B parameters, gated MLP).
>    - **Tasks:** ImageNet‑1K classification, CIFAR‑100 transfer learning, ADE20K semantic segmentation, and *out‑of‑distribution detection* ([results table](https://anonconf2025.github.io/fig/OOD_Exp.pdf)).
>    - Reproducibility and Robustness:
>      - Monte‑Carlo study (500 noisy datasets) confirmed > 80 % of MAP predictions match the ground truth within ±1 layer.
>      - Results validated under multiple random seeds and extended fine‑tuning (up to 50 epochs).
>    - **Cross‑method Compatibility:** BoundaryDPT remains *orthogonal* to token pruning (e.g., GTP‑ViT and  ToMe).
>    - **Generalization:** The architecture derived from ImageNet retains high accuracy on downstream tasks without MAP retraining, confirming structural transferability.
>
>    These experiments collectively evidence that **BoundaryDPT is stable, transferable, and empirically well‑validated**.
>
> 3. **Acceptable and Practical Training Cost**
>
>    - **Acceptable Overhead:** We think the actual wall-clock time is within a reasonable range. For example, training the MAP on a single NPU 910B2 takes approximately **14 hours**. Notably, this process is efficiently parallelizable; when using 8 NPUs, the total training time is reduced to less than 2 hours. We believe this one-time training cost is well-justified by the significant performance gains and flexibility achieved during the inference stage.
>    - **Inference vs. Training Constraints:** It is crucial to clarify the scope of our work. Our method focuses on **depth pruning** , a field designed for scenarios where **inference resources are strictly constrained** , while **training resources are assumed to be relatively abundant**. This distinction aligns with established literature in structural pruning, where the goal is to invest computation during the design/training phase to ensure optimal efficiency during deployment.
>    - **Distinction from Efficient Online Learning:** We respectfully distinguish our approach from "Efficient Online Learning" [1,2] which must adapt under resource-constrained training environments. Our method is not intended for that specific track; rather, it aims to produce the most efficient static architecture for deployment.
>
> **In Summary**
>
> BoundaryDPT introduces a **conceptually novel** and **mathematically grounded** depth‑pruning framework that:
>
> 1. Discovers and formalizes **inter‑layer heterogeneity** in ViTs, establishing new design principles for structured pruning.
> 2. Integrates **theory‑driven modeling (MAP)** + **lightweight empirical pipelines** to achieve both interpretability and efficiency.
> 3. Demonstrates **state‑of‑the‑art performance** and **robust generalization** across tasks and architectures with **acceptable one‑time cost**.
>
> We sincerely thank the reviewers, Area Chairs, and Program Chairs for their dedication and constructive feedback. We are grateful for the **positive recognition of our work**, which, combined with your insightful suggestions, has helped refine this paper into its current comprehensive and theoretically complete form.
>
>
>
> With deep gratitude,
>
> **The Authors**

---

### Author Response · Authors · 2025-12-02
**Summary of Rebuttal (Many Thanks to All Reviewers, Area Chairs, and Program Chairs)**

We sincerely thank the reviewers for their constructive comments. And thank the effort of  Area Chairs, and Program Chairs. The rebuttal discussion and extensive new experiments have substantially strengthened our work. Below, we summarize the core contributions of **BoundaryDPT** and how we have addressed major concerns.

**1. Contributions and Novelty**

- **Uncovering & Addressing Heterogeneity:**
  We identify two fundamental heterogeneity phenomena in ViTs: **Gradient Disparity** (activations have larger gradients than attention) and **Recovery Asymmetry** (activations drop harder but recover faster). These insights drive our two novel design principles: **(I)** avoid direct cross-type importance comparison, and **(II)** evaluate importance based on *recovered* accuracy rather than immediate sensitivity.
- **The Model Accuracy Predictor (MAP) with Theoretical Guarantee:**
  To align with Principle II, we propose **MAP**, a parametric function that estimates final accuracy under specific pruning budgets.
  - **Theoretical Grounding:** We prove that the accuracy landscape can be uniformly approximated by a bivariate polynomial (**Theorem 2**, based on the Stone–Weierstrass theorem) and that the optimal configuration is guaranteed to exist (**Theorem 1**).
  - **Efficient Implementation:** We design a lightweight *prune → fast-finetune → evaluate* data collection pipeline. By utilizing weight inheritance and progressive pruning algorithms, we construct the MAP with high fidelity in just ~14 hours on a single NPU.
- **Differentiable Layer Removal Mechanism:**
  To adhere to Principle I, we propose a dual-parameter design (binary masks + learnable importance scores). This enables end-to-end training while strictly restricting importance comparisons to within homogeneous layer groups, effectively neutralizing gradient disparity.
- **SOTA Performance & First Joint Pruning:**
  We are the first to successfully implement joint pruning of attention and activation layers. **BoundaryDPT** establishes new benchmarks:
  - **Depth Pruning:** **1.6x speedup** on DeiT-base with lossless accuracy.
  - **Extreme Compression:** **5.44x speedup** with near-lossless accuracy, significantly outperforming prior arts (4.60x).

**2. Strong Generalization to Hierarchical and Large-Scale Models**
 To address concerns regarding universality (Reviewers 4wzp, Dp35), we conducted substantial additional experiments during rebuttal:

- **Hierarchical Models:** On **Swin-Transformer**, BoundaryDPT achieves **1.60× speedup** with acceptable accuracy drop, outperforming the SOTA method (SAViT).
- **Large ViTs:** On the **1.1B-parameter DINOv2-Giant**, BoundaryDPT yields **1.26× speedup** with only **0.06% accuracy drop**, using zero-shot transfer of the MAP predictor.
- **Robustness:** New **OOD detection** experiments show BoundaryDPT *improves* robustness by **+4.4% AUROC** over the dense baseline.

**3. Rigorous Theoretical Foundation for MAP**
 Addressing concerns about heuristic behavior (Reviewers Z6dS, 4wzp), we provided a complete theoretical analysis (Theorems 1–3). We prove that the **Model Accuracy Predictor (MAP)** is well-posed and statistically consistent, ensuring convergence to the global optimum even with noise from fast fine-tuning.

**4. State-of-the-Art Performance and Practical Efficiency**

- **Performance:** BoundaryDPT delivers **1.6× lossless speedup** on DeiT-B (SOTA). With width pruning (BoundaryDPT+), we achieve **5.44× speedup** for extreme ViT compression.
- **Cost:** Constructing the MAP and determining pruning locations takes **14 hours on a single GPU** (or **under 2 hours on 8 GPUs**). Fine-tuning cost is strictly matched to the sota method NOSE , ensuring fair comparison.

**5. Improved Presentation and Logical Flow**
 We took feedback on clarity and structure very seriously (Reviewers JuCa, Z6dS) and performed a comprehensive revision:

- **Clear Narrative:** The Introduction and Method sections now follow a clean progression: *Observations → Challenges → Design Principles → Method*.
- **Clarity and Precision:** All mathematical notation was standardized, and key concepts were explicitly defined.
- **Completeness:** Detailed algorithm descriptions and proofs were moved to the Appendix for reproducibility without burdening the main text.

We believe the revised manuscript presents both a fundamentally new perspective on ViT redundancy and a rigorous, polished, and practically impactful pruning method. We hope that BoundaryDPT will be considered favorably.

Best regards,
The Authors

---

### Meta-Review · Area_Chair_xpZt · 2026-01-12

**Summary:**

Most reviewers consider the innovation of this work limited, including the overall idea appears to be preexisting, the contribution is quite incremental; while the motivation is clear, the technical novelty is only moderate. The authors have addressed these concerns in the rebuttal, but the moderate novelty remains an issue.

The reviewers also pointed out the unclear structure and layout, poor presentation, and informal writing of the paper. Although the author responded to this aspect, the rebuttal seems to suffer from the same problems of being complex and disorganized. For example, Reviewer JuCa did not pose explicit questions but only highlighted weaknesses, yet the response was divided into Q1 through Q6, making it difficult to follow and compare.
Overall, this part has been partially addressed and revised, but it is still not fully complete or satisfactory.

**Reviewer Concerns:**

Other concerns include training cost, limitations of the approach, clarity of method explanation, theoretical grounding of the Model Accuracy Predictor (MAP) polynomial approximation, the rationale for pruning nonlinear layers, overlooking residual and LayerNorm structures in ViT, issues with the core claim, insufficient evaluation of performance in detection and segmentation, and challenges in reproducibility. They have mostly been addressed. A limitation is that the current method is applied only to ViT; its application to Transformers in other task domains, such as natural language processing, remains unjustified.

**Reviewer Scores:**

Due to the issue of novelty, I believe Reviewers Z6dS and JuCa will maintain their scores. Reviewers Dp35 and 4wzp are also likely to maintain their scores because of concerns about novelty and limitations. A few reviewers may slightly increase their scores, but the average would still not be high enough for acceptance.

---

### Decision · Program_Chairs · 2026-01-26

Reject